# Mapping Arctic landfast sea ice stability with Sentinel-1 interferometry

Dyre O. Dammann[1], Leif E.B. Eriksson[1], Andrew R. Mahoney[2], Hajo Eicken[3], Franz J. Meyer[2]

[1]Department of Space, Earth and Environment, Chalmers University of Technology, Gothenburg, 412 96, Sweden
[2]Geophysical institute, University of Alaska Fairbanks, Fairbanks, 99775, USA
[3]International Arctic Research Center, University of Alaska Fairbanks, Fairbanks, 99775, USA

*Correspondence to*: Dyre O. Dammann (dyre.dammann@chalmers.se)

**Abstract.** Arctic landfast sea ice has undergone substantial changes in recent decades affecting ice stability with potential impacts on ice travel by coastal populations and on industry ice roads. The role of landfast ice as an important habitat has also evolved. We present a novel approach to evaluate landfast sea ice stability on a pan-Arctic scale using Synthetic Aperture Radar Interferometry (InSAR). Using Sentinel-1 images from spring 2017, we discriminate between bottomfast, stabilized, and non-stabilized landfast ice over the main marginal seas of the Arctic Ocean (Beaufort, Chukchi, East Siberian, Laptev and Kara Seas). The approach draws on evaluation of relative changes in interferometric fringe patterns. This first comprehensive assessment of Arctic bottomfast sea ice extent revealed that by area, most of the bottomfast sea ice is situated around river mouths and coastal shallows in the Laptev and East Siberian Seas, covering roughly 4.1 and 5.1 thousand km$^2$ respectively. These seas also contain the largest extent of stabilized and non-stabilized landfast ice, but are subject to the largest uncertainties surrounding the classification scheme. Even so, we demonstrate the potential for using InSAR in assessing the stability of landfast ice in several key regions around the Arctic, providing a new understanding of how stability may vary between regions. InSAR-derived stability may serve as a strategic planning and tactical decision-support tool for different uses of coastal ice. In a case study, we examined an ice arch situated in Nares Strait demonstrating that interferograms may reveal early-warning signals for the break-up of stationary sea ice.

## 1 Introduction

### 1.1 Landfast sea ice stability and stakeholder dependence

Sea ice is an important component of Arctic ecosystems and provides important services as a climate regulator (Screen and Simmonds, 2010), habitat for marine biota (Thomas, 2017), and a platform for coastal populations (Krupnik et al., 2010). During the last century, an expansion of transportation and resource extraction have led to increased human presence in the Arctic and further diversification of ice use (Eicken et al., 2009). The recent retreat of sea ice observed throughout the past several decades (Stroeve et al., 2012;Comiso and Hall, 2014;Meier et al., 2014) has already resulted in widespread consequences for ice users (ACIA, 2004;Aporta and Higgs, 2005;Fienup-Riordan and Rearden, 2010;Orviku et al., 2011;Druckenmiller et al., 2013) and increasing hazards (Ford et al., 2008;Eicken and Mahoney, 2015). At the same time, the related increased accessibility to Arctic waters (Stephenson et al., 2011) is leading to more ship traffic and resource exploration (Lovecraft and Eicken, 2011;Eguíluz et al., 2016). It is recognized that the sea ice conditions for future Arctic marine operations will be challenging and will require substantial monitoring and improved regional observations (Arctic Council, 2009) at the scale necessary for assessing environmental hazards and effective emergency response (Eicken et al., 2011).

Most of the Arctic Ocean is dominated by drifting pack ice, whereas stationary landfast ice occupies much of the Arctic coastlines roughly between November and June depending on location (Figure 1) (Yu et al., 2014). Sections of landfast ice, often several km to up to hundreds of km wide, are held in place by grounded ridges, islands, or coastline morphology, such as embayments or

fjords. Similar to the drifting pack ice, landfast ice has declined significantly during the last few decades, in particular in terms of delayed freeze up in the Beaufort (Mahoney et al., 2014) and Laptev (Selyuzhenok et al., 2015) Seas as well as significantly reduced extent in the Chukchi Sea (Mahoney et al., 2014). Later freeze up critically impacts stakeholders through reduced stability of the landfast ice in response to fewer grounded ridges that can withstand wind, ocean, or ice forcing (Dammann, 2017). Previous

research suggests that landfast ice stability can be expressed in terms of the combined frictional resistance provided by relevant grounding or attachment points (e.g., islands and grounded ridges) (Mahoney et al., 2007;Druckenmiller, 2011). Although the landfast ice is stationary, it deforms at the cm- to m-scale on timescales of days to months due to forcing from wind, currents and drifting ice (Dammann et al., 2016). The stability in part determines the rate at which the ice deforms and ultimately the severity of break-out events or magnitude of structural defects. We suggest that landfast sea ice can be further categorized into three regimes,

defined through their respective stability. These categories include (1) bottomfast ice, (2) floating ice enclosed in lagoons or fjords or sheltered by point features such as grounded ridges or islands, and (3) floating ice extensions (Table 1). A typical landfast ice regime is illustrated in Figure 2, where the stability of the landfast ice area decreases from the coast towards the open ocean (Dammann et al., 2016).

Bottomfast sea ice can grow laterally to the km-scale during winter depending on local bathymetry (Solomon et al., 2008;Stevens et

al., 2010). The bottomfast ice allows for heat loss from the sea floor and is therefore an integral part of aggregating and maintaining subsea permafrost (Solomon et al., 2008;Stevens et al., 2008;Stevens et al., 2010;Stevens, 2011) and controlling coastal stability/morphology (Are and Reimnitz, 2000;Eicken et al., 2005). Bottomfast ice is also relevant for fish as it reduces habitable shallow waters during winter (Hirose et al., 2008). Bottomfast ice is also of importance for on-ice operations as it can support a much larger load than floating ice. High to moderately stable landfast ice is of relevance to industrial (Potter et al., 1981) and

subsistence ice use (Druckenmiller et al., 2013), as well as for habitats (Tibbles et al., 2018). For instance, ringed seals are dependent on stable landfast ice for denning (Smith, 1980). Low-stability ice is potentially relevant for ocean-based operations such as shipping through trans-Arctic passages close to the coast where patches of landfast ice occasionally break off and drift into nearby shipping lanes, potentially causing hazards. Even areas hundreds of km from landfast ice can be impacted through the failure of ice arches.

Ice arches may be considered as an additional zone of "temporarily stabilized pack ice". Ice arches form when ice passing through a narrow passage experiences flow stoppage as a result of confining pressure and behaves like landfast ice, though potentially without cohesive strength between individual floes (Hibler et al., 2006). Ice arches typically form between November and March (Moore and McNeil, 2018) and can block export of ice through straits as wide as 100 km (Melling, 2002). When formed, such arches represent a significant obstacle to marine traffic due to the high confining pressures that make icebreaking impossible for

all but the most powerful vessels. The arches can in some locations prevail into the following season (Melling, 2002), but typically collapse in July – August (Kwok, 2005). Conversely, their break-up can lead to advection of large amounts of thick multiyear ice into high-traffic shipping routes (Barber et al., 2018) which is a well-known hazard for shipping (Bailey, 1957;Wilson et al., 2004;Howell et al., 2013). Stability is also of relevance for destinational cargo shipping in the Arctic as less stable, thinner ice is easier to break through resulting in opportunities for docking in areas of landfast ice.  For navigating through landfast ice,

stabilization through ridging is also important to identify since ridges can be problematic to navigate and are often associated with hazards (Hui et al., 2017).

## 1.2 Remote sensing of landfast ice stability

Satellite remote sensing is an important tool for measuring ice conditions in the Arctic, including mapping of landfast ice (Muckenhuber and Sandven, 2017). Optical/thermal satellite data such as from the Advanced Very High Resolution Radiometer

(AVHRR) were used to produce operational ice charts until the early 1990s when SAR was introduced into the charting production (Yu et al., 2014) as a superior data set due its independence of light and weather conditions and due to its higher (~100 m) resolution, both advantageous to stakeholders (Eicken et al., 2011). Different techniques exist to map the boundary of landfast sea ice, typically derived by evaluating unchanged sections of ice between consecutive SAR backscatter scenes (Johannessen et al.,

2006;Giles et al., 2008;Mahoney et al., 2014). In addition to its use in mapping of landfast ice, SAR backscatter can also discriminate between multiyear and first-year ice (Onstott, 1992) and identify different roughness regimes (Dammann et al., 2017). SAR has also been used to estimate the advection of ice through straits in the Canadian Archipelago (Melling, 2002;Kwok, 2006;Howell et al., 2013). One location of particular interest is the Nares Strait situated in between Greenland and Ellesmere Island, which features a seasonal ice arch (Kwok, 2005;Kwok et al., 2010) with important implications on the multiyear ice budget

of the Arctic Ocean (Kwok et al., 2010). However, backscatter does not give information pertaining to ice stability of the landfast ice or "temporarily stabilized pack ice" since the internal movement of the landfast ice is too small (mm/day) to be identified with change detection.

SAR interferometry (InSAR) is a signal processing technique, which extracts the phase difference between two SAR images acquired from similar viewing geometries. This phase difference (typically referred to as interferometric phase) can either signify

topography if acquisitions are separated in space (i.e., non-zero perpendicular baseline) or measures the line-of-sight motion if acquisitions are separated in time (non-zero temporal baseline) (Bamler and Hartl, 1998;Ferretti et al., 2007). InSAR is dependent on similar acquisition geometry, scatterers, and atmospheric conditions at the time of the two acquisitions to retain a non-zero interferometric coherence, which ranges between 0 and 1. InSAR has been used to successfully map the boundary of landfast ice (Meyer et al., 2011) through identifying the ice that has not shifted more than a few meters over weeks and hence retain

interferometric coherence. InSAR has also provided information pertaining to landfast ice dynamics (Li et al., 1996;Morris et al., 1999;Vincent et al., 2004;Marbouti et al., 2017) and topography (Dammann et al., 2017;Dierking et al., 2017) by evaluating the phase change between acquisitions. InSAR has also been shown to reveal plausible rheologies for landfast ice (Dammert et al., 1998) and has been used to determine the origin of internal ice stresses (Berg et al., 2015). Combined with inverse modeling, InSAR also allows to determine ice deformation modes (Dammann et al., 2016), rates, and the associated stress and fracture

potentials (Dammann et al., 2018b).

These studies have demonstrated the potential of InSAR as a tool to assess landfast ice dynamics and stability through case studies and utility as a planning tool for on-ice operations (Dammann et al., 2018a;Dammann et al., 2018b). They also laid the foundation for applying InSAR on a larger scale, potentially as a mean to generate operational information products and evaluate long-term trends. We argue the utility of InSAR and potential applications also extend to maritime activities and shipping. In regards to the

latter, vessel traffic typically does not traverse landfast ice. However, the assessment of landfast ice stability and spatio-temporal extent can potentially aid management of conflicting ice uses such as in the case of the access route to the Voisey's Bay mine in the Canadian Arctic which cuts through landfast ice that is part of a traditional Nunatsiavummiut use area (Bell et al., 2014). For vessel traffic through ice-covered straits or archipelagos, the approach outlined here can also possibly help identify and evaluate hazards associated with ice arches.

The coverage and access to InSAR-compatible SAR scenes has been an obstacle in the past but has improved significantly since the launch of Sentinel-1. The suitability of Sentinel-1 for automatic SAR processing has been shown, e.g., in Meyer et al. (2015). Hence, we explore InSAR as a tool to provide pan-Arctic information relevant to subsea permafrost, biological habitats, and sea ice use. The goal of this work is to determine the Sentinel-1 interferometric data availability along substantial parts of the circumpolar coastlines, and explore applications to consistently map landfast sea ice stability in different geographic regions. We

further explore limitations of the technology and possible applications.

## 2 Data and methods

### 2.1 InSAR principles

The interferometric phase may be related to the lateral (e.g., thermal contraction or displacement due to compressional or shear forces) or vertical (e.g., through buckling or tidal displacement) sea ice motion occurring in between the acquisition times of the two InSAR images. A phase signature can sometimes also be attributed to factors not related to surface motion such as atmospheric phase delay. Of the phase change attributed to motion, only displacement in line-of-sight direction ($\Delta r_{LOS}$) results in a phase change $\Delta \Phi_{disp}$ according to $\Delta \Phi_{disp} = 4\pi \Delta r_{LOS}/\lambda$ . The observed phase is measured within the wrapped interval of $[0; 2\pi]$. The interferogram is a series of fringes representing the projection of the true three-dimensional ice motion onto the line-of-sight vector. The orientation of the fringes can be used to interpret the direction of the three-dimensional motion field. The fringe spacing is an indicator of the deformation rate. The interpretation of observed fringe patterns is, however, not straightforward, and it typically requires the use of an inverse model (Dammann et al., 2016). The interferometric phase values will only be useful if scattering elements remain largely unchanged throughout the time interval bracketed by the image pairs used in processing. Coherence is a measure of the quality of the interferogram. Coherence is in general high if scatterers remain unchanged and low if there is significant change in the scattering medium (Meyer et al., 2011).

### 2.2 Sentinel-1 data

This study utilizes Sentinel-1, a constellation of two C-band SAR systems (Sentinel-1A and B) operating since 2014 and 2016, respectively, with a repeat-pass interval of 6 to 12 days depending on if both satellites acquire data or only one of them. Owing to the free-and-open data policy and large spatial coverage, Sentinel-1 acquisitions were obtained for five marginal seas of the Arctic Ocean, enabling mapping of landfast sea ice on a pan-Arctic scale. All images used were captured in interferometric wideswath (IW) mode with a single-look resolution of roughly 3 m x 22 m in slant range and azimuth respectively and a ~250 km swath width. Images were almost exclusively acquired between March and May 2017 (see supplementary data for full list of images used). We generated a total of 52 interferograms that cover almost the entire continental coastlines of the Beaufort, Chukchi, East Siberian, Laptev, and Kara Seas. To reduce computational costs, we omitted Greenland, some island groups, and the Canadian Archipelago, which are characterized by extensive coastline lengths. The Alaskan and Russian coastlines have high economic significance for the shipping and natural resource industries. They feature dynamically diverse ice regimes and large areas of bottomfast ice are expected in these regions. Except for one approximately 50 km-long section of coast in the Kara Sea and the eastern Laptev Sea, multiple InSAR compatible pairs were available for the specified time frame. This allowed us to select interferograms centered around the end of April, when most Arctic landfast ice is at its maximum extent and thickness.

In addition to images obtained for mapping of stability zones, a series of six consecutive image pairs were acquired covering the breakup of an ice arch in Nares Strait during spring 2017. This failure event occurred relatively early as compared with past events (Kwok, 2005) partly in response to thinner ice conditions and northerly winds (Moore and McNeil, 2018). The image sequence featured a 6-day temporal baseline covering a timespan of 36 days. The resulting interferograms revealed the ice deformation around the location of fracture up until the failure event.

### 2.3 Data processing

The complex Sentinel-1 data was processed to obtain the backscatter in order to interpret features that sometimes can be visibly identified (e.g. the landfast ice edge, fracturing, and ice roughness and types). The data was further processed for interferometry. Depending on the perpendicular baseline, sea ice topography can have a modest impact on the phase difference. Due to the tight

baseline limits (<50 m standard deviation) of the Sentinel-1 constellation, and as sea ice topography rarely exceeds 10 m, impacts on the interferogram interpretation are minimal for the data shown here. In this work we predominately utilize acquisitions with a temporal lag of either 12 or 24 days depending on data availability. For this timespan, the coherence over landfast ice was found to be generally high due to its stationary nature. However, coherence loss was evident in some areas, in particular in the Chukchi

Sea, such as in the Kotzebue Sound. This was likely partly due to ice motion, subsurface thinning from river runoff, and low signal-to-noise ratio. Significant decorrelation can also occur in late spring as the onset of melt at that time causes substantial changes in the scattering medium. In this work, we obtained images as close to late April as possible. This timeframe was found to be ideal for our purpose as ice thickness is near its maximum leading to maximum stability without risking impacts from the onset of melt. To ensure a realistic representation of what an operationally-produced synoptic, contiguous pan-Arctic interferogram would look

like, we did not attempt to derive alternative interferograms in cases of low coherence.

All interferograms in this work were produced using a standard Sentinel-1 workflow in the Gamma software (Werner et al., 2000). The IW images initially consist of independent bursts and swaths which we combined to utilize the full extent of the acquisition. We further coregistered pairs of acquisitions to ensure that the images cover exactly the same area with sub-pixel accuracy. The images were then multi-looked by averaging 10 pixels in range and 2 pixels in azimuth, resulting in reduced speckle and a final

pixel spacing of roughly 23x28 m. Next, spectral filtering was performed to ensure both images comprise the same spectral range, reducing phase noise in the final interferogram. The interferometric phase was calculated for each pixel of the coregistered and filtered images. Furthermore, the expected phase ramp in cross-track direction from a stationary flat earth surface was removed. The phase noise of the final interferogram was reduced using an adaptive phase filter (Goldstein and Werner, 1998).

## 2.4 Mapping of landfast ice zones

In this work, we evaluate relative ice stability based on fringe spacings within individual interferograms. This allows us to identify variations within an area imaged under largely the same conditions. Trends from higher to lower fringe density will likely correspond to increasing ice stability. Therefore, interferograms can provide information related to the relative ice spatial variations. Meyer et al. (2011) demonstrated that interferometry can be used to map the landfast ice based on a coherent phase response. Their work also suggested that fringe patterns are significantly impacted by grounded ridges by reduced density of the

fringes. Furthermore, Dammann et al. (2018c) showed that bottomfast ice can be mapped based on a near-zero phase change where the ice is frozen to the seafloor. We build on this work by suggesting that InSAR can be used to map three different zones of relative stability: bottomfast ice, stabilized ice, and non-stabilized ice (Table 1).

These three zones are subjectively and manually mapped without the use of specific threshold values. Here, bottomfast ice is identified with near-zero phase change in the interferogram. It can often be distinguished from the adjacent floating ice commonly

featuring a non-zero phase change or low coherence (Figure 3a). Bottomfast sea ice appears with near identical phase values to low-lying coastal areas, but is discriminated from land by identifying the coastline using the backscatter signature in a composite image with backscatter and phase (Figure 3b). Subtle coastal features such as sediment bars are often not captured by the landmask (Wessel and Smith, 1996). This can give the wrongful appearance that (1) areas of near-zero phase should have been mapped as bottomfast and (2) bottomfast ice appear in sporadic areas along the coast separated by floating ice. We can often identify stabilized

ice by a stark fringe discontinuity between different fringe densities (Figure 3c). However, in some regions such as the Laptev and East Siberian Seas, changes in stability are more gradual between zones. Mapping of such regions are therefore more subjective and possibly less exact. In the case of lacking stark fringe discontinuities, stabilized ice is also mapped in regions featuring a very slight phase response with no clear fringe patterns similar to a pattern on freshwater lakes with thick ice (Figure 3d). Non-stabilized ice is mapped as the remaining ice featuring non-zero interferometric coherence and clear fringe patterns (Figure 3e). Only the

outer margin of the stabilized and non-stabilized ice is mapped and the coastal boundary represents either the boundary of another stability zone or the coastline as represented by the landmask.

The zones themselves are based on relative stability in terms of whether the ice is anchored or sheltered. Determining absolute stability (i.e. whether an area is stable enough for a specific use, such as ice roads) would be problematic to determine from fringe density alone. This is because there are many factors that affect fringe density in addition to stability, including changing wind and ocean currents, satellite viewing geometry, and the prevalent mode of ice deformation (Dammann et al., 2016). A measure of whether the ice is stable would also depend on the specific stakeholders and their dependence on stability. As an example, on shorter time scales, industry ice roads would be able to accommodate less strain than community ice trails due to different mode of transportation and user specific needs. Further steps to identify such thresholds are outlined in Dammann et al. (2018a).

## 2.5 Validation areas and data

The Beaufort Sea coast of Alaska was used for validation, as the sea ice in this area includes all three landfast ice regimes (bottomfast ice, stabilized ice and non-stabilized ice), and ample validation data is available from previous landfast sea ice studies. Alaska's Beaufort Sea coast is of major interest in the context of local and indigenous ice use as well as industry resource exploration and extraction. Some areas along this coastline feature similar landfast ice extent over time scales from months to years. It was found that these regions ("nodes") of consistent landfast ice extent are often tied to the location of the 20-m isobath, a water depth associated with grounding of pressure ridges stabilizing the landfast ice (Mahoney et al., 2014). Indigenous knowledge and a field study also indicate persistent grounded ridges in the location of the node closest to Utqiaġvik, Alaska (Meyer et al., 2011). We also evaluated our approach near Stolbovoy Island in the Laptev Sea. This area features a shoal of < 10 m water depth leading to earlier formation of fast ice than the surrounding areas (Selyuzhenok et al., 2015) likely due to the formation of grounded ridges on the shoal resulting in increased stability.

## 3 Results

### 3.1 Evaluating landfast ice stability

We constructed a series of Sentinel-1 interferograms along the coastlines of five marginal seas of the Arctic Ocean during 2017: the Beaufort, Chukchi, East Siberian, Laptev, and Kara Seas. As seen in the in the interferograms (Figures 4-8), the landfast sea ice varies substantially between the seas in terms of the extent and interferometric fringe density.

The landfast sea ice extent in the Beaufort Sea ranges from almost zero up towards 100 km (Figure 4a). River outlets such as the Colville and Mackenzie Deltas feature extensive regions of bottomfast ice several kilometers wide (Figure 4b). Bottomfast ice is also prominent in many lagoons along the coast. Much of the floating ice along the western coast of the Beaufort Sea from Foggy Island Bay (east of Prudhoe Bay) to Point Barrow by Utqiaġvik is stabilized. The floating ice can be identified by ice shoreward of a stark fringe discontinuity separating regions of different fringe density and stability (Figure 4a,b). The line of discontinuity features several seaward points (see arrow in Figure 4b) consistent with the expected stabilization of the ice cover immediately shoreward of grounded ridges. Similar patterns are also apparent near the Mackenzie Delta (Figure 4c). The landfast ice in the eastern part of the Beaufort Sea also consists of large areas of stabilized ice. Here, the landfast ice is noticeably sheltered by land features resulting in lower density fringes directly downstream from land (Figure 4d).

The landfast sea ice in the Chukchi Sea is generally less extensive than in the Beaufort Sea, particularly along the Russian coast (Figure 5a). Bottomfast ice in the Chukchi Sea is constrained mostly to lagoons. Some of the lagoons, such as the Kasegaluk, consist almost exclusively of bottomfast ice (Figure 5b). Only a few areas in the landfast ice appear to be stabilized, including the

northern coast of Alaska near Peard Bay (Figure 5c) and the Bearing Strait (Figure 5d). The Chukchi Sea consists predominantly of non-stabilized ice with the most extensive region of landfast ice situated off shore from the village of Shishmaref (Figure 5d). The Chukchi Sea features consistent coherence loss in several regions such as the Kotzebue Sound (Figure 5a).

The landfast ice in the East Siberian Sea is more extensive than in the Chukchi and Beaufort Seas and can extend over 100 km from shore (Figure 6a). Bottomfast ice is also more extensive than in the Beaufort and Chukchi Seas. The bottomfast ice in the East Siberian Sea follow several sections of coastline even tens of km away from major rivers. Even so, most of the bottomfast ice is situated near the Kolyma and Indigirka Deltas (Figure 6b). In contrast to the Beaufort and Chukchi Seas, stabilized ice extends several tens of km offshore without being sheltered by coastline morphology or islands (Figure 6c). These large areas also lack clear indications of the presence of grounded ridges as found by smaller areas of stabilized ice (Figure 6d) and in the Beaufort Sea.

The landfast ice in the Laptev Sea, similar to the East Siberian Sea, extends upwards of 100 km from shore (Figure 7a). In the Laptev, most of the bottomfast ice is situated around river outlets and in particular near the Lena Delta extending tens of km from shore (Figure 7b). The delta features a large amount of small low-laying land areas (e.g. gravel islands) only partly covered by the landmask. This made it problematic to delineate all areas of bottomfast ice and led to more approximate delineations than in the other deltas mapped. On the east side of the Lena Delta and south of the Great Lyakhovsky Island, there are extensive sections of stabilized ice (Figure 7c). Some regions of the eastern Laptev Sea lack a clear discontinuity, but at the same time feature locally reduced fringe density, indicative of stabilized ice (Figure 7c). We also consider these areas to be stabilized (Figure 7c), but possibly as a result of different ice type or thickness rather than through grounding or sheltering. However, one offshore area is clearly identified as stable by a lack of consistent fringe patterns and a clear discontinuity likely due to grounded ridges (Figure 7d).

The landfast ice in the Kara Sea features much smaller landfast ice extent than the other Russian Seas (Figure 8a). Bottomfast ice is also much less prevalent and largely situated by the Payasina River (Figure 8b). The landfast ice extends tens of km from shore predominately in areas supported by offshore islands and archipelagos (Figure 8c and d). The ice surrounding the archipelagos is largely non-stabilized, but the ice confined by the islands is predominately stable (Figure 8c).

## 3.2 Large-scale mapping of stability zones

Interferograms enabled the mapping of landfast ice stability zones based on subjective interpretation of interferometric fringes (Figure 9). The resulting stability map allows for a large-scale comparison and analysis of bottomfast, stabilized, and non-stabilized landfast ice within and between the different seas. For this comparison, we have listed the 2017 area extent of each stability zone and marginal sea in Table 2. However, it is important to note that this list is not complete because the analysis omitted some island groups and included some data gaps.

Most areas with extensive bottomfast ice reaching several km from shore are located either in the vicinity of river deltas or within lagoons (Figure 9). However, a prominent exception is the coastline of the western East Siberian Sea, where our analysis shows substantial area extent of bottomfast ice even tens of kilometers away from any major rivers. The East Siberian Sea with its three large river systems (the Indigirka, Bogdashkina, and Kolyma Rivers) contains the most bottomfast ice of the regions considered here with $5.1 \times 10^3$ km$^2$. The Laptev Sea also contains a large fraction of the Arctic bottomfast sea ice with $4.1 \times 10^3$ km$^2$, largely concentrated around the Lena and Yana Deltas. The Chukchi Sea features extensive bottomfast ice ($1.8 \times 10^3$ km$^2$), but almost exclusively within large lagoons such as the Kasegaluk Lagoon. The bottomfast ice in the Beaufort Sea coast ($2.5 \times 10^3$ km$^2$) can be found in lagoons and around the Colville and Mackenzie Deltas. The Kara Sea contains a bottomfast ice extent comparable with the Beaufort Sea with $2.6 \times 10^3$ km$^2$.

Stabilized ice was found in all marginal seas (Figure 9), though their relative contributions to overall landfast ice extent varied widely. The largest extent of stabilized landfast ice in our study region are found in the Laptev and East Siberian Seas featuring a

total areal extent of $74 \times 10^3$ km$^2$ and $45 \times 10^3$ km$^2$ respectively. These regions feature particularly large continuous areas of stabilized ice labeled A-F in Figure 9. Even so, as we delineate here, the Beaufort Sea is the only sea that features more stabilized ice than non-stabilized ice featuring an areal fraction (stabilized ice / non-stabilized ice) of 0.86. This is likely attributed to the large grounded sections as well as areas sheltered by coastal morphology resulting in $35 \times 10^3$ km$^2$ stabilized ice. The Laptev Sea also features large areas confined by coastlines. However, in the Laptev sea, these regions also commonly feature non-stabilized ice. A large part of the landfast ice in the Kara Sea is mapped as stabilized ($16 \times 10^3$ km$^2$), largely due to the fraction of landfast ice situated between islands and archipelagos. With a relatively narrow landfast ice extent as compared to other seas and absence of regions of sheltered ice, the Chukchi Sea contains the lowest total extent of stabilized ice with $4.6 \times 10^3$ km$^2$.

In the Chukchi Sea, we identified the vast majority of the landfast ice as non-stabilized with $29 \times 10^3$ km$^2$ (Figure 9) resulting in the largest areal fraction (stabilized ice / non-stabilized ice) with 5.4. However, the largest areas of non-stabilized ice can be found in the Laptev Sea ($127 \times 10^3$ km$^2$) and the East Siberian Sea ($80 \times 10^3$ km$^2$). Here, the distinction between stabilized and non-stabilized landfast ice is not as straightforward as in the Beaufort and Chukchi Seas due to a lack of clear boundaries between areas of different fringe densities. Even so, it is clear that landfast ice extent in the East Siberian and Laptev Seas is dominated by vast areas of non-stabilized ice. However, unlike the Chukchi Sea, we also identified significant areas of stabilized landfast ice along these two seas. The Kara Sea features predominately non-stabilized ice ($37 \times 10^3$ km$^2$) along the coast and along the outer margins of archipelagos.

## 4 Discussion

### 4.1 Validating stability zones with areas of known ice stability

The technique to evaluate bottomfast sea ice was thoroughly validated in several regions by Dammann et al. (2018c). However, there is limited information that can be used to validate further stability classes, namely the separation between stabilized- and non-stabilized ice. Even so, we compare our mapping approach here with one region in the Beaufort and one in the Laptev Sea with known stabilized ice. We examine three acquisitions from 8-17 Apr along the Beaufort Sea coast. These images exhibit a sharp discontinuity in backscatter, which in this case can be used to identify the location of the landfast ice edge (see white arrows in Figure 10a). It is worth noting that relying on backscatter to discriminate landfast or drifting ice only works in cases where there are noticeable differences in backscatter between the landfast and drifting ice or when there is a severely deformed landfast ice edge as a result of shear interaction with the pack ice (Druckenmiller et al., 2013).

The landfast ice edge in this analysis is consistent with the three "nodes" (A-C) identified by Mahoney et al. [2007a; 2014] (Figure 10b). The nodes signify a persistent landfast ice edge. This is believed to be a result of reoccurring grounded ice features (Mahoney et al., 2014). The ice shoreward of these three nodes are expected to be stabilized because grounded ridges are known to stabilize the landfast ice leading to reduced strain shoreward of the grounding points (Mahoney et al., 2007; Druckenmiller, 2011). The interferograms exhibit a phase response suggesting stabilized ice directly shoreward of node A and C (Figure 10c). Here, node A is known to correspond to the location of large grounded ridges offering stability to the ice cover (Meyer et al., 2011). Node B and C are also expected to be regions of persistent grounded ridges since the nodes coincide roughly with the 20 m isobath (Mahoney et al., 2014). However, the ice directly shoreward of B appears non-stabilized and the stabilization occurs further in. This may be due to reduced keel depth of ridges in 2017 or possibly reduced grounding strength of ridges present in B. The border between stabilized and non-stabilized ice feature multiple curves towards land ending in seaward points (see black arrows in Figure 10d). At these points, the stability is higher than adjacent areas with the same distance from shore. This is consistent with increased stability behind grounded ridges. Although the landfast ice edge can in some instances be mapped using a single backscatter image,

the mapping of stabilized ice cannot easily be discriminated. This is apparent when comparing grounding locations as obtained with InSAR with the backscatter images (see black arrows in Figure 10a).

Similar patterns indicating grounded ridges were found in the Laptev Sea where an April interferogram exhibits a section of stabilized ice roughly 100 km off shore (see "A" in Figure 11). The full area extent of the stabilized ice cannot be established due to limited data availability in the region and thus the surrounding interferogram had to be acquired as early as February before this region had stabilized. Stabilized ice is expected in this region, which features a large shallow shoal, earlier ice formation, and grounded ridges (Selyuzhenok et al., 2015). The location of this shoal along with smaller ones are obtained from Jakobsson et al. (2012) and displayed in Figure 11b. Here, it is apparent that even some of the smaller shoals are associated with stabilized ice (see "B" and "C" in Figure 11b). It is also clear that the extensive stabilized ice that stretches out halfway between Great Lakhovsky Island and Stolbovoy Island is potentially anchored between the coast and shallow areas (see "D" in Figure 11b).

## 4.2 Methodological limitations for mapping stability zones

There are a number of sources of uncertainty that affect our map of landfast ice and its relative stability. Dammann et al. (2018c) determined that in some instances, bottomfast ice has to be approximated on the sub-km-scale due to ambiguities associated with low fringe density or fringes parallel to the bottomfast ice edge. We also acknowledge that small islands or sandbars not represented by our landmask may be erroneously identified as bottomfast ice. However, we greatly reduced such errors by not mapping areas that appears to be low-laying land in the SAR backscatter images. In areas where the landmask does not appear to fit the coastline due to errors or coastline changes, mapping the intricate coastal morphology can be a time-consuming task, hence mapping on a pan-Arctic scale will inevitably contain inaccuracies.

In this work, we did not apply strict mapping thresholds to distinguish between stabilized and non-stabilized ice, but rather made subjective determinations based on fringe patterns. This approach works well in the Chukchi and Beaufort Seas, where regions of low fringe density lie adjacent to the coast or bottomfast ice and can be easily distinguished from regions of higher fringe density. However, in some regions, especially in the Russian Arctic, there is often a lack of distinct boundaries between regions of different fringe spacing, introducing ambiguities between stabilized and non-stabilized ice on scales from km to even tens of km (Figure 6c and 7c). The difficulty distinguishing these two zones may result from reduced pack ice interaction along the Russian shelf, given the predominately divergent ice regime (Reimnitz et al., 1994;Alexandrov et al., 2000;Jones et al., 2016) in contrast to the western Arctic. Such ice regime is expected to feature reduced dynamically-induced strain (and therefore fewer interferometric fringes) in non-stabilized ice making it appear more stable. This is visible in the different fringe densities of the non-stabilized ice in Figure 4d and 6d). Additionally, the greater extent of landfast ice on the shoreward side of grounding points provides a greater fetch, which may cause stabilized ice on the Russian Shelf to exhibit higher fringe densities than in the Chukchi or Beaufort Seas. This suggests, that there is likely a spectrum of landfast ice stability. Additional zones may be necessary to fully characterize the landfast ice regimes in different regions for different ice uses or research aims. Expanding upon the classes presented here would likely require a different set of evaluation criteria for fringes depending on regions. Additional data such as bathymetry would also likely strengthen such analysis.

We focused on some examples with possibly suboptimal classification. One potential candidate for reclassification is landfast ice in sheltered bays such as the Khatanga Gulf in the western Laptev Sea, which exhibited predominantly high fringe densities (Figure 7a). Hence, the Khatanga Gulf was identified largely as non-stabilized despite being nearly landlocked (Figure 9). Due to the shallow water in this region, it is likely that the high fringe density is caused in part by vertical motion associated with tides and coastal set up. Since vertical motion has less impact on stability in well-confined landfast ice, such examples suggest the potential need for an additional zone of stability that allows higher fringe densities in coastally confined regions. Such additional

classification would depend on other datasets such as a landmask or bathymetry to identify level of restricted ice movement in response to likely forcing conditions. Another, larger-scale example is the eastern Laptev sea, which is an area of landfast ice sheltered by the New Siberian Islands and is typically considered stable (Eicken et al., 2005). However, based on relatively high fringe density, in particularly offshore of the Lena Delta, we classify much of the landfast ice in this region as non-stabilized

(Figure 9). This suggests that the criteria for stabilized ice used in this analysis is different than in Eicken et al. (2005) and can provide new information related to stability in the region. Based on the overall fringe counts and patterns, the majority of the phase response is due to lateral displacement and potentially only partially due to vertical displacement (circular fringe patterns with low density – see Dammann et al. (2016)) due to tidal motion. It is possible that landfast ice in this region may be less stable than previously thought and a "partially stabilized" zone may be appropriate. This would be consistent with a recent SAR backscatter

analysis of landfast ice in the Laptev Sea (Selyuzhenok et al., 2017), which showed that areas identified as landfast ice in operational ice charts may actually contain pockets of partly mobile ice. This was shown for the month after initial landfast ice formation, but could possibly result in more dynamic ice throughout spring due to reduced ice thickness.

Sensitivity to specific atmospheric and oceanographic conditions during the time period between SAR acquisitions may place a limitation on the number of stability zones that can be mapped. For example, in the absence of dynamic interaction with pack ice,

there may be little difference in fringe spacing between landfast ice seaward and shoreward of stabilizing anchor points. Without evaluating the phase response for each area of interest in detail during different forcing scenarios, it may be problematic to understand under what conditions the ice remains stable. Classification of stability based on relative differences in fringe density is also complicated by the use of non-simultaneous interferograms to provide complete coverage of a region. The interferograms used here were obtained as close to maximum ice extent and stability as possible (roughly late April), but sometimes had to be

obtained as early as February. Fringe density tends to decrease over the winter as the ice thickens (Dammann et al., 2016). Hence, the use of interferograms based on different dates can aid interpretation by confirming consistent fringe patterns and discontinuities that identify temporal changes. Temporal changes result in phase discontinuities at the image stitchings not related to different stability zones, which further complicates the mapping process.

Sentinel-1 IW imagery is predominately acquired over land, so it is likely not possible to construct interferograms away from the

coast to cover extensive landfast ice approaching the 250 km IW swath such as that in the East Siberian Sea. The data availability of these images further restricts the temporal baseline between images to a minimum of 12 days, shorter than past work to identify landfast ice (Mahoney et al., 2004; Meyer et al., 2011; Dammann et al., 2016). Further studies should investigate the effect of different temporal baselines on the stability product. A shorter baseline will result in higher temporal resolution. However, with a shorter baseline (e.g. Sentinel-1 6-day baseline), the mapping of the seaward landfast ice edge may incorporate stationary pack ice.

A longer baseline will result in lower interferometric coherence. With a 12-day baseline, some regions already feature consistent coherence loss such as the Kotzebue Sound region. Such regions can most often be identified through a spatially inconsistent progression from high to a complete loss of coherence. In such cases, the mapping of landfast ice type boundaries is not possible. It is worth mentioning that this technique can only be used before the onset of melt when widespread coherence loss occurs. Therefore, it is not possible to evaluate the retreat of bottomfast ice or reduction of ice stability in response to melt.

**4.3 Temporarily stabilized pack ice**

Sentinel-1 SAR backscatter imagery captured the location and break-up of the ice arch in Nares Strait in 2017 (Figure 12). In this case, the arch appeared stable on 6 May (Figure 12b) before eventually failing sometime before 12 May (Figure 12c). As seen in the interferograms, however, the ice arch features various levels of cm- to m-scale deformation and fractures prior to break-up resulting in fringe discontinuities (Figure 13a) most pronounced near the arch terminus to the south.  Near the failure line, there is

no sign of a fringe discontinuity up until 12 April (Figure 13a) when the interferogram displays near cross-track parallel fringes indicating compression towards the terminus (Figure 13b). There is a significant contrast in fringe density on either side of the line which may be indicative of a fracture where ice to the west is being compressed more rapidly than the ice close to the coast. The interferogram between 18 – 24 April features widespread coherence loss possibly due to continued compression (Figure 13c). Deformation is less severe from 24 April when fringe density is significantly reduced. However, we notice a fringe discontinuity to the east of the failure line featuring perpendicular intermediate fringe patterns towards late April (see arrows in Figure 13d). These patterns develop further into circular patterns often associated with vertical lifts and depressions (Figure 13e) before the whole arch appears to fail through shear motion along this same fault (Figure 13f).

This example demonstrates that it may be possible to detect precursors to break-out events without rigorous inverse model-based interpretation of fringe patterns (e.g., Dammann et al., 2016;Dammann et al., 2018b), which does not easily lend itself to operational workflows. The evaluation of the interferograms leading up to the failure of the ice arch suggests that InSAR has the ability to inform stakeholders of changing stability and ice movement with potential value for an early warning system designed to alert ice users of hazards related to ice movement. Recent and ongoing sea ice decline is leading to an increasing presence of thinner ice in the Canadian Archipelago (Haas and Howell, 2015). Weaker ice due to warmer temperatures (Melling, 2002) may lead to earlier breaching of ice arches. This could result in a larger quantity of advected ice with potentially longer travel paths increasing the severity of such events (Melling, 2002;Barber et al., 2018).

**5 Conclusion**

In a time of rapidly changing sea ice conditions and continued interest in the Arctic by a range of stakeholders, we stress the need for new assessment strategies to enable safe and efficient use of sea ice. InSAR is gaining growing attention in the sea ice scientific community, and here we demonstrate its value for identifying zones of landfast ice stability. We are also highlighting application of InSAR in the development of operational sea ice information products for both long-term strategic planning as well as short-term tactical decisions. Using interferograms generated by a standardized workflow, we show that three stability zones of landfast ice can be identified based on fringe density and continuity, which are indicative of differential ice motion occurring between SAR acquisitions. Along the Beaufort Sea coast of Alaska, we find that the landfast ice regime can be well described with three stability zones: bottomfast ice, where the sea ice is frozen to or resting on the seabed; stabilized ice, which is floating but sheltered by coastlines or anchored by islands or grounded ridges; and non-stabilized ice, which represent floating extensions seaward of any anchoring points. This finding was supported by comparison with the location of stable "nodes" identified through analysis of hundreds of landfast ice edge positions over the period 1996-2008. Not only does this provide some validation of our results, but it demonstrates the ability of InSAR to capture useful information in just two snapshots compared to previously requiring analysis over many years.

Based on our findings, it is likely that InSAR-derived maps could provide substantial value as a stand-alone product for some regions such as the Beaufort Sea. With that said, the stability zones in the Beaufort Sea and the Russian Arctic appear to be qualitatively different. This makes it challenging to directly adapt the proposed scheme to the East Siberian and Laptev Sea, which is associated with substantial uncertainties. Even so, we have demonstrated the data availability and application of this InSAR-based approach, which can provide added value to ice charts and other products. In ice charting, multiple information products are evaluated with local knowledge to create final products. Similarly, the value of InSAR may be greatly enhanced by complimenting with other products (e.g. InSAR time series analysis, SAR-based and optical remote sensing products, local knowledge, coastal morphology and bathymetry, and atmospheric and ocean forcing data).

The use of a standardized workflow facilitates large-scale application of this approach, which we demonstrated on a near-pan Arctic scale using 52 Sentinel-1 acquisition pairs during spring 2017. This allowed us to map the same zones of landfast ice in the Beaufort, Chukchi, East Siberian, Laptev, and Kara Seas. To our knowledge, our results represent the first mapping of bottomfast ice extent at this scale and the first attempt at any scale to map the extents of different landfast ice stability zones. It also enabled us to estimate and compare the total area covered by each stability zone in each marginal sea. However, we note that these comparisons are based on the assumptions that the landfast ice regimes in all these seas can be well described by the same three stability zones. Although the delineation of different zones can be subjective, in particular in the Russian Arctic, our results clearly show that not all landfast ice is equally stable. Here, InSAR is potentially able to detect small-scale motions up to hundreds of km from shore that have previously been overlooked. In addition, there are uncertainties associated with the exact mapping of stability zones, in particular in terms of the exact delineation between stabilized and non-stabilized ice in the East Siberian and Laptev Seas. Here, the boundaries between stabilized and non-stabilized ice is more difficult to discriminate likely due to fewer pinning points where the ice is grounded or supported. Therefore, what we present here is not an operational ice chart, but we demonstrate the ability and application to discriminate stability classes on a pan-Arctic scale using InSAR.

The method presented in this work has a broad set of potential applications for monitoring including subsea permafrost, biological habitats both beneath and above the ice surface, and ice use by a range of stakeholders. Bottomfast ice is important because it helps aggregating subsea permafrost, which serves to constrain the location of permafrost-rich shorelines. Utilizing InSAR, it is likely possible to monitor changes in bottomfast ice over time with significant implications for erosion and spring flooding and the release of methane hydrates. With respect to ice users, sea ice navigation near or through landfast sea ice is presently predominately supported by sea ice charts that map areas occupied by landfast ice. However, the sea ice charts do not provide information as to the relative stability of the ice. The information provided here would likely be useful in the context of navigation and supporting on-ice operations. The InSAR-based approach described here can potentially provide support by identifying the following stability-related features:

1) Low-stability ice that may break off and drift into shipping lanes.
2) Grounded ridges that may be problematic for ice navigation, but at the same time may provide added stability for on-ice operations.
3) Stable areas to use for equipment staging by coastal community hunters and industry.
4) Bottomfast ice for development of ice roads for transportation of heavy loads.

We further demonstrate the scientific and operational value of InSAR over sea ice through the examination of interferograms of ice arches. In this context, they can be considered part of an additional stability zone of quasi-landfast ice (i.e. "temporarily stabilized pack ice"). Preliminary analysis of the Nares Strait ice arch in 2017 suggests that interferograms may reveal early-warning signals of an imminent break-up. The use of inverse modeling may further help derive the small-scale strain field from interferograms which may improve our ability to predict their failure. We expect that InSAR can provide valuable information for stakeholders enabling tracking of ice dynamics and stability on seasonal timescales. The ability to provide stability information to stakeholders also opens up for development of operational guidelines in terms of what stability zones should be prioritized or avoided.

This work builds on previous applications of InSAR to study landfast ice and demonstrates what can be achieved over large areas with a standardized workflow. 2017 was the first year Sentinel-1 covered the Arctic coast with IW images necessary for this analysis. If this coverage continues, there will be considerable opportunity for development beyond what is presented here, including development of automated methods for mapping and classifying landfast ice suitable for incorporation into operational ice charts. Furthermore, through additional analysis of landfast ice and ice arches subject to different forcing conditions, we

anticipate improving our understanding of stabilizing and destabilizing mechanisms. This would allow improved prediction of formation and break-up. Not only will this enhance operational sea ice information available to stakeholders, but it also allow us to better understand the response of coastal sea ice to a changing Arctic environment.

## Acknowledgements

This work was funded by the Swedish National Space Agency (Dnr. 192/15). Sentinel-1 data are provided free of charge by the European Union Copernicus program and were downloaded through Alaska Satellite Facility (ASF). We thank Bill Hauer (ASF) for valuable data support and Christopher Stevens (SRK Consulting) and Joost van der Sanden (Canada Centre for Mapping and Earth Observation) for valuable guidance. We thank two anonymous reviewers who substantially contributed to improving this manuscript.

## 10 Competing interests

The authors declare that they have no conflict of interest.

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

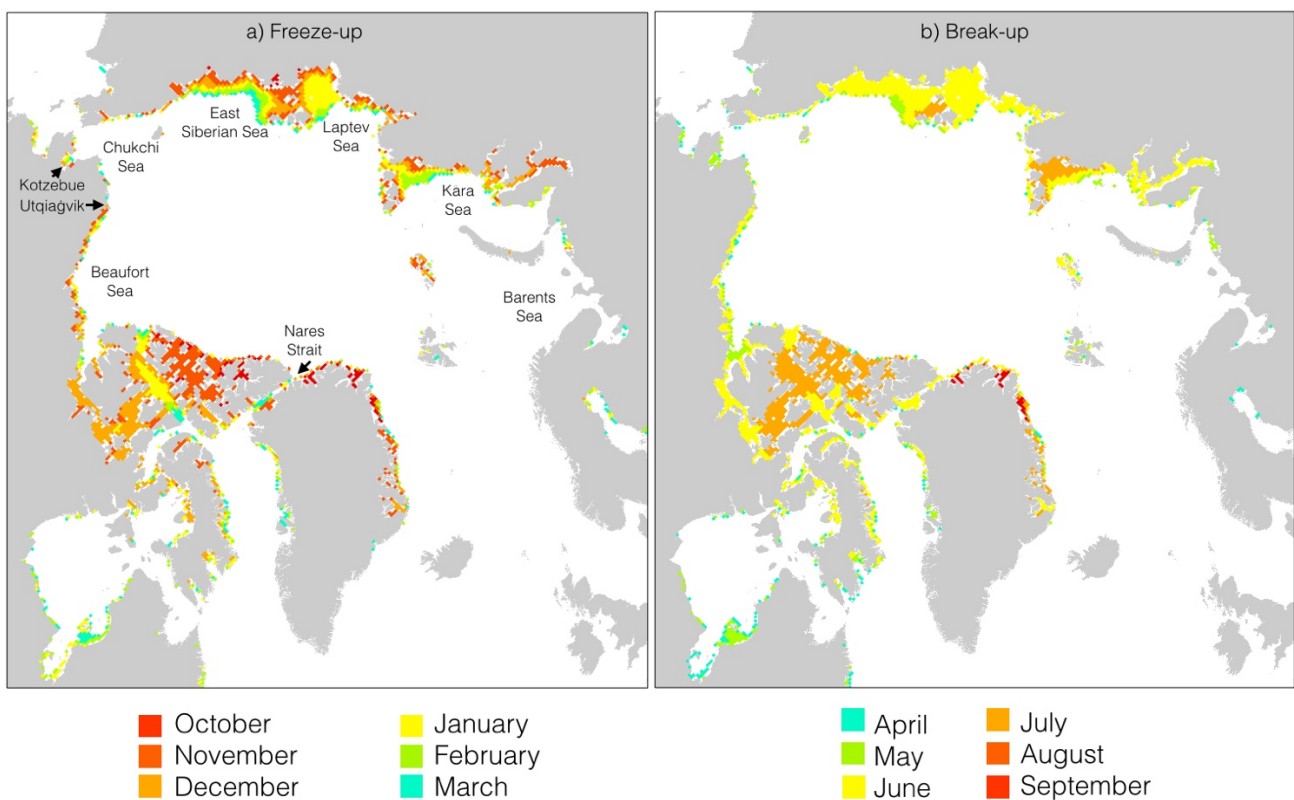

**Figure 1: (a) Oct - Mar (Freeze-up) and (b) Apr - Sep (break-up) monthly mean landfast sea ice extent (1976 - 2007) derived from sea ice charts based on optical instruments and SAR. The data for this figure was obtained from the National Snow and Ice Data Center (Yu et al., 2014).**

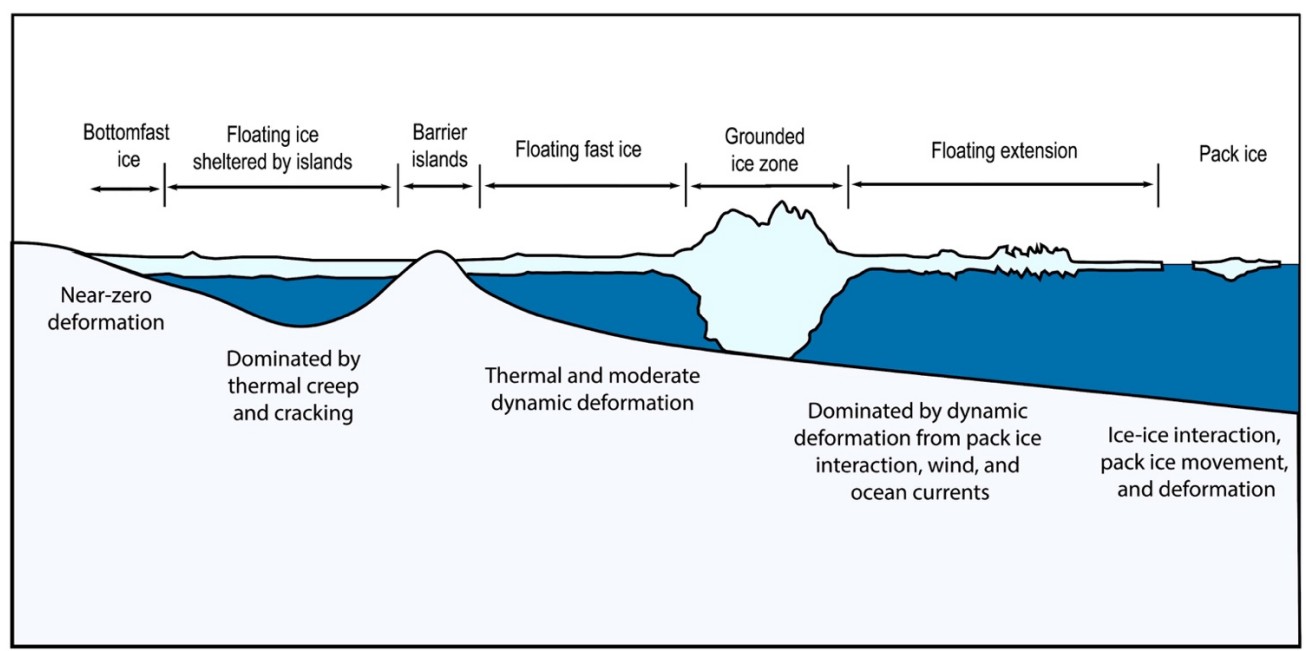

**Figure 2: Conceptual scheme of landfast sea ice where different regimes possess different levels of stability.**

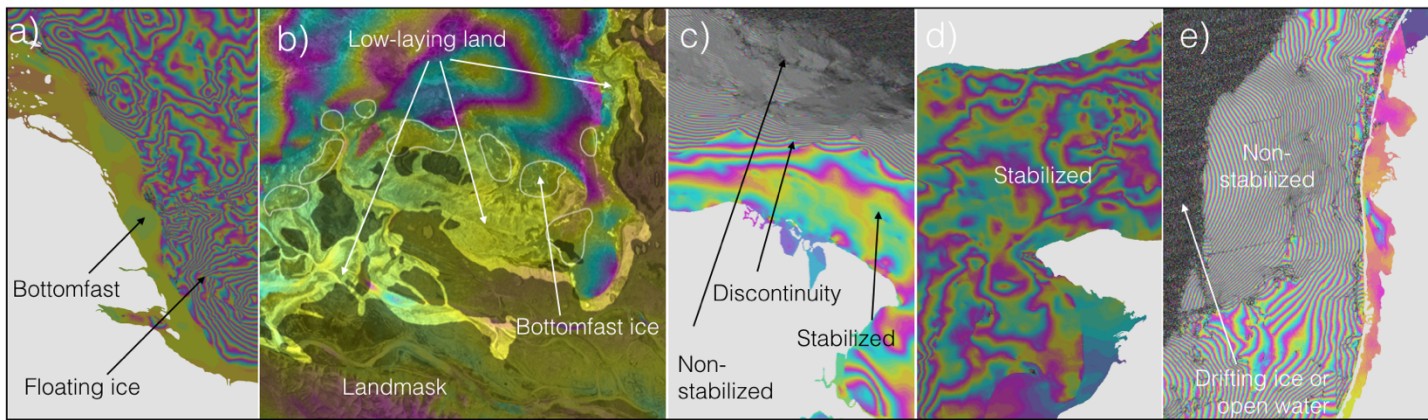

5    **Figure 3: (a) example of interferometric phase response over bottomfast ice. (b) example of a poor match between the landmask (transparent black shading) and low-laying coastal areas near a delta hence bottomfast ice (white outline) had to be mapped against the coastline as identified in a phase/backscatter composite. (c) example of stabilized ice as identified based on a phase discontinuity. (d) example of stabilized ice as identified by low fringe density and non-consistent fringe patterns. (e) example of non-stabilized ice as identified by high fringe density. Land is masked out in light gray in a,c,d, and e.**

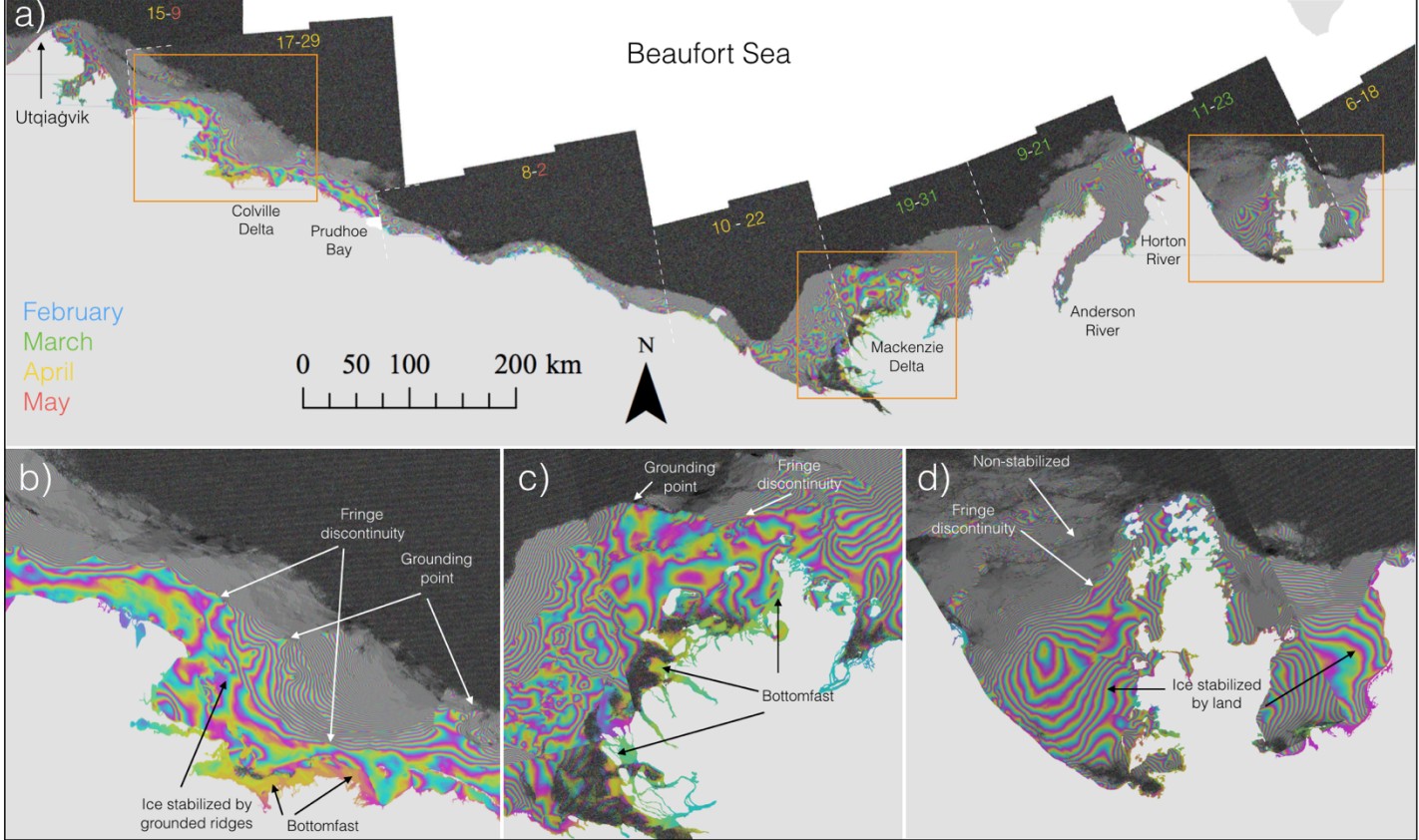

**Figure 4: (a)** Sentinel-1 interferograms derived from image pairs acquired over the Beaufort Sea between March and May, 2017. Numbers on images represent date ranges where the colors blue, green, yellow, and red signify the months of February-May respectively. **(b)-(d)** represent three enlarged areas identified in (a) further discussed in the text.

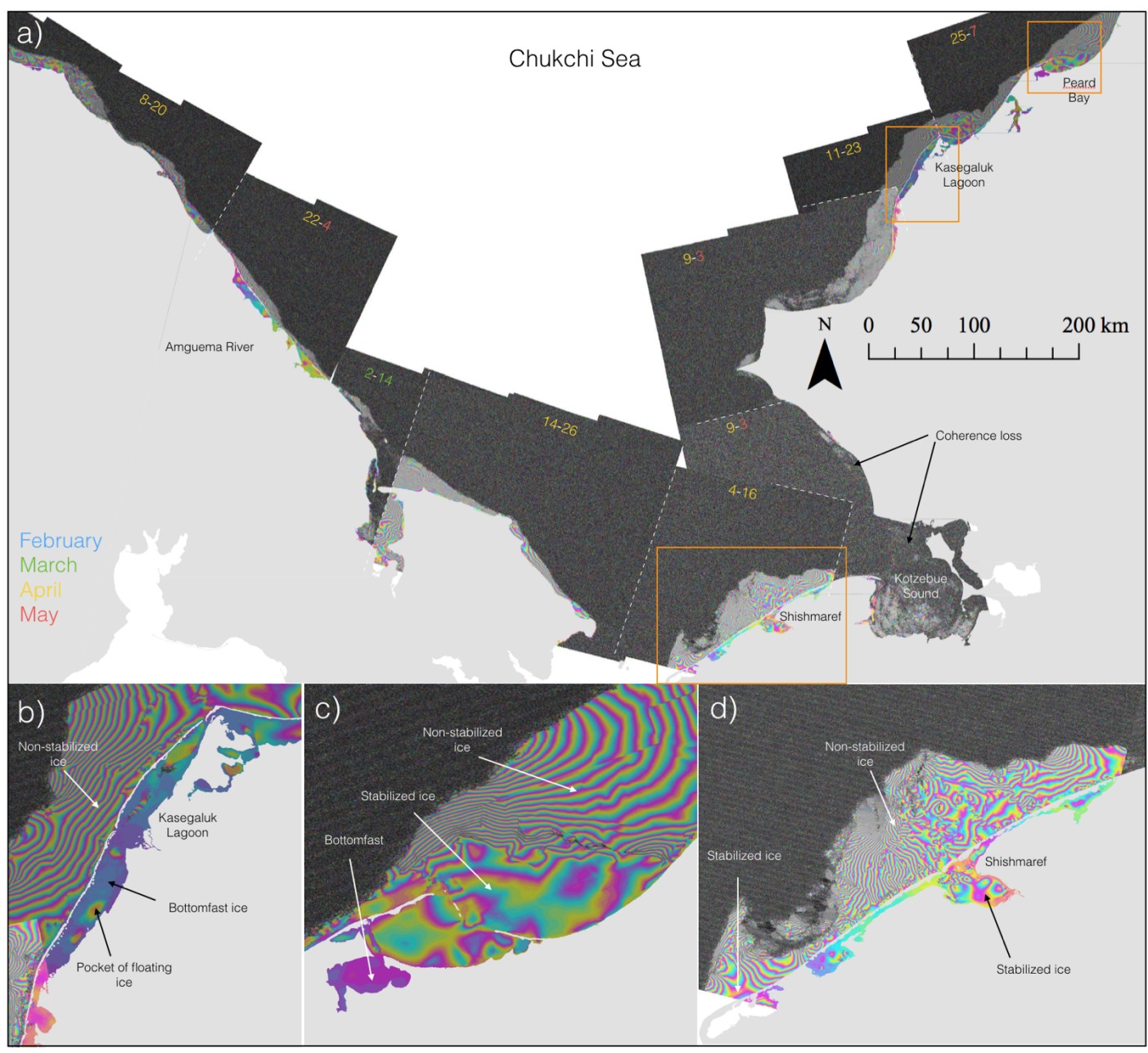

**Figure 5: (a) Sentinel-1 interferograms derived from image pairs acquired over the Chukchi Sea between March and May, 2017. Numbers on images represent date ranges where the colors blue, green, yellow, and red signify the months of February-May respectively. (b)-(d) represent three enlarged areas identified in (a) further discussed in the text.**

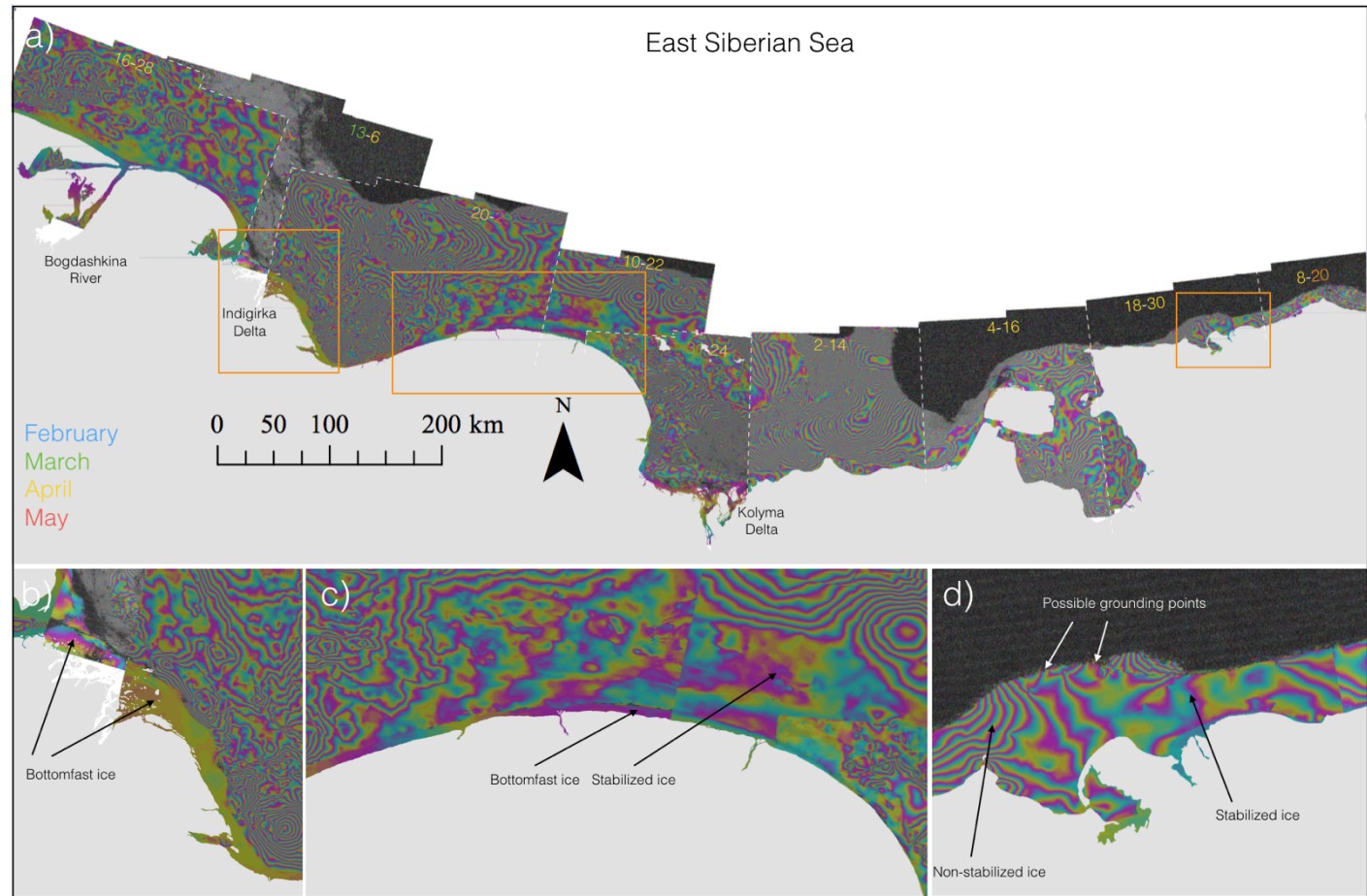

**Figure 6: (a) Sentinel-1 interferograms derived from image pairs acquired over the East Siberian Sea between March and May, 2017. Numbers on images represent date ranges where the colors blue, green, yellow, and red signify the months of February-May respectively. (b)-(d) represent three enlarged areas identified in (a) further discussed in the text.**

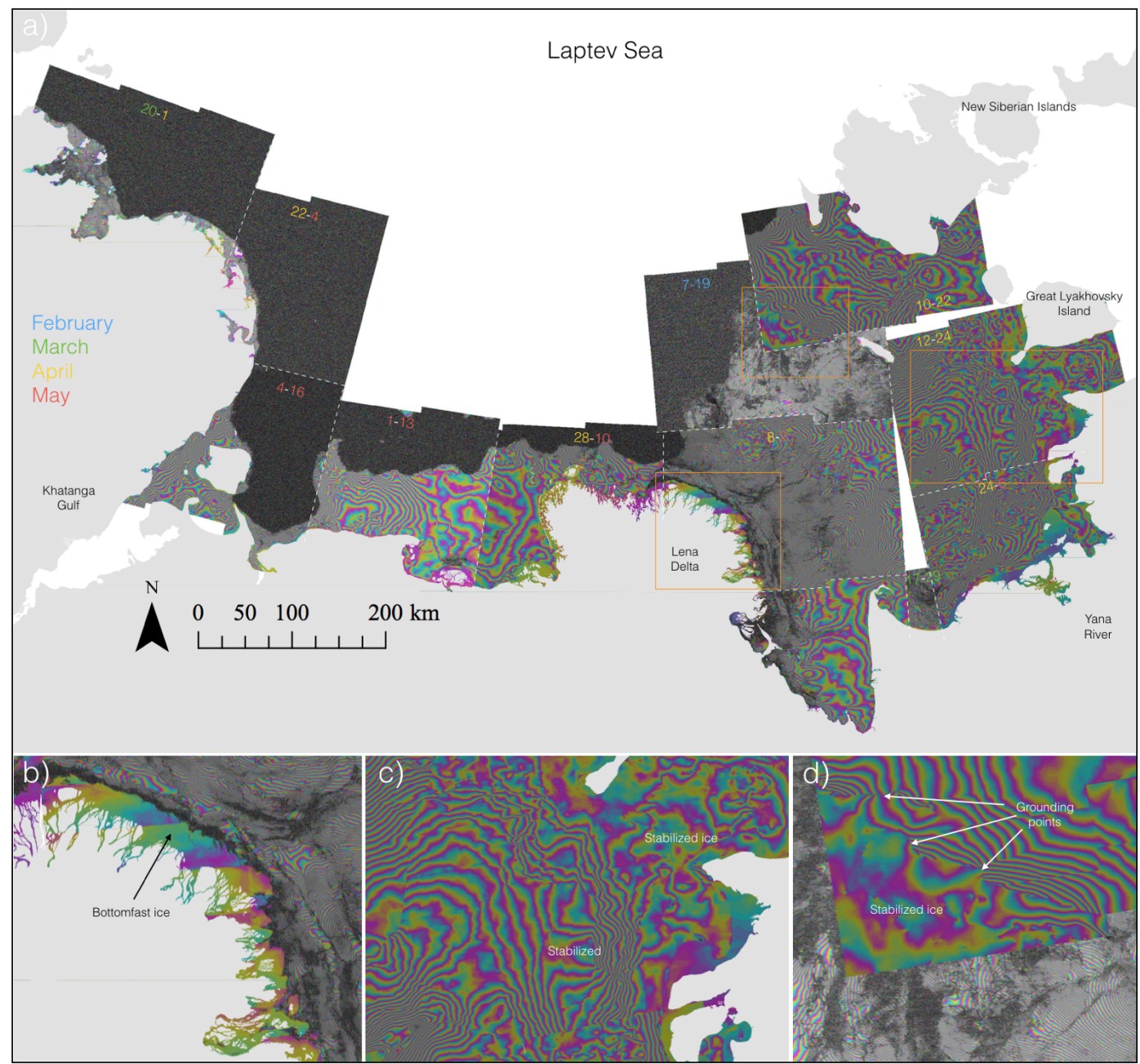

**Figure 7: (a) Sentinel-1 interferograms derived from image pairs acquired over the Laptev Sea between February and May, 2017. Numbers on images represent date ranges where the colors blue, green, yellow, and red signify the months of February-May respectively. (b)-(d) represent three enlarged areas identified in (a) further discussed in the text.**

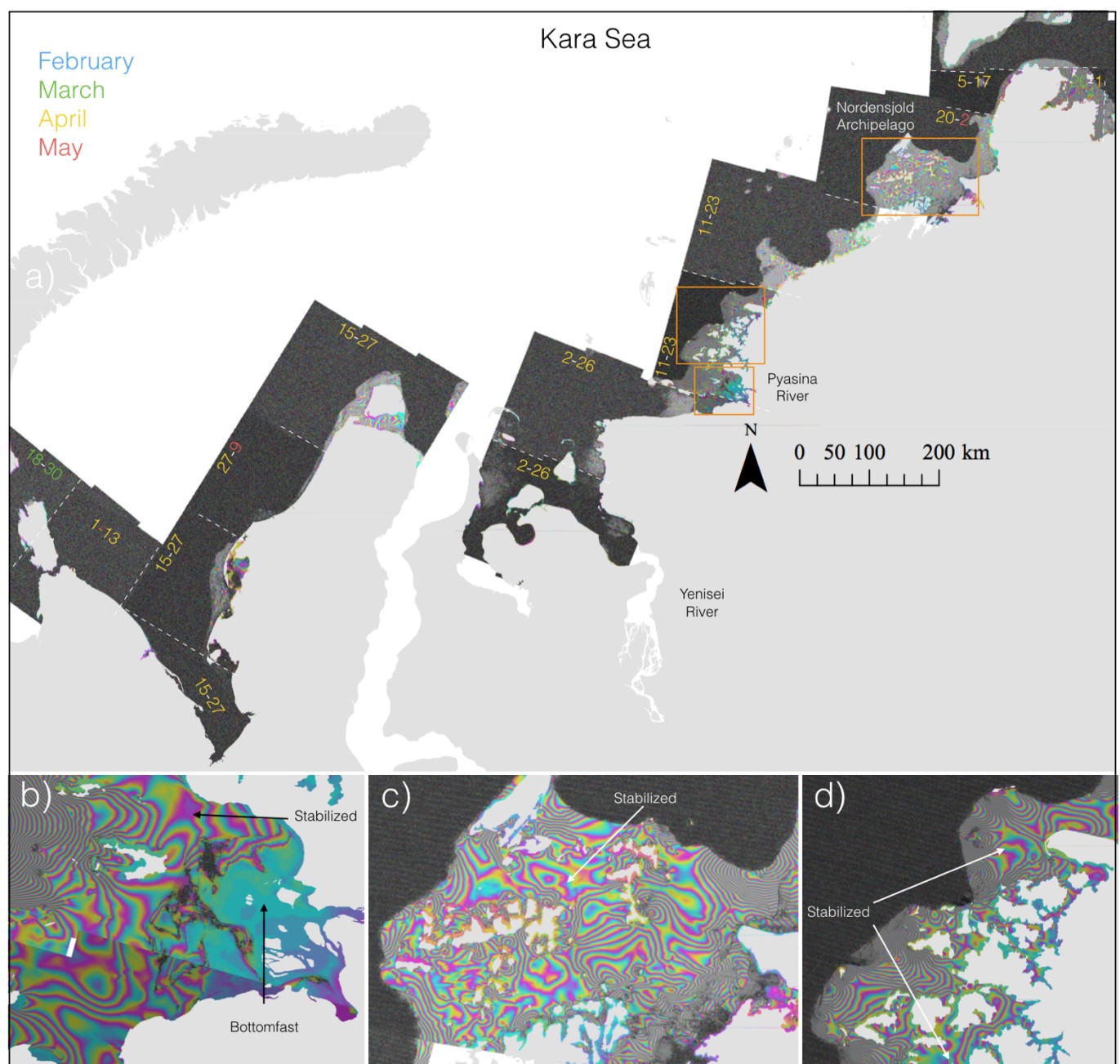

**Figure 8: (a) Sentinel-1 interferograms derived from image pairs acquired over the Kara Sea between March and May, 2017. Numbers on images represent date ranges where the colors blue, green, yellow, and red signify the months of February-May respectively. (b)-(d) represent three enlarged areas identified in (a) further discussed in the text.**

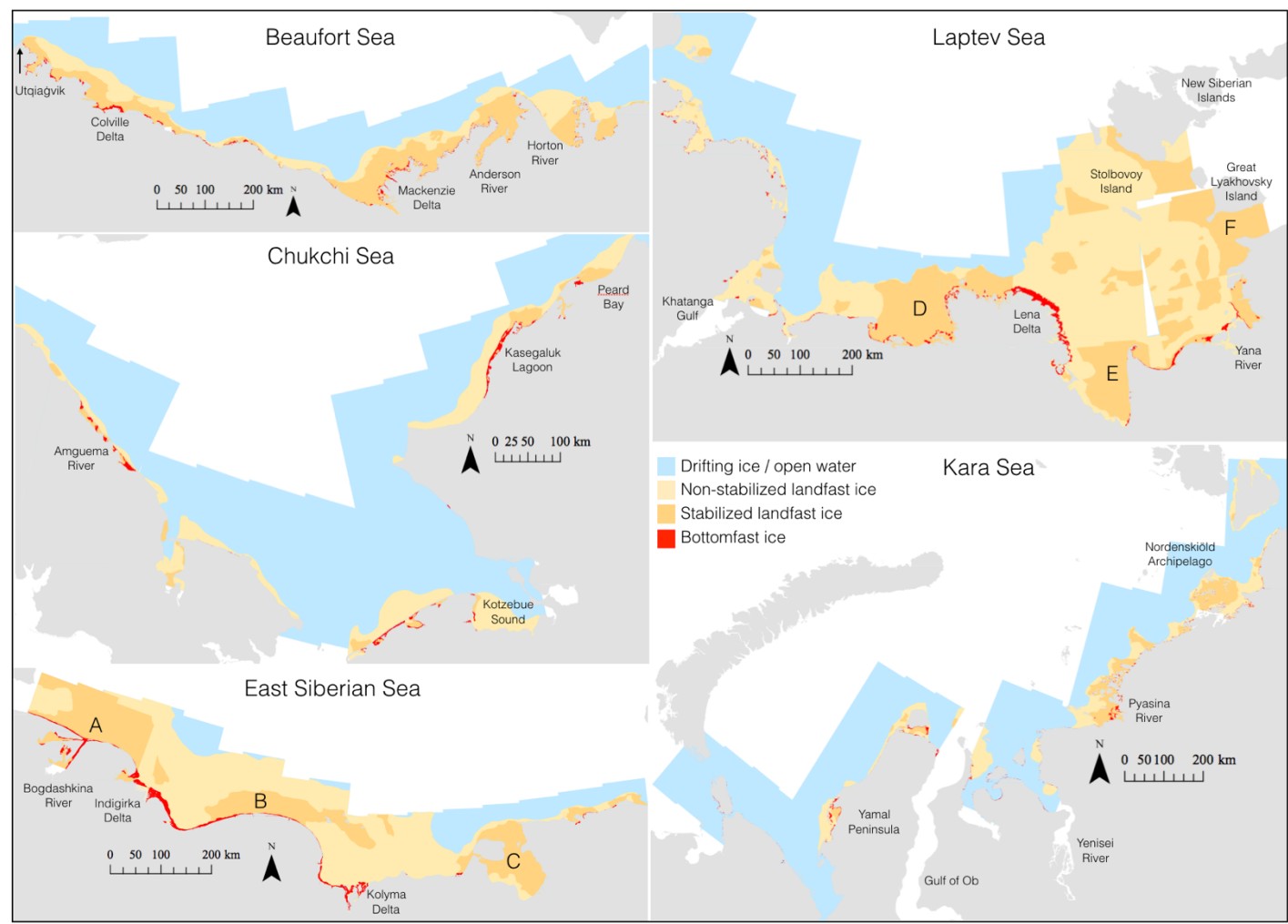

**Figure 9: InSAR-derived map of non-stabilized and stabilized landfast ice and bottomfast ice from Sentinel-1 image pairs acquired predominately between March and May, 2017. Letters "A"-"G" mark areas discussed in the text. Land is masked out in light grey. This map of stability zones is subject to limitations and uncertainties outlined in the text.**

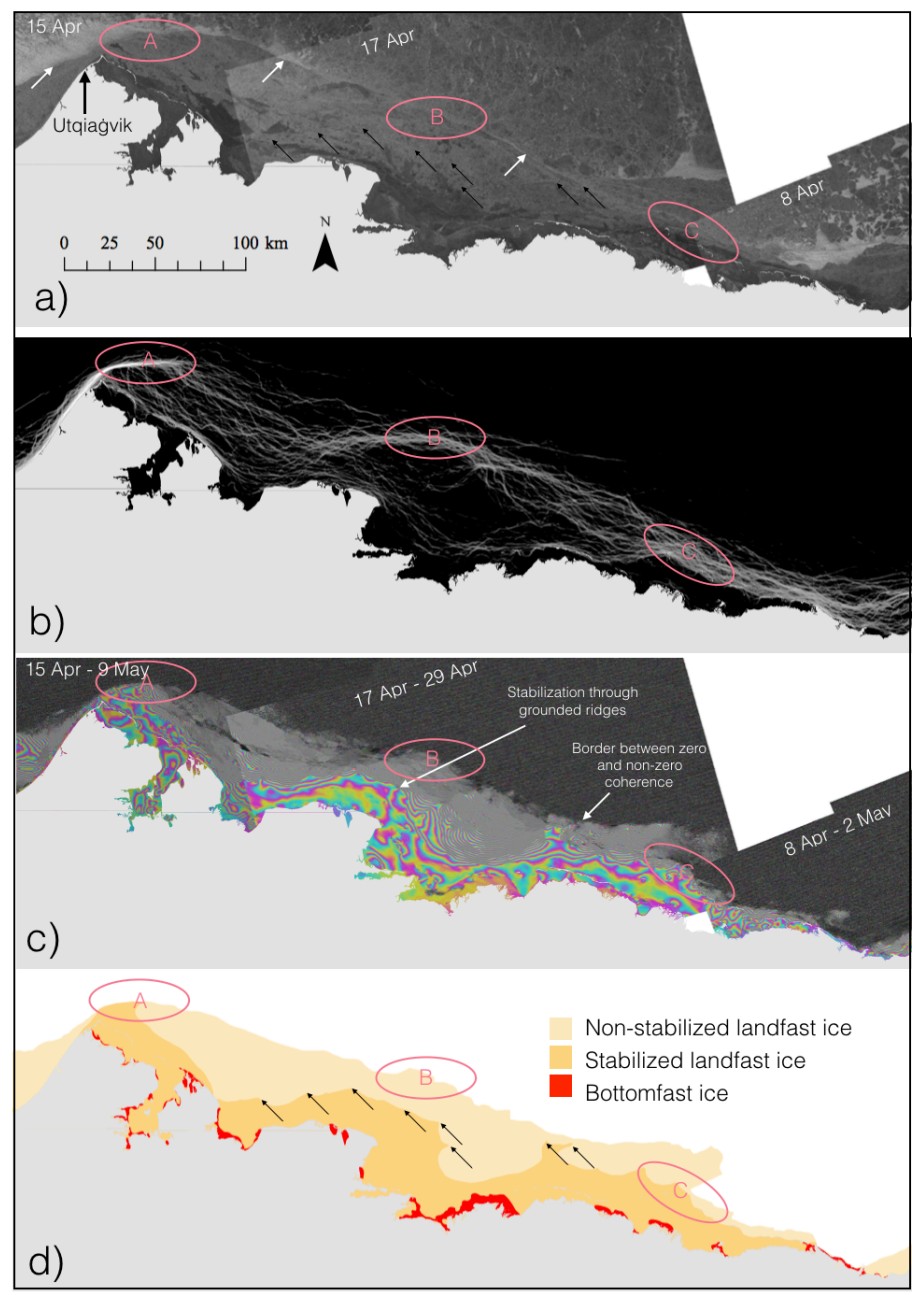

**Figure 10: Sentinel-1 backscatter images over the western Beaufort Sea. White arrows signify the landfast ice edge as identified by contrasting backscatter. (b) landfast ice edge occurrence mapped for the period 1996–2008 over the Alaska Beaufort Sea (Mahoney et al. 2014). Light red circles correspond to areas of frequent landfast ice edge formation referred to as "nodes. (c) interferograms between mid April and mid May 2017. (d) different stability zones derived from (c). Potential grounding points as identified in (d) are marked with black arrows in (d) and (a). Land is masked out in light gray.**

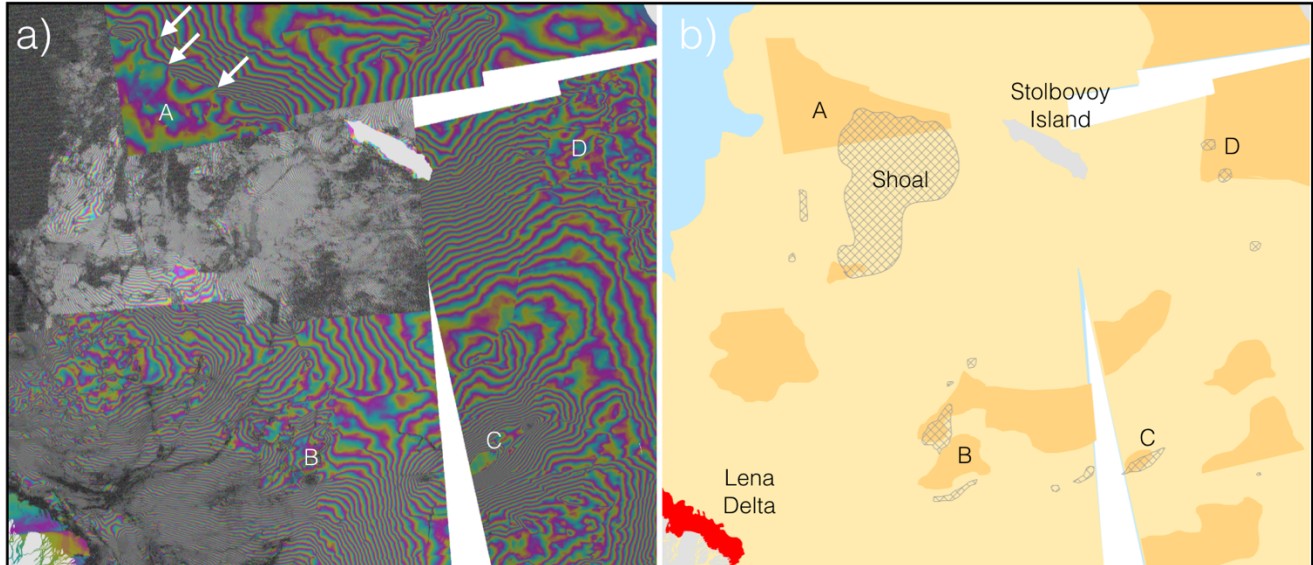

**Figure 11: (a) Sentinel-1 interferograms over Laptev Sea near Stolbovoy Island between February and May 2017. (b) outlined non-stabilized (light orange) and stabilized (dark orange) ice. Shallow areas (< 10 m) (Jakobsson et al., 2012) are marked with gray cross hatching. Stabilized ice that is likely supported by grounding near shallow features are marked "A"-"D" and further discussed in the text. Land is masked out in light gray.**

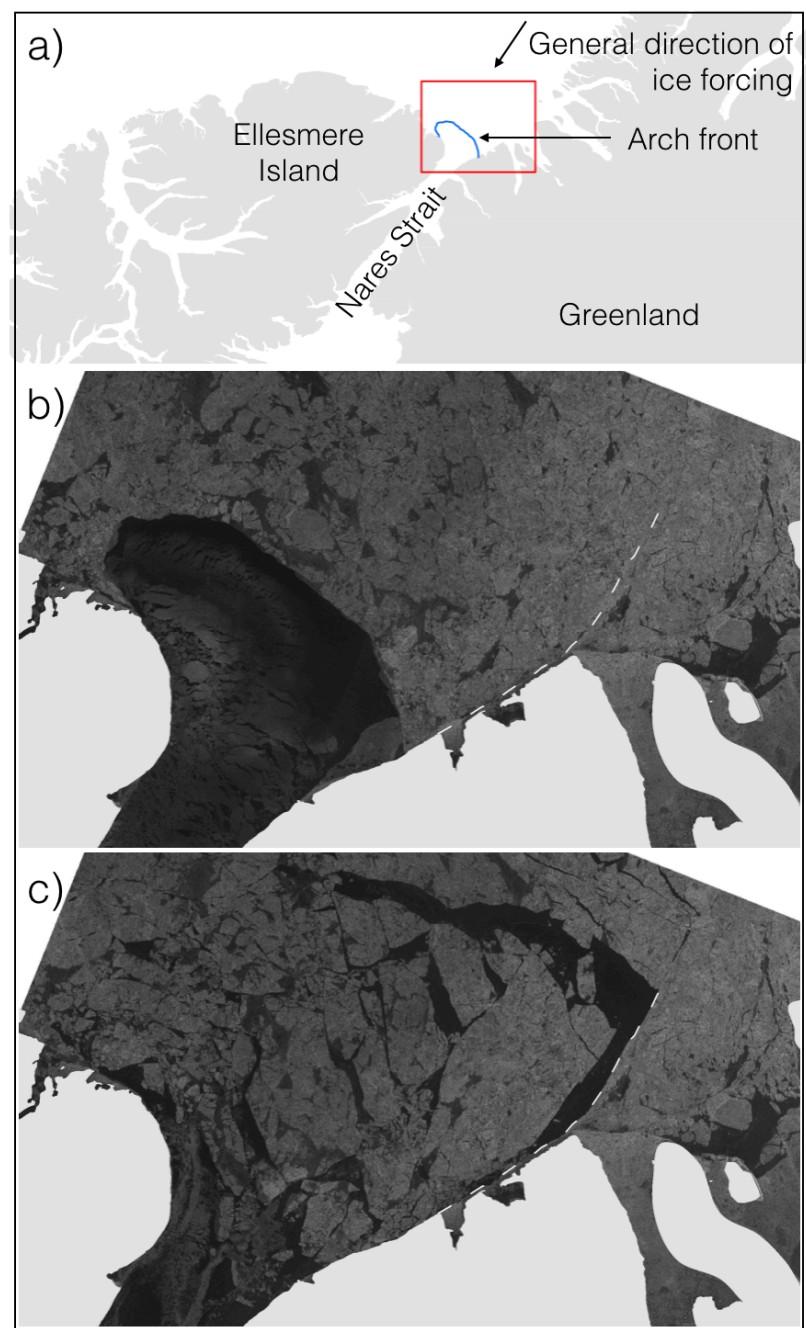

**Figure 12: Map of Nares Strait (a), and Sentinel-1 backscatter images over the 2017 ice arch (blue line in a) before (b) and after (c) failure. The line of failure is identified in (c) and marked as a dashed line in both (b) and (c). Land is masked out in light gray.**

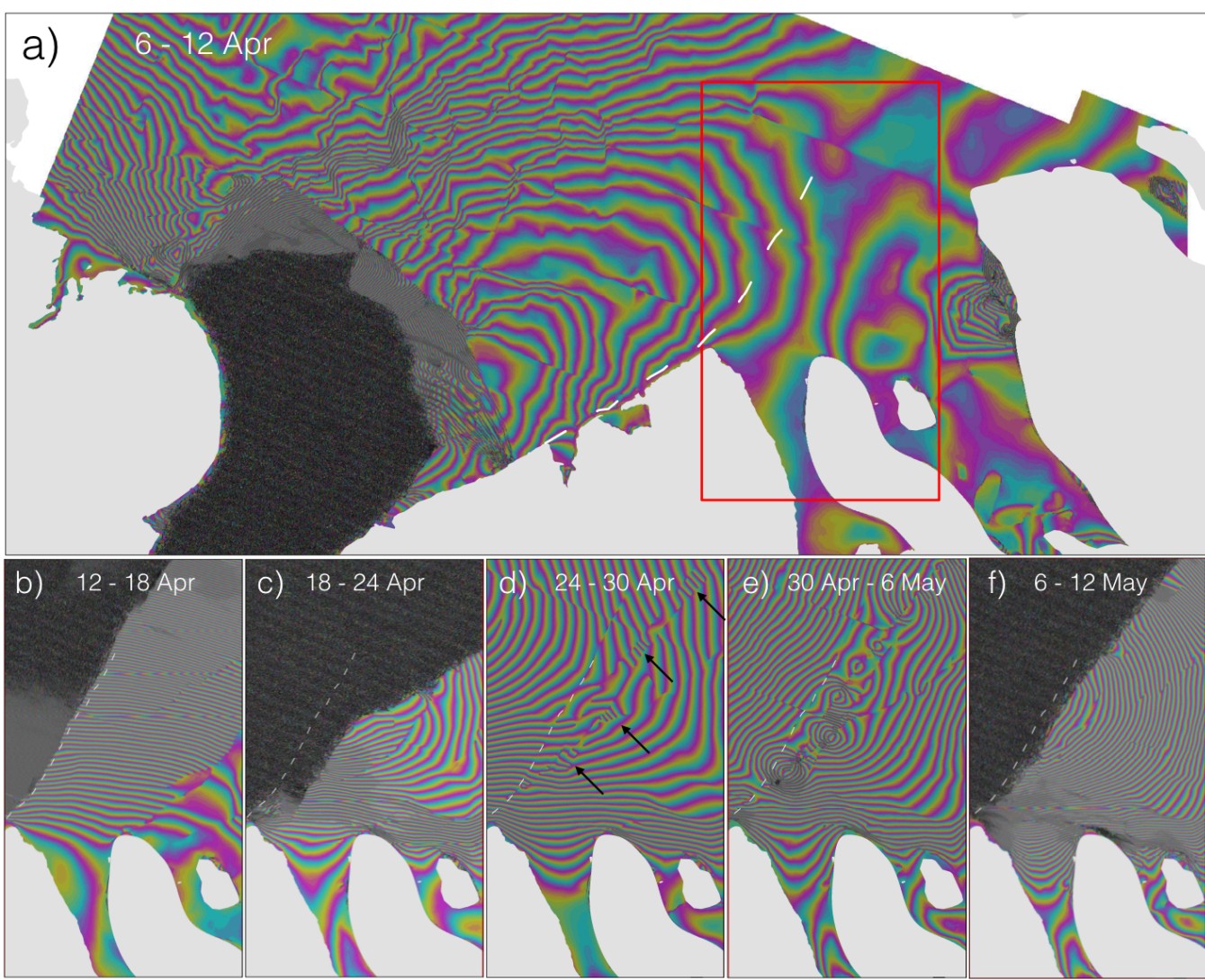

**Figure 13: Interferogram over the Nares Strait ice arch in 2017 covering the time period 6 - 12 Apr. (a). Smaller panels show consecutive interferograms within the box for 12 - 18 Apr (b), 18 - 24 Apr (c), 24 - 30 Apr (d), 30 Apr - 6 May (e), and 6 - 12 May (f). The dashed line represents the line separating the fast and moving ice in Figure 12c. The black arrows in (d) indicate fringe patterns further discussed in the text. Land is masked out in light gray.**

**Table 1:** Landfast sea ice stability regimes and assigned stability zones identified using InSAR

| | Landfast ice regime | Stability | Stability zone | Identified by |
|---|---|---|---|---|
| 1 | Bottomfast sea ice (i.e., ice frozen to or in broad contact with the sea floor) | Completely stable. Ice is frozen to or resting on the sea floor restricting lateral motion. Vertical tide jacking may occur as the ice thickens. | Bottomfast | No identifiable phase difference from the adjacent land |
| 2 | Floating ice sheltered by point features such as grounded ridges or islands or fully enclosed in lagoons or fjords | Moderate stability. Ice is supported by coastlines or point features completely or largely inhibiting break out events. In addition to thermal creep, internal stress from more dynamic ice can propagate in between pinning points resulting in dm- to m-scale non-elastic deformation. | Stabilized | Poorly defined, widely spaced fringes or abruptly reduced fringe spacing compared to offshore ice |
| 3 | Floating ice extensions | Low stability. Dominated by m-scale deformation from ice, wind, and ocean forcing. Persistent inelastic deformation can lead to accumulated strain on the order of tens of meters on time-scales exceeding several weeks. The ice may remain attached (Mahoney et al., 2004) or can break-off from the stabilized ice. | Non-stabilized | Well defined fringe orientation or patterns |

**Table 2:** Approximate area coverage of landfast ice (in thousand km$^2$).

| Area | Bottomfast | Stabilized | Non-stabilized | Total area of landfast ice | Area fraction: non-stabilized / stabilized |
|---|---|---|---|---|---|
| Beaufort Sea | 2.5 | 35 | 29 | 67 | 0.83 |
| Chukchi Sea | 1.8 | 4.6 | 25 | 31 | 5.43 |
| East Siberian Sea | 5.1 | 45 | 80 | 130 | 1.78 |
| Laptev Sea | 4.1 | 74 | 127 | 205 | 1.72 |
| Kara Sea | 2.6 | 16 | 37 | 56 | 2.3 |

The bottomfast ice zone is constrained between its outer extent interpreted from the phase and the coast as interpreted from the backscatter scenes. The stabilized zone is constrained between its outer extent as interpreted from the phase and the bottomfast ice or the landmask (Wessel and Smith, 1996). The non-stabilized ice is constrained between the outer extent of non-zero coherence and the bottomfast ice, stabilized ice, or the landmask.