# Peer review of "Mapping Arctic landfast sea ice stability with Sentinel-1 interferometry"

_The Cryosphere, 2018_

## Referee Comment (RC1) · Anonymous Referee #1 · 14 Aug 2018

Dammann and others present a nice study demonstrating a novel utility of SAR Interferometry (InSAR) for mapping landfast sea ice. They show it is possible to classify several different types of landfast sea ice as well as demonstrate potential predictability for ice arch collapse using Nares Strait as an example. This paper is well-written and the figures clearly fit the text for which they were created. I do however have a couple suggestions that I think need to be addressed prior to publication.

General Comments

1. It is possible to classify landfast ice directly from calibrated SAR imagery. In this respect, it would be useful to illustrate to the reader how much additional information

the interferogram provides. A good way to do this would be show a SAR image in Figure 3 and point out how identifying the individual landfast classes without the use of an interferogram would be difficult.

2. Discussion on ice arch collapse embodies more literature than cited in the manuscript. The Barber et al. (2018) reference is good and very appropriate but investigations into this process have been ongoing for many years prior. Melling (2002) originally suggested it with respect to the Northwest Passage in the Canadian Arctic under a warming climate. This process was then quantified by Kwok (2006) and more recently by Howell et al. (2013) using Radarsat imagery, with the latter study showing evidence for multi-year ice increases in recent years. Haas and Howell (2015) further provided ice thickness measurements for a gradient from the Arctic Ocean to the southward shipping channels of the Northwest Passage. Overall, the relevance of the technique presented in the manuscript for predicting ice arch collapse from InSAR is important but some additional citing literature (as outlined above) providing a more comprehensive discussion is required.

Specific Comments

Page 3, Line 28

Specify the dates and beam mode for the S1 images. Not the entire list of images just more details on the beam modes used on the analysis.

Page 4, Line 35

I would think (and argue) that landfast regions in the Canadian Archipelago are as just as important for shipping. They would also be useful a region to test the ice collapse prediction process similar to Nares Strait. See General Comment 2.

Page 8, Line 12 and Lines 33-34

The process of ice arch breaking and thick multi-year ice from the Arctic Ocean being transported southward and into shipping lanes extends beyond the work cited in the

manuscript. See General Comment 2.

Page 8, Lines 14-28

A recent study by Moore and McNeil (2018, GRL) documents the 2017 ice arch collapse in Nares Strait. It might be worthwhile verifying if the interferograms are in agreement with what is documented by Moore and McNeil.

References

Haas, C., and S. E. L. Howell (2015), Ice thickness in the Northwest Passage, Geophys. Res. Lett., 42, 7673–7680, doi:10.1002/2015GL065704.

Howell, S. E. L., T. Wohlleben, M. Dabboor, C. Derksen, A. Komarov, and L. Pizzolato (2013), Recent changes in the exchange of sea ice between the Arctic Ocean and the Canadian Arctic Archipelago, J. Geophys. Res. Oceans, 118, 3595–3607, doi:10.1002/jgrc.20265.

Kwok, R. (2006), Exchange of sea ice between the Arctic Ocean and the Canadian Arctic Archipelago, Geophys. Res. Lett., 33, L16501, doi:10.1029/2006GL027094.

Melling, H. (2002), Sea ice of the northern Canadian Arctic Archipelago, J. Geophys. Res., 107(C11), 3181, doi:10.1029/2001JC001102

Moore, G.W.K and K. McNeil (2018), The early collapse of the 2017 Lincoln Sea ice arch in response to anomalous sea ice and wind forcing, Geophys. Res. Lett, https://doi.org/10.1029/2018GL078428

---

## Referee Comment (RC2) · Anonymous Referee #2 · 21 Aug 2018

**General comment**

This study analysed the potential of Sentinel-1 interferometry to distinguish three classes of landfast ice stability at the pan-Arctic scale. The method uses the fringe density pattern in wrapped interferograms. The delineation of the classes is performed then manually. The approach is clear and definitely shows a potential of the Sentinel-1 InSAR for the operational monitoring of the landfast ice at the pan-Arctic scale. However, in my opinion, the manual delineation of the classes is a serious weakness of this study. My feeling is that the manual separation of the fringe patterns was not performed accurately enough to go on with a quantitative and perhaps even qualitative analysis of the classes' area. By looking at the interferograms I often could not see a difference between fringe patterns in the areas, which you separated to different classes. Maybe in some areas this is due to a lack of details in Figure 4, and if this is the case, you should provide detailed examples. The following examples illustrate my concerns.

[Figure]

[Figure]

Laptev Sea

How did you distinguish these 2 classes?

20 Mar - 1 Apr

22 Apr - 4 May

12 Apr - 24 Apr

1 May - 16 May

1 May - 13 May

28 Apr - 10 May

8 Apr - 2 May

24 Apr - 6 May

Khatanga Gulf

Lena Delta

17 Mar - 29 Mar

2 May - 14 May

N

0   50  100      200 km

Why these areas were not included to the stable ice class?

[Figure]

East Siberia

A

Bogdashkina River

Indigirka Delta

B

N

How did you identify these bottomfast ice areas?

What is the difference to the pattern here, for example?

[Figure]

I have the feeling that the delineation is too subjective and that the resulting classes could look very different if analysed by other operator. Therefore, the areas, provided in Table 3 can substantially differ depending on the delineation criteria. Also, a number of gaps are present in the data (Khatanga Gulf, Laptev Sea to the south from New Siberian Islands, eastern and central part of East Siberian Sea, Gulf of Ob), which makes the areal analysis incomplete. Are there Sentinel-1 data available? Could they also be included to the analysis? Moreover, I think that bathymetry should be used together with interferograms to make an adequate delineation. IBCAO data can be used to identify potential bottomfast ice areas and also can clearly exclude areas, erroneously mapped as bottomfast ice if water depth exceeds a certain threshold (for instance, 2 m, see the figure below).

[Figure]

As the Referee 1 suggests, I also think you could use the backscatter images to support the delineation of the bottomfast ice, and potentially other classes (or to show that the backscatter has no / limited use for this purpose).

Furthermore, a big improvement could be an unwrapping of interferograms w.r.t. the land area, which should be stable during the winter. By doing so, you could derive the magnitude of the ice movement, and then the ice stability classes could be distinguished based on the magnitude. In this case, the subjective delineation could be substituted by an automated classification. Moreover, the vertical and horizontal movements could potentially also be distinguished using ascending and descending orbits. This is probably beyond the study intention, but definitely should be discussed. You suggest that the used approach is rather simple and does not require SAR expertise. However, you do need to have some expertise to decide on the phase change gradient, and this seems to be a major problem at the moment. The unwrapping step is included into every standard interferometric workflow, and therefore, could be tried, at least exemplarily. This, perhaps, could also improve the identification of the bottomfast ice, as the InSAR phase in this case should be the same as over the land.

The authors provide a validation of their result for the Beaufort Sea, by comparing stability zones with long-term frequent position of the landfast ice edge from Mahoney et al. (2014). As stated in the Results chapter, they associate the second discontinuity in the InSAR phase with the nodes, identified in Mahoney et al. (2014). This discontinuity separates the stable/non-stable zones, whereas the nodes coincide with the edge of the landfast ice. Therefore, it appears to me that the comparison with the nodes from Mahoney et al. (2014) can only validate the overall extent of the fast ice, without being able to validate stable/non-stable classes identification. Perhaps, it can be partly attributed to the interannual variability of the landfast ice edge position (and position of the border between stable and non-stable fast ice). Nevertheless, I find such comparison important, and strongly suggest to add comparisons for other areas, for instance, for the Laptev Sea, where the positions of the fast ice edge are presented in Selyuzhenok et al. (2015). For that, the full extent of the fast ice in the Laptev Sea would be required, while at the moment, the important ice grounding zones are excluded from the analysis (see Figure in my general comment).

Some concerns about a proper terminology: can you use simply "stable" and "non-stable" (or unstable") ice classes, instead of stabilised / non-stabilised? Or do you want to emphasize that the stabilised class can be unstable before grounding? Also, the three stability classes are called "regimes", "zones", or "classes" throughout the paper. I suggest using one term consistently, and for me personally, the term "regime" does not seem to be correct, as you are focusing on a snapshot and not time series. Also, I'm not sure but I think the term "delineation" imply a linear feature, so when you refer to an areal object, maybe it is better to use "mapping"?

I also recommend the language proofreading to simplify and smooth the style and improve the clarity. Some sentences are very long, using redundant words, too many subordinate clauses. A tip from Elsevier, for instance: "Nowadays, the average length of sentences in scientific writing is about 12 to 17 words". There also are a few typos.

**Specific comments**

p.2 lines 5-8: sentence is too long, and the second part of it is not clear ("defined here as the immobility" – defined where?)

p.2 lines 8-11: could you split the sentence in two?

p.2 line 13: "from left to right" – better use "from coast towards open ocean" as you should not describe a figure in the main text.

p.2 line 14: can grow laterally

p.2 line 19: "high to moderately stable landfast ice…"

p.2 line 21-27: I suggest to add more structure to this part. Do you want to say that low-stability ice is simply dangerous for ocean-based operations? How are grounded ridges related to the low-stability ice? Do ice arches belong to the stable class?

p.2 line 28: No need to specify "sea", just landfast ice

p.2 line 29: I believe you do not need to mention in situ data, it is absolutely clear

p.2 line 33-36: consider to compact the sentence

p.2 line 37: what kind of types? Age?

p.2 line 38: because the movements are too small to use change detection?

p.3 line 1: between two SAR images

p.3 line 2: either specify that this is the case of the sea ice, or leave it more general for any kind of displacement/topography analysis

p.3 line 11: "These studies have demonstrated…"

p. 3 line 13: "as a mean…"

p.3 line 15-17: consider to split and compact the sentence.

p.3 line 17: you do not need to determine the ability to create interferogram. It can be created for any interferometric pair. Reformulate or simply delete this.

p.3 line 17-22: reformulate and simplify the sentences and do not refer to the Table and Sections, it is distracting.

p.3 line 27: in slant or ground range? This sentence should go after you specify that you use IW mode

p.3, line 33: "a sample.."?

p.3 line 36-37: this is a repetition from the previous chapter

p. 3 line 37: "the InSAR phase may be related…"

p.4 line 3: can you specify the limits? I believe that the critical topography height can be calculated from the maximum baseline.

p. 4 line 5: what means "only" here? You should also mention here atmospheric phase delay and other contributors to the InSAR phase.

p. 4 line 7-8: "wrap around" – find more rigorous phrasing

p. 4 line 8: "The results is…" – "The interferogram is…"

p. 4 line 12: again, any interferogram can be successfully created, please reformulate

p. 4 line 13: ranging between 0 and 1?

p. 4 line 14: "A complete list.." – this sentence does not flow logically from the previous one.  Either remove it or provide a logical connection.

p. 4 line 19-21: After such a long technical introduction to the InSAR, the processing flow in two lines is way too short. Please provide more details.

p. 4 line 23: in my view, "determining relative strain rates", even roughly, is a little bit of exaggeration of what you really do. Just say that you use fringe density as an indicator of stability.

p. 4 line 24-25: I don't see how this sentence helps in understanding the following one. Consider removing.

p. 4 line 25: "abrupt changes"?

p. 4 line 26: "…to identify three stability zones: *list them*". Consider joining Tables 1 and 2.

p. 4 line 33: in what study?

p. 5 line 2: was coherence low for all available pairs for these regions? I see that you mention that you do not "attempt to derive alternative interferograms in these cases" but this data gap can probably be easily filled, even in potentially operational mode.

p. 5 line 7: I suggest not to start your Results with a description of results from the other study, even though relevant here. I suggest to move the chapter to the end of Results or even to Discussion.

p. 5 line 10-11: consider simplify the sentence; you do not need here the technical explanation how these edges were derived.

p. 5 line 14-15: do not describe the figure in the main text.

p. 5 line 21: message in parentheses is confusing. I think you can see a pattern in these areas as well. Depends on what you call a pattern.

p. 5 line 22: "This discontinuity has been shown…". I do not think it can be called a gradient.

p. 5 line 24: see my general comment

p. 5 line 25: "reoccurring"

p. 5 line 27: what is meant by reduced phase response? Also, the sentence is too long.

p. 5 line 27-29: please reformulate and compact the sentence, it is hard to follow. What are the points of higher stability and how the points can be seaward?

p. 5 line 32: Mapping pan-Arctic landfast ice stability zones?

p. 5 line 34: please do not refer to the Sections

p. 5 line 35: see my general comment. Without a convincing explanation how the bottomfast ice was distinguished from the non-bottomfast, I think that the current mapping is not accurate enough.

p. 5 line 36, please do not refer to the Sections

p. 6 line 1-2: what is meant by substantial bottomfast ice?

p. 6 line 3: what means Accordingly here?

p. 6 line 7-8 please do not refer to the figures and Sections here, it should be clear without that.

p. 6 line 12: revise the sentence, something is missing

p. 6 line 14: "…landfast ice…"

p. 6 line 15-18: please shorten and clarify the sentence

p. 6 line 21-22: identified where?

p. 6 line 22-23: as in my general comment examples, I do not see how this area is different from, for instance, the area around Bely Island (right above the Yamal peninsula).

p. 6 line 24-25: I think in this case you simply should not mention it here at all.

p. 6 line 27: "…limitations for mapping stability classes"?

p. 6 line 29: typically or in your study?

p. 6, line 36-p.7 line 5: this repeats the paragraph in the Result section. I suggest to remove it from the Results as it is a discussion point with speculations.

p. 7 line 14: how the definition of the "highly stable" from Eicken et al. (2005) is related to the definition of stable in your study? Can they be compared?

p. 7 line 15: or this part is also prone to the tidal movements?

p. 7 line 17: SAR backscatter analysis?

p. 7 line 18-19: but the mobile ice would be incoherent in your results? Also "one month after the initial freeze up" – while you consider late winter situation. As recommended in the general comment, please include more scenes for the Laptev Sea to have an entire fast ice extent – maybe it can shed more light on the situation here.

 p. 7 line 27: why to mention it here if it is discussed below?

p. 7 line 28-30: or it can aid in the interpretation, e.g. by confirming the pattern, or by featuring the temporal changes between interferograms. Also, I do not see what is meant to be in the northernmost region of the Lena Delta in Figure 4.

p. 7 line 33: is this connected to the previous sentence? If not please restructure

p. 7 line 35: so they **are not** available in late winter 2017 or **may not** be available?

p. 7 line 36: as you say "consistent coherence loss", now I probably got an answer to my question to p. 5 line 2. Can it simply mean that there is no landfast ice?

p. 7 line 36-37: temporal progression or spatial? Not clear what is meant here.

p. 7 line 38: why would we be interested in using this analysis for the melt onset period, when the ice starts to degrade and be unstable anyway? Please develop you thought here.

p. 8 line 13: "…multiyear ice…"

p. 8 line 14: In our study?

p. 8 line 15: Sentinel-1 backscatter imagery? Did it capture the formation?

p. 8 line 16-17: this goes to the figure caption

p. 8 line 36: potential applications?

p. 9 line 23-26: please split the sentence

p. 9 line 27: "multiple" → three?

p. 9 line 28-31: aren't these classes used for all seas? Why to specify Beaufort Sea here?

p. 10 line 3-8: this is too detailed for the Conclusions, and mainly repeats the paragraph from the Results. Please generalise your findings in Conclusions.

p. 10 line 19: did you actually consider year-to-year timescale?

p. 14 line 4: remove the word "ice" before "sea ice"

p. 15 Figure 2 caption: Conceptual scheme?

p. 16 Figure 3: place the red ovals a,b,c on top of the b) panel; caption: explain a,b,c nodes in a).

p. 17 Figure 4. I suggest to redistribute the panels in such a way that all of them can be enlarged.

p. 18, Figure 5: enlarge, the same as figure 4. The river you refer as to Angara, is Yenisei! Please check other geographical names and add river shapes on the maps.

p. 19, Figure 6: please add the overview figure showing the location of the arch. Use additional graphic to delineate the arch and the flow direction. Some readers are unfamiliar with this sea ice feature.

---

## Author Comment (AC1) · 27 Sep 2018

Reviewer 1

Dear reviewer,

We greatly appreciate your insightful comments. We have followed all your recommendations, which we acknowledge has led to an improved manuscript. Please see detailed responses below.

With best regards, Dyre Oliver Dammann

Dammann and others present a nice study demonstrating a novel utility of SAR Interferometry (InSAR) for mapping landfast sea ice. They show it is possible to classify several different types of landfast sea ice as well as demonstrate potential predictability for ice arch collapse using Nares Strait as an example. This paper is well-written and the figures clearly fit the text for which they were created. I do however have a couple suggestions that I think need to be addressed prior to publication.

General Comments

1. It is possible to classify landfast ice directly from calibrated SAR imagery. In this respect, it would be useful to illustrate to the reader how much additional information C1 the interferogram provides. A good way to do this would be show a SAR image in Figure 3 and point out how identifying the individual landfast classes without the use of an interferogram would be difficult.

Great point. We have followed your suggestion and included the backscatter now as Figure 5b. We have also commented on this by including:

P6 L26: "The respective master images exhibit a sharp discontinuity in backscatter (see arrows in Figure 5b) along the general location of the landfast ice edge from Figure 5a and can be assumed to be the landfast ice edge. Determining the landfast ice edge can in some instances be achieved by evaluating a single amplitude image as here, but is not consistent and only works in cases where there are stark discontinuities in backscatter due to different ice types or a severely deformed landfast ice edge. The interferograms indicate in this case a similar landfast ice edge by a complete loss of coherence seaward of the discontinuity apparent in Figure 5b (Figure 5c)."

And later:

P7 L7: "Although the landfast ice edge can in some instances be delineated using a single backscatter image, the delineation of bottomfast and stabilized ice cannot be discriminated as is apparent by comparing the delineations from Figure 5e with Figure 5b."
2. Discussion on ice arch collapse embodies more literature than cited in the manuscript. The Barber et al. (2018) reference is good and very appropriate but investigations into this process have been ongoing for many years prior. Melling (2002) originally suggested it with respect to the Northwest Passage in the Canadian Arctic under a warming climate. This process was then quantified by Kwok (2006) and more recently by Howell et al. (2013) using Radarsat imagery, with the latter study showing evidence for multi-year ice increases in recent years. Haas and Howell (2015) further provided ice thickness measurements for a gradient from the Arctic Ocean to the southward shipping channels of the Northwest Passage. Overall, the relevance of the technique presented in the manuscript for predicting ice arch collapse from InSAR is important but some additional citing literature (as outlined above) providing a more comprehensive discussion is required.

Thank you for providing reference to great literature, which we acknowledge has contributed to strengthen the discussion in Section 4.2. All your suggested literature has been included.

Specific Comments

Page 3, Line 28 Specify the dates and beam mode for the S1 images. Not the entire list of images just more details on the beam modes used on the analysis.

We have restructured the section and state this more clearly now. We also now include a mentioning of the full list in the supplementary data:

P3L30: "All images used were captured in interferometric wideswath (IW) mode ($\sim$250 km swath width) during the months of March through May 2017 (see supplementary data for full list of images used)."

Page 4, Line 35 I would think (and argue) that landfast regions in the Canadian Archipelago are as just as important for shipping. They would also be useful a region to test the ice collapse prediction process similar to Nares Strait. See General

Comment 2.

We agree with this and rephrased the sentence, which we acknowledge could be read like Canada was less significant in this respect. It now reads:

P5L15: "The Alaskan and Russian coastlines have high economic significance for the shipping and natural resource industries and feature diverse ice stability regimes and large areas of bottomfast ice are expected in these regions."

Page 8, Line 12 and Lines 33-34 The process of ice arch breaking and thick multi-year ice from the Arctic Ocean being transported southward and into shipping lanes extends beyond the work cited in the C2 manuscript. See General Comment 2.

See comment to general comment 2.

Page 8, Lines 14-28 A recent study by Moore and McNeil (2018, GRL) documents the 2017 ice arch collapse in Nares Strait. It might be worthwhile verifying if the interferograms are in agreement with what is documented by Moore and McNeil.

Great suggestion. Yes, the findings seem to match quite well. A comment has also been added:

P9L15: "This failure event occurred relatively early as compared with past events (Kwok, 2005) partly in response to thinner ice conditions and northerly winds (Moore and McNeil, 2018)."

References

Haas, C., and S. E. L. Howell (2015), Ice thickness in the Northwest Passage, Geophys. Res. Lett., 42, 7673–7680, doi:10.1002/2015GL065704.

Howell, S. E. L., T. Wohlleben, M. Dabboor, C. Derksen, A. Komarov, and L. Pizzolato (2013), Recent changes in the exchange of sea ice between the Arctic Ocean and the Canadian Arctic Archipelago, J. Geophys. Res. Oceans, 118, 3595–3607, doi:10.1002/jgrc.20265.

[Figure]

Kwok, R. (2006), Exchange of sea ice between the Arctic Ocean and the Canadian Arctic Archipelago, Geophys. Res. Lett., 33, L16501, doi:10.1029/2006GL027094.

Melling, H. (2002), Sea ice of the northern Canadian Arctic Archipelago, J. Geophys. Res., 107(C11), 3181, doi:10.1029/2001JC001102

Moore, G.W.K and K. McNeil (2018), The early collapse of the 2017 Lincoln Sea ice arch in response to anomalous sea ice and wind forcing, Geophys. Res. Lett, https://doi.org/10.1029/2018GL078428

Reviewer 2

Dear reviewer, Thank you for conducting a very thorough review. This is much appreciated and have helped us to substantially improve this manuscript by following your suggestions. Please find detailed responses to each of your comments below. Thanks again! Best regards, Dyre Dammann

General comment This study analysed the potential of Sentinel-1 interferometry to distinguish three classes of landfast ice stability at the pan-Arctic scale. The method uses the fringe density pattern in wrapped interferograms. The delineation of the classes is performed then manually. The approach is clear and definitely shows a potential of the Sentinel-1 InSAR for the operational monitoring of the landfast ice at the pan-Arctic scale. However, in my opinion, the manual delineation of the classes is a serious weakness of this study. My feeling is that the manual separation of the fringe patterns was not performed accurately enough to go on with a quantitative and perhaps even qualitative analysis of the classes' area.

Please see the reply below regarding implications of the subjective nature of this analysis

By looking at the interferograms I often could not see a difference between fringe patterns in the areas, which you separated to different classes. Maybe in some areas this is due to a lack of details in Figure 4, and if this is the case, you should provide detailed

examples. The following examples illustrate my concerns.

Thank you for pointing out these concerns. We have numbered these into 11 separate concerns (See numbers in Fig 1-3), which are here each addressed in detail:

1) These sections had been separated previously when considering a different delineation approach. Due to an oversight, this area was now still classified as stabilized by a mistake. This area have been marked as non-stabilized in the new version of the figure.

2) We initially tried to constrain ourselves to strictly the coastline, to make the computational effort manageable. However, your idea is good, to include this area due to the known shoals. We have now included this area and the findings are discussed below in response to your other comments regarding this area.

3) Thank you for pointing out this oversight. This is explained in Dammann et al. (2018c), but was not mentioned here. We therefore include in Section 4.1: "However, we greatly reduced such errors by not delineating areas that appears to be low-laying land and sediment bars in the SAR backscatter images. This can lead to the appearance of sporadic areas of bottomfast ice, separated by areas with a phase resembling that of bottomfast ice, but which in reality is land. In areas where the landmask does not appear to fit the coastline, delineating the intricate coastal morphology can be a time-consuming task, hence delineation on a pan-Arctic scale will inevitably contain inaccuracies." We have also included an explanation in the Figure 5 caption: "Due to minor inaccuracies in the landmask, the occurrence of bottomfast ice can appear sporadic."

We have further cropped out an example of the analysis in the area of your concerns (Fig 4) showing the interferogram overlaid the backscatter image with the areas classified as bottomfast ice (top), the classification (middle), and a zoomed-in area (black box in the middle figure) and the delineated bottomfast ice (bottom). In the bottom figure, it is clear that there exist minor areas that are not delineated. Outlining absolutely every

little piece would not be practical at the scales addressed here, but is possible on the scales used in Dammann et al. (2018c). This is now addressed in Section 4.1.

4) Thank you for pointing this out. We agree that this area was not delineated properly according to our criteria outlined in Table 2. We have now delineated this area more carefully.

5) Please see point (1).

6) The bottom area is not delineated since there are no strong fringe gradients and there is a persistent fringe signature. Per our delineation criteria outlined in Table 1, this area has not been identified as stabilized. The contrast between low and high fringe density near the top of your circle is due to the image boundary between two interferograms acquired at different times. We realize that it is not always easy to discriminate between different interferograms. We have therefore now marked each boundary with a dashed white line to make this easier (see new version of the figure).

The easternmost area within the circle should be counted as stabilized ice and have has now been included. The westernmost area resembles bottomfast ice and is marked as a shallow area in your attached figure, but in fact appears to be largely comprised of land (see Fig 5) and is hence not marked as bottomfast.

7 and 8) This image was not properly geocoded for some reason resulting in a poor match between landmask and land in the image. This has now been adjusted and corrected for. Therefore, most of the areas identified as bottomfast ice is now apparent and not covered by the landmask. Even so, a couple of areas still exist that we identify as land in the backscatter image, but is not covered by the landmask. These areas feature zero phase change, but are still not identified as bottomfast ice. The most prominent example is the westernmost part of the northernmost bay.

Furthermore, the gaps you point out were filled out shoreward of the respective class, but we agree this is not really useful or correct. It is better to leave these areas out of

the analysis, which has now been done.

I have the feeling that the delineation is too subjective and that the resulting classes could look very different if analysed by other operator. Therefore, the areas, provided in Table 3 can substantially differ depending on the delineation criteria.

It is true that different operators could delineate differently, which occurs on a daily basis with ice charting of other properties. However, the subjectivity does not take away the value of ice charts and we argue that is similarly the case for stability.

Also, a number of gaps are present in the data (Khatanga Gulf, Laptev Sea to the south from New Siberian Islands, eastern and central part of East Siberian Sea, Gulf of Ob), which makes the areal analysis incomplete. Are there Sentinel-1 data available? Could they also be included to the analysis?

Most of these are available, but as stated through the manuscript, we limited our analysis to 50 interferograms (now 52 with the inclusion of the Eastern Laptev Sea) and the general coastline. If we were to include the island groups and all the gulfs we would have to increase the analysis substantially, which is very time consuming. We have not attempted to make a complete pan-Arctic dataset, but aim here to show what is possible on a large scale. The only lack of accessible data we observed for the coastline was the Gulf of Ob. We have analyzed imagery in this Gulf in 2018 when imagery was available. However, we did not include this imagery to stick with a one-year analysis in this work.

Moreover, I think that bathymetry should be used together with interferograms to make an adequate delineation. IBCAO data can be used to identify potential bottomfast ice areas and also can clearly exclude areas, erroneously mapped as bottomfast ice if water depth exceeds a certain threshold (for instance, 2 m, see the figure below).

Thank you for this suggestion. Our concern in that the IBACO or any other bathymetry dataset is not known to be accurate in shallow areas. By including such dataset in the

analysis would then at the same time introduce added uncertainties and sources for error. For instance, a large part of the red area you highlight is interpreted as above sea level in the backscatter image (see Fig 5). Dammann et al. (2018c) identified good correlation between phase and areas of known bottomfast ice. They used a high resolution local bathymetry dataset, but also there, the necessary interpolation lead to uncertainties, which would be much larger for a large-scale dataset such as IBACO. Furthermore, Dammann et al. (2018c) did not find reason to suspect substantial areas to be falsely identified as bottomfast ice simply because floating ice will inevitably exhibit some lateral motion during the 12-day temporal lag considered here.

As the Referee 1 suggests, I also think you could use the backscatter images to support the delineation of the bottomfast ice, and potentially other classes (or to show that the backscatter has no / limited use for this purpose).

Agree. This has now been included

Furthermore, a big improvement could be an unwrapping of interferograms w.r.t. the land area, which should be stable during the winter. By doing so, you could derive the magnitude of the ice movement, and then the ice stability classes could be distinguished based on the magnitude. In this case, the subjective delineation could be substituted by an automated classification. Moreover, the vertical and horizontal movements could potentially also be distinguished using ascending and descending orbits. This is probably beyond the study intention, but definitely should be discussed.

Thank you for these comments. This is all something we initially contemplated as well, but we found not to be an ideal strategy for the following reasons:

Phase unwrapping over sea ice can be tricky due to a large amount of discontinuities as discussed in Morris et al. (1999) and Dammann et al. (2016). It would therefore be difficult to make sure this can be done in a straight-forward manner for use in operational analysis and to provide consistent results everywhere. With that said, it could be done to some degree, but most likely without any improvement in the stability product.

The derived deformation (through phase unwrapping) will only signify deformation in cross track, hence the total deformation will strongly depend on the mode and direction of deformation. To resolve this one would have to use inverse modeling similar to the approach used by Dammann et al. (2018b) to determine the level of deformation. However, this approach would not easily lend itself to an analysis at this spatial scale. Even if one were to be able to back out the deformation, it would still be difficult to determine the stability of the ice cover since the deformation depends on forcing conditions between acquisitions. In other words, it would be difficult to determine whether low deformation is due to stable ice or low wind speed or other forcing conditions.

In the discussion section we now explain: "Without evaluating the phase response for each area of interest in detail during different forcing scenarios, it may be problematic to understand under what conditions the ice remains stable. Classification of stability based on relative differences in fringe density is also complicated by the use of non-simultaneous interferograms to provide complete coverage of a region. "With the approach we suggest here, we consider changes in fringe spacing within the same scene, hence a strong gradient will not be due to a change in forcing conditions, but rather a sign of different stability.

Vertical and horizontal motion cannot easily be discriminated using ascending and descending passes as discussed further in Dammann et al. (2016), since deformation changes too much between passes as opposed to when used for glaciers. However, by evaluating the fringe patterns, it is possible to identify vertical deformation as described in detail in Dammann et al. (2016).

You suggest that the used approach is rather simple and does not require SAR expertise. However, you do need to have some expertise to decide on the phase change gradient, and this seems to be a major problem at the moment.

This has now been moderated to: P10L18: "Hence, the approach can potentially be adapted by organizations without the need for trained SAR experts."

We agree some training will be needed, but we argue one doesn't need to be a SAR expert. The subjectivity is addressed above.

The unwrapping step is included into every standard interferometric workflow, and therefore, could be tried, at least exemplarily. This, perhaps, could also improve the identification of the bottomfast ice, as the InSAR phase in this case should be the same as over the land.

As discussed above, the phase unwrapping is not expected to add much value to the analysis we present in this work, but would add an added step and complication to the approach.

The authors provide a validation of their result for the Beaufort Sea, by comparing stability zones with long-term frequent position of the landfast ice edge from Mahoney et al. (2014). As stated in the Results chapter, they associate the second discontinuity in the InSAR phase with the nodes, identified in Mahoney et al. (2014). This discontinuity separates the stable/non-stable zones, whereas the nodes coincide with the edge of the landfast ice. Therefore, it appears to me that the comparison with the nodes from Mahoney et al. (2014) can only validate the overall extent of the fast ice, without being able to validate stable/non-stable classes identification. Perhaps, it can be partly attributed to the interannual variability of the landfast ice edge position (and position of the border between stable and non-stable fast ice). Nevertheless, I find such comparison important, and strongly suggest to add comparisons for other areas, for instance, for the Laptev Sea, where the positions of the fast ice edge are presented in Selyuzhenok et al. (2015). For that, the full extent of the fast ice in the Laptev Sea would be required, while at the moment, the important ice grounding zones are excluded from the analysis (see Figure in my general comment).

It would be great with more validation data, but we are here limited in terms of data availability of landfast sea ice stability.

Mahoney et al. (2014) find that the landfast ice edge in this section of the Beaufort Sea

coincide with the 20 m isobath. Due to the dynamic ice conditions in this region, it is expected that the prevailing outermost margin will be supported by grounded ridges. The nodes will thus only occur in areas of frequently stabilized ice.

Comparing our analysis with Selyuzhenok et al. (2015) is a great idea. We have now added two more images in the Laptev Sea to make that possible. We have now added to Section 3.2:

P6L12: "Due to limited data availability in the central part of the Laptev Sea, the interferogram of the ice surrounding the Stolbovoy Island had to be acquired as early as February before the time of maximum ice stability (Figure 3). Even so, it is still clear that the area to the northwest of the island features stabilized ice (see "D" in Figure 4). This exact area features a shoal of < 10 m water depth leading to earlier formation of fast ice than the surrounding areas (Selyuzhenok et al., 2015) likely due to the formation of grounded ridges on the shoal resulting in increased stability. The ability of interferometry to signify the stabilized ice in this region lends support to our approach."

Some concerns about a proper terminology: can you use simply "stable" and "non-stable" (or "unstable") ice classes, instead of stabilised / non-stabilised? Or do you want to emphasize that the stabilised class can be unstable before grounding?

Good question. The use of the terms stabilized and non-stabilized is essential. Whether the ice is stable depend on the individual ice users and scenarios as outlined in Dammann et al. (2018a). For instance, unstable ice for an industrial ice road, may not be viewed as unstable ice by coastal communities. Therefore, we choose to focus on relative local stabilization of the ice rather than trying to derive boundaries between stable and unstable ice as seen from the individual stakeholders perspectives. However, even if one wanted to use a threshold to determine stabile ice, it would be difficult to determine this based on the variable forcing scenarios which largely impacts the fringe density. For instance, it would be difficult to determine whether low fringe density is due to low forcing conditions or stable ice. This is why we chose to focus
on local stabilization as we also discuss above in response to your earlier suggestion around phase unwrapping.

Also, the three stability classes are called "regimes", "zones", or "classes" throughout the paper. I suggest using one term consistently, and for me personally, the term "regime" does not seem to be correct, as you are focusing on a snapshot and not time series. Also, I'm not sure but I think the term "delineation" imply a linear feature, so when you refer to an areal object, maybe it is better to use "mapping"?

Great point. We have changed to mapping of zones throughout.

I also recommend the language proofreading to simplify and smooth the style and improve the clarity. Some sentences are very long, using redundant words, too many subordinate clauses. A tip from Elsevier, for instance: "Nowadays, the average length of sentences in scientific writing is about 12 to 17 words". There also are a few typos.

We have conducted another proofreading to simplify and sorting out additional typos.

Specific comments

Thank you so much for providing such through review in terms of these specific comments. It is a great help!

p.2 lines 5-8: sentence is too long, and the second part of it is not clear ("defined here as the immobility" – defined where?)

This sentence has been split and simplified:

P2L5: "Similar to the drifting pack ice, landfast ice has declined significantly during the last few decades, in particular in terms of delayed freeze up (Mahoney et al., 2014;Selyuzhenok et al., 2015). Later freeze up critically impacts stakeholders through increased mobility (reduced stability) of the landfast ice in response to wind, ocean, or ice forcing (Dammann, 2017)."

p.2 lines 8-11: could you split the sentence in two?

Done

p.2 line 13: "from left to right" – better use "from coast towards open ocean" as you should not describe a figure in the main text.

Good suggestion. Changed.

p.2 line 14: can grow laterally

Included p.2 line 19: "high to moderately stable landfast ice..."

Good catch. Changed.

p.2 line 21-27: I suggest to add more structure to this part. Do you want to say that low-stability ice is simply dangerous for ocean-based operations? How are grounded ridges related to the low-stability ice? Do ice arches belong to the stable class?

Good point. This was not very clear. We have changed this to:

P2L21: "Low-stability ice is potentially relevant for ocean-based operations such as shipping trans-Arctic passages close to the coast where, patches of landfast ice occasionally break off and drift into nearby shipping lanes, potentially causing hazards. Even in areas hundreds of km from landfast ice can be impacted through the failure of ice arches. Ice arches consist of stationary ice forming between islands during freeze-up and when collapsing in the spring can send hazardous old ice into shipping routes (Bailey, 1957;Wilson et al., 2004;Barber et al., 2018). Stability is also of relevance for destinational cargo shipping in the Arctic as less stable, thinner ice is easier to break through resulting in opportunities for docking in areas of substantial landfast ice. For navigating through landfast ice, stabilization through ridging is also important to identify since ridges can be problematic to navigate and associated with hazards (Hui et al., 2017)."

p.2 line 28: No need to specify "sea", just landfast ice

Taken out

p.2 line 29: I believe you do not need to mention in situ data, it is absolutely clear

Taken out

p.2 line 33-36: consider to compact the sentence

Done

p.2 line 37: what kind of types? Age?

Specified to multiyear and first-year.

p.2 line 38: because the movements are too small to use change detection?

Yes. Included:

P3L1: "However, backscatter does not give information pertaining to ice stability since the internal movement of the landfast ice is too small (mm/day) to be identified with change detection."

p.3 line 1: between two SAR images

Included

p.3 line 2: either specify that this is the case of the sea ice, or leave it more general for any kind of displacement/topography analysis

Took out sea ice.

p.3 line 11: "These studies have demonstrated..."

Done

p. 3 line 13: "as a mean..."

Done

p.3 line 15-17: consider to split and compact the sentence.

Changed to:

P3L17: "Hence, we explore InSAR as a tool to provide pan-Arctic information pertaining to stability relevant to subsea permafrost, biological habitats, and sea ice use."

p.3 line 17: you do not need to determine the ability to create interferogram. It can be created for any interferometric pair. Reformulate or simply delete this.

Taken out

p.3 line 17-22: reformulate and simplify the sentences and do not refer to the Table and Sections, it is distracting.

Simplified to:

P3L18: "The goal of this work is to determine the Sentinel-1 interferometric data availability along substantial parts of the circumpolar coastlines and explore whether the different ice stability zones can consistently be analyzed and mapped in different geographic regions."

p.3 line 27: in slant or ground range? This sentence should go after you specify that you use IW mode

Specified. Done

p.3, line 33: "a sample.."?

Supposed to be as is: ample

p.3 line 36-37: this is a repetition from the previous chapter
p. 3 line 37: "the InSAR phase may be related..."

Agree. Took out first sentence and moved the citation to past chapter.

p.4 line 3: can you specify the limits? I believe that the critical topography height can be calculated from the maximum baseline.

This has now been specified to 50 m.

p. 4 line 5: what means "only" here? You should also mention here atmospheric phase delay and other contributors to the InSAR phase.

This has been changed to:

P4L8: "Of the phase change attributed to motion, only displacement in line-of-sight direction. . ."

And we have included:

P4L4: "A phase signature can sometimes also be attributed to factors not related to surface motion or topography such as atmospheric phase delay and coregistration errors, but these effects are small compared to ice motion and can often be corrected."

p. 4 line 7-8: "wrap around" – find more rigorous phrasing

Changed to:

P4L11: ". . .phase values to result in more than one fringe and ambiguous phase values."

p. 4 line 8: "The results is..." – "The interferogram is..."

Changed

p. 4 line 12: again, any interferogram can be successfully created, please reformulate

Changed to:

P4L16: "The interferometric phase values will only be useful if scattering elements. . .

p. 4 line 13: ranging between 0 and 1?

Agree. Changed

p. 4 line 14: "A complete list.." – this sentence does not flow logically from the previous

one. Either remove it or provide a logical connection.

Taken out

p. 4 line 19-21: After such a long technical introduction to the InSAR, the processing flow in two lines is way too short. Please provide more details.

This has now been included:

P4L24: "The images were first geometrically coregistered to ensure that the images cover exactly the same area with sub-pixel accuracy. The images were then mulit-looked 10 and 2 pixels in ground range and azimuth respectively resulting in reduced speckle and a final pixel spacing of roughly 23x28 m. Next, spectral filtering were performed to ensure both images comprise the same spectral range, which reduces phase noise in the final interferogram. The interferometric phase was calculated for each pixel in the geocoded and filtered images. Furthermore, the expected ramping phase in cross-track direction from a stationary flat earth surface was removed. The phase noise of the final interferogram was also reduced using an adaptive phase filter. All of these steps were completed for each interferogram using the GAMMA RS software (Werner et al., 2000)."

p. 4 line 23: in my view, "determining relative strain rates", even roughly, is a little bit of exaggeration of what you really do. Just say that you use fringe density as an indicator of stability.

Done

p. 4 line 24-25: I don't see how this sentence helps in understanding the following one. Consider removing.

We agree this sentence was unclear. It is now rewritten:

P4L33: "There are many factors that affect fringe density in addition to stability, including the atmospheric and ocean forcing conditions, satellite viewing geometry, and

the prevalent mode of ice deformation (Dammann et al., 2016), making it problematic to evaluate absolute stability from fringe density alone. Instead, we focus on abrupt changes in fringe spacing within individual interferograms that allow us to identify variations within an area imaged under largely the same conditions, but where fringe gradients towards lower fringe density corresponding to increasing ice stability and possibly a different stability zone outlined in Table 2."

p. 4 line 25: "abrupt changes"?

Done

p. 4 line 26: "...to identify three stability zones: list them". Consider joining Tables 1 and 2.

Done:

P5L5: "The three zones (i.e. bottomfast ice, stabilized ice, non-stabilized ice) are subjectively…"

p. 4 line 33: in what study?

Changed to:

P5L11: "The approach we present here"

p. 5 line 2: was coherence low for all available pairs for these regions? I see that you mention that you do not "attempt to derive alternative interferograms in these cases" but this data gap can probably be easily filled, even in potentially operational mode.

Through all of our InSAR work utilizing different sensors and time frames over the years, this region does not feature good coherence, so utilizing different scenes would likely not result in improved coherence.

p. 5 line 7: I suggest not to start your Results with a description of results from the other study, even though relevant here. I suggest to move the chapter to the end of

Results or even to Discussion.

The first and second sections have now been switched

p. 5 line 10-11: consider simplify the sentence; you do not need here the technical explanation how these edges were derived.

This has been simplified

p. 5 line 14-15: do not describe the figure in the main text.

This has been taken out

p. 5 line 21: message in parentheses is confusing. I think you can see a pattern in these areas as well. Depends on what you call a pattern.

Taken out

p. 5 line 22: "This discontinuity has been shown...". I do not think it can be called a gradient.

This has been changed

p. 5 line 24: see my general comment

Addressed in response to the general comment

p. 5 line 25: "reoccurring"

Changed

p. 5 line 27: what is meant by reduced phase response? Also, the sentence is too long.

Taken out, which also lead to shorter sentence.

p. 5 line 27-29: please reformulate and compact the sentence, it is hard to follow. What are the points of higher stability and how the points can be seaward?
This has been clarified:

P7L3: "The discontinuity is not a straight line, but features multiple curves towards land ending in seaward points. At these points, the stability is higher than adjacent areas with the same distance from shore similar to the expected increased stability directly shoreward of grounded ridges."

p. 5 line 32: Mapping pan-Arctic landfast ice stability zones?

Regimes has been changed to zones throughout the manuscript when referring to the three zones.

p. 5 line 34: please do not refer to the Sections

Taken out

p. 5 line 35: see my general comment. Without a convincing explanation how the bottomfast ice was distinguished from the non-bottomfast, I think that the current mapping is not accurate enough.

A short explanation has now been included:

P5L1: "The often-strong fringe gradient leading to an area of near-zero phase change has been shown to represent the boundary of where the ice is frozen to the sea floor and can subsequently be used to map bottomfast ice (Dammann et al., 2018c)."

We are also stating in Section 3.2:

P6L35 "One discontinuity separates the area of near-zero phase change from an area with relatively low fringe density. This discontinuity indicates the boundary between bottomfast and floating ice as two of these interferograms were validated both on Elson Lagoon near UtqiaĄ̈vik and the Colville Delta (Dammann et al., 2018c)"

p. 5 line 36, please do not refer to the Sections

OK. Taken out throughout the manuscript

p. 6 line 1-2: what is meant by substantial bottomfast ice?

specified to:

P5L29: "...with extensive bottomfast ice reaching several km from shore are..."

p. 6 line 3: what means Accordingly here?

Doesn't belong. Taken out

p. 6 line 7-8 please do not refer to the figures and Sections here, it should be clear without that.

Agree. Done

p. 6 line 12: revise the sentence, something is missing

This has been changed to:

P6L1: " Conversely, the greatest area of stabilized landfast ice was found in the adjacent Beaufort Sea, with almost equal extent of non-stabilized and stabilized landfast ice."

p. 6 line 14: "...landfast ice..."

Done

p. 6 line 15-18: please shorten and clarify the sentence

Done

p. 6 line 21-22: identified where?

Added:

P6L5: "along the Russian Arctic coast"

p. 6 line 22-23: as in my general comment examples, I do not see how this area is different from, for instance, the area around Bely Island (right above the Yamal penin-

sula).

Yes, some of the areas by Yamal should be categorized as stabilized. This has now been changed

p. 6 line 24-25: I think in this case you simply should not mention it here at all.

Agree. Taken out

p. 6 line 27: "...limitations for mapping stability classes"?

Agree. That is better. Changed.

p. 6 line 29: typically or in your study?

Changed to:

P7L13: "...have been identified"

p. 6, line 36-p.7 line 5: this repeats the paragraph in the Result section. I suggest to remove it from the Results as it is a discussion point with speculations.

Good catch. This has been taken out and merged with the text in the discussion

p. 7 line 14: how the definition of the "highly stable" from Eicken et al. (2005) is related to the definition of stable in your study? Can they be compared?

This reference is to the entire region, so it wouldn't be possible to do a direct comparison p. 7 line 15: or this part is also prone to the tidal movements?

Yes, it is possible, but if the fringes were solely a sign of changed vertical motion, it would mean that the ice was elevated several meters in certain regions, which is implausible. We now include this:

P8L9: "Based on the overall fringe counts and patterns, only part of the phase response is likely attributed to tidal motion"

p. 7 line 17: SAR backscatter analysis?

Changed

p. 7 line 18-19: but the mobile ice would be incoherent in your results? Also "one month after the initial freeze up" – while you consider late winter situation. As recommended in the general comment, please include more scenes for the Laptev Sea to have an entire fast ice extent – maybe it can shed more light on the situation here.

We are here merely speaking to the stability of this region as a whole and that it may not be as stable as previously thought. The time frame is indeed different. More scenes are now included and are discussed above.

p. 7 line 27: why to mention it here if it is discussed below?

Taken out

p. 7 line 28-30: or it can aid in the interpretation, e.g. by confirming the pattern, or by featuring the temporal changes between interferograms. Also, I do not see what is meant to be in the northernmost region of the Lena Delta in Figure 4.

Changed to:

P8L18: "Hence, the use of interferograms based on different dates can aid interpretation by confirming consistent fringe patterns and identify temporal changes. However, the temporal change will also result in a phase gradient at the image stitching not related to different stability zones, which may further complicate the mapping process." Took out reference to Lena Delta since not strictly necessary

p. 7 line 33: is this connected to the previous sentence? If not please restructure

Not related. Swapped the following sentences to make it more clear.

p. 7 line 35: so they are not available in late winter 2017 or may not be available?

May not be available. This has been further specified:

P8L29: "Furthermore, IW imagery are predominately acquired over land, hence it is

likely not possible to construct interferograms away from the coast to cover extensive landfast ice approaching the 250 km IW swath such as that in the East Siberian Sea."

p. 7 line 36: as you say "consistent coherence loss", now I probably got an answer to my question to p. 5 line 2. Can it simply mean that there is no landfast ice?

In reference to your earlier question, there may be some areas in the Kotzebue region where the dynamics is large enough to the point where the ice may not be called landfast, such as in the Kotzebue Sound. However, the coherence loss is more widespread covering inlets that are known to be stable.

p. 7 line 36-37: temporal progression or spatial? Not clear what is meant here.

Spatial. This is now included

p. 7 line 38: why would we be interested in using this analysis for the melt onset period, when the ice starts to degrade and be unstable anyway? Please develop you thought here.

Included:

P8L27: "It is worth mentioning that this technique can only be used before the onset of melt when widespread coherence loss occurs, hence it is not possible to evaluate the retreat of bottomfast ice or reduction of ice stability in response to melt"

p. 8 line 13: "...multiyear ice..."

included

p. 8 line 14: In our study?

Not limited to our study, so suggest to keep it as is

p. 8 line 15: Sentinel-1 backscatter imagery? Did it capture the formation?

Included. Took out "formation"

p. 8 line 16-17: this goes to the figure caption

Done

p. 8 line 36: potential applications?

Agree. Changed

p. 9 line 23-26: please split the sentence

Done

p. 9 line 27: "multiple"→three?

Changed

p. 9 line 28-31: aren't these classes used for all seas? Why to specify Beaufort Sea here?

They are, but we are arguing that the classes fit good in the Beaufort, but that the Russian sector potentially warrant additional classes

p. 10 line 3-8: this is too detailed for the Conclusions, and mainly repeats the paragraph from the Results. Please generalise your findings in Conclusions.

Taken out

p. 10 line 19: did you actually consider year-to-year timescale?

Good point. Taken out

p. 14 line 4: remove the word "ice" before "sea ice"

Done

p. 15 Figure 2 caption: Conceptual scheme?

Included

p. 16 Figure 3: place the red ovals a,b,c on top of the b) panel; caption: explain a,b,c nodes in a).\

Done

p. 17 Figure 4. I suggest to redistribute the panels in such a way that all of them can be enlarged.

The images are currently distributed so they can take up a page in landscape mode. Unless you suggest images on separate pages in separate figures, we think this will work quite well.

p. 18, Figure 5: enlarge, the same as figure 4. The river you refer as to Angara, is Yenisei! Please check other geographical names and add river shapes on the maps.

Same as above. Good catch. Checked. Names of rivers are added for rough geographical context in relation to the rivers. We feel that adding river shapes of all the different channels etc. may be distracting and not strictly relevant.

p. 19, Figure 6: please add the overview figure showing the location of the arch. Use additional graphic to delineate the arch and the flow direction. Some readers are unfamiliar with this sea ice feature.

Good point. This has been included

Please also note the supplement to this comment:
https://www.the-cryosphere-discuss.net/tc-2018-129/tc-2018-129-AC1-supplement.pdf

―――――――――――――――――――

[Figure]

**Fig. 1.**

The map shows the Laptev Sea region with numbered annotations:

**1** How did you distinguish these 2 classes?

Laptev Sea

**2** This area is known to have some shoals where the ice is grounded. Can you include more scenes, covering the entire extent of fast ice?

New Siberian Islands

**4** How was this peculiar area identified? Why adjacent areas were not included to this class?

C

Lena Delta

**3** it is really hard to understand how decision on the shape of all these tiny features, classified as bottomfast ice, was made

Yana River

N

0  50  100    200 km

[Figure]

**Fig. 2.**

[Figure]

**Fig. 3.**

[Figure]

**Fig. 4.**

[Figure]

**Fig. 5.**

---

## Referee Report (RR1)

Dear authors,

Thanks for incorporating my comments and suggestions from the first round. The manuscript has already been improved. However, there is still some work to do in my opinion. I commented the manuscript with suggestions and questions. Main points are:

Still not clear how you classify fringe patterns. Many areas which look very similar are classified differently in different regions. See my comments in the manuscript to the Laptev Sea, for example. Therefore, I suggest to describe the manual procedure and add representative examples in Methods, as well as in the Results.

Structure of the paper can be improved. The entire chapter on the ice arch can be split and parts could be moved to the Introduction, Methods, Results and Discussion accordingly. The chapter on the comparison of your results with previous studies should be a part of Discussion. Your very results could be presented in a more detailed way, again with examples.

Furthermore, I insist on a professional proofreading to make the writing clearer and smoother. Many sentences could be simplified and shortened.

I hope to see the updated version of your manuscript soon!

Best regards,

Your reviewer
* * *
Dyre O. Dammann[1], Leif E.B. Eriksson[1], Andrew R. Mahoney[2], Hajo Eicken[3], Franz J. Meyer[2]

[1]Department of Earth, Space, and Environment, Chalmers University of Technology, Gothenburg, 412 96, Sweden
[2]Geophysical institute, University of Alaska Fairbanks, Fairbanks, 99775, USA
[3]International Arctic Research Center, University of Alaska Fairbanks, Fairbanks, 99775, USA

*Correspondence to*: Dyre O. Dammann (dyre.dammann@chalmers.se)

**Abstract.** Arctic landfast sea ice has undergone substantial changes in recent decades affecting ice stability with potential impacts on ice travel by coastal populations and industry ice roads. The role of landfast ice as an important habitat has also evolved. We present a novel approach to evaluate sea ice stability on a pan-Arctic scale using Synthetic Aperture Radar Interferometry (InSAR). Using Sentinel-1 images from spring 2017, the approach discriminates between bottomfast,  stabilized and non-stabilized floating landfast ice over the main marginal seas of the Arctic Ocean (Beaufort, Chukchi, East Siberian, Laptev and Kara Seas). The analysis draws on evaluation of  relative changes in interferometric fringe patterns. This first comprehensive assessment of Arctic bottomfast sea ice extent revealed that by area most of the bottomfast sea ice is situated around river mouths and coastal shallows in the Laptev and East Siberian Seas, covering roughly 4.1 and 5.1 thousand km2 respectively. The  between non-stabilized and stabilized ice is lowest in the  at almost unity, and highest in the adjacent Chukchi Sea. Beyond the simple mapping of landfast ice zones, this work provides a new  of how stability zones may vary between regions and over time. InSAR-derived stability  may serve as a strategic planning and tactical decision-support tool for different uses of coastal ice. Such information may also inform assessments of important sea ice habitats. In a case study, we examined an ice arch situated in Nares Strait demonstrating that interferograms may reveal early-warning signals for the break-up of stationary sea ice.

**1 Introduction**

**1.1 Landfast sea ice stability and stakeholder dependence**

Sea ice is an important component of Arctic ecosystems and provides important services as a climate regulator (Screen and Simmonds, 2010), habitat for marine biota (Thomas, 2017), as well as a platform for coastal populations (Krupnik et al., 2010). During the last century, an expansion of transportation and resource extraction have led to increased human presence in the Arctic and further diversification of ice use (Eicken et al., 2009). The recent retreat of sea ice observed throughout the past several decades (Stroeve et al., 2012;Comiso and Hall, 2014;Meier et al., 2014) has already resulted in widespread consequences for ice users (Druckenmiller et al., 2013;Aporta and Higgs, 2005;Fienup-Riordan and Rearden, 2010;Orviku et al., 2011;ACIA, 2004) and increasing hazards (Eicken and Mahoney, 2015;Ford et al., 2008). At the same time, the related increased accessibility to Arctic waters (Stephenson et al., 2011) is leading to increasing ship traffic and resource exploration (Lovecraft and Eicken, 2011;Eguíluz et al., 2016). It is recognized that the sea ice conditions for future Arctic marine operations will be challenging and will require substantial monitoring and improved regional observations (Ellis and Brigham, 2009) at the scale necessary for assessing environmental hazards and effective emergency response (Eicken et al., 2011).

Most of the Arctic ocean is dominated by drifting pack ice, whereas stationary landfast ice occupies much of the Arctic coastlines roughly between November and June depending on location (Figure 1) (Yu et al., 2014). Although the landfast ice is stationary, it does deform internally at the cm- to m-scale on timescales of days to months (Dammann et al., 2016). The often several km to up to hundreds of km wide sections of landfast ice are held in place by grounded ridges, islands, or coastline morphology, such as embayments or fjords. Similar to the drifting pack ice, landfast ice has declined significantly during the last few decades, in particular in terms of delayed freeze up (Mahoney et al., 2014;Selyuzhenok et al., 2015). Later freeze up critically impacts stakeholders through increased mobility (reduced stability) of the landfast ice in response to wind, ocean, or ice forcing (Dammann, 2017). Previous research suggests that landfast ice stability can be expressed in terms of the combined frictional resistance provided by relevant grounding or attachment points (e.g., islands and grounded ridges) (Druckenmiller, 2011;Mahoney et al., 2007b). The stability in part determines the rate at which the ice deforms and ultimately the severity of break-out events or magnitude of structural defects. We therefore suggest that landfast sea ice can be further categorized into four regimes, defined through their respective stability (Table 1). A typical landfast ice regime is illustrated in Figure 2, where the stability of the landfast ice area decreases from the coast towards the open ocean (Dammann et al., 2016).

Bottomfast sea ice can grow laterally to the km-scale during winter depending on local bathymetry (Solomon et al., 2008;Stevens et al., 2010). The bottomfast ice allows for heat loss from the sea floor and is therefore an integral part of aggregating and maintaining subsea permafrost (Stevens et al., 2008;Stevens et al., 2010;Stevens, 2011), controlling coastal stability/morphology (Eicken et al., 2005;Are and Reimnitz, 2000), and sediment properties (Solomon et al., 2008). Bottomfast ice is also relevant for fish as it reduces habitable shallow waters during winter (Hirose et al., 2008). Bottomfast ice is also of importance for on-ice operations as it can support a much larger load than floating ice. High to moderately stable landfast ice is of relevance to industrial (Potter et al., 1981)

and subsistence ice use (Druckenmiller et al., 2013), but also for habitats (Tibbles et al., In press). For instance, ringed seals are dependent on stable landfast ice for denning (Smith, 1980). Low-stability ice is potentially relevant for ocean-based operations such as shipping through trans-Arctic passages close to the coast where patches of landfast ice occasionally break off and drift into nearby shipping lanes, potentially causing hazards. Even areas hundreds of km from landfast ice can be impacted through the failure of ice arches. Ice arches consist of stationary ice forming between islands during freeze-up and when collapsing in the spring can send hazardous old ice 
[revised manuscript text omitted]
., 2016), making it problematic to evaluate absolute stability from fringe density alone. Instead, we focus on abrupt changes in fringe spacing within individual interferograms that allow us to identify variations within an area imaged under largely the same conditions. Trends from higher to lower fringe density will, in such cases, likely correspond to increasing ice stability.  three different stability zones: bottomfast ice, stabilized ice, and non-stabilized ice (Table 1). The often-strong fringe gradient leading to an area of near-zero phase change has been shown to represent the boundary of where the ice is frozen to the sea floor and can subsequently be used to map bottomfast ice (Dammann et al., 2018c). In Table 1, the two sheltered regimes will both lead to reduced fringe density and can be difficult to discriminate based on InSAR data alone. These regimes are therefore assigned to the zone "stabilized ice". The three zones (i.e. bottomfast ice, stabilized ice, non-stabilized ice) are subjectively and manually mapped without the use of specific threshold values.

The zones themselves are therefore based on relative stability in terms of whether the ice is anchored or sheltered. A measure of whether the ice is stable would depend on the specific stakeholders and their dependence on stability. As an example, on shorter time scales, industry ice roads would be able to accommodate less strain than community ice trails due to different mode of transportation and user specific needs. Further steps to identify such thresholds are outlined in Dammann et al. (2018a).

The approach we present here, opens up the possibility of mapping landfast sea ice zones on a pan-Arctic scale. To demonstrate,
we used Sentinel-1 data acquired March through May 2017 and generated 52 interferograms that cover almost the entire continental coastlines of the Beaufort, Chukchi, East Siberian, Laptev, and Kara Seas. To reduce computational costs, we omitted Greenland, some island groups and in particular the Canadian Archipelago, which are characterized by extensive coastline lengths. The Alaskan and Russian coastlines have high economic significance for the shipping and natural resource industries and feature dynamically diverse ice regimes and large areas of bottomfast ice are expected in these regions. Except for one approximately 50
km-long section of coast in the Kara Sea and the eastern Laptev Sea, multiple InSAR compatible pairs were available for the specified time frame. This allowed us to select interferograms centered around the end of April, when most Arctic landfast ice is at its maximum extent. Coherence loss was evident in some areas, in particular in the Chukchi Sea, such as in the Kotzebue Sound likely partly due to ice motion, subsurface thinning from river runoff, and low signal-to-noise ratio. However, to ensure a realistic representation of what an operationally-produced synoptic, contiguous pan-Arctic interferogram would look like, we did not
attempt to derive alternative interferograms in these cases.

**3 Results**

**3.1 Mapping pan-Arctic ice stability zones**

The interferograms produced in this work (Figure 3) allowed for a detailed map of landfast ice including the identification of three landfast ice stability zones: bottomfast ice, stabilized landfast ice, and non-stabilized landfast ice. To our knowledge, our results
(Figure 4) represent the first mapping of bottomfast ice extent at this scale and the first attempt at any scale to map the extents of different landfast ice stability zones.  most areas with extensive bottomfast ice reaching several km from shore are located either in the vicinity of river deltas or within lagoons. However, a prominent exception is the coastline of the western East Siberian Sea, where our analysis shows substantial amounts of bottomfast ice even tens of kilometers away from any major rivers. The East Siberian Sea with its three large river systems (the Indigirka,
Bogdashkina, and Kolyma Rivers) contains the most bottomfast ice of the regions considered here (Table 2). The Laptev Sea also contains a large fraction of the Arctic bottomfast sea ice  concentrated around the Lena and Yana Deltas.

The map of "stabilized" and "non-stabilized" landfast ice is based on subjective interpretation of the interferograms. Both stabilized and non-stabilized landfast ice zones were found in all marginal seas (Table 2), though their relative contributions to overall landfast ice extent varied widely. For example, in the Chukchi Sea, we identified the vast majority of the landfast ice as non-
stabilized, with stabilized landfast ice occupying less area than the bottomfast ice. Conversely, the greatest area of stabilized landfast ice was found in the adjacent Beaufort Sea, with a larger extent of stabilized than non-stabilized landfast ice.

In the East Siberian, Laptev and Kara Seas, the distinction between stabilized and non-stabilized landfast ice is not as straightforward as in the Beaufort and Chukchi Seas. Even so, it is clear that landfast ice extent in the East Siberian, Laptev and Kara Seas is dominated by vast areas of non-stabilized ice. Unlike the Chukchi Sea, we  identified significant areas of stabilized landfast ice along the Russian Arctic coast. In the Kara Sea, these are primarily found between the islands of the Nordenskiöld Archipelago in the east, but the most extensive regions of stabilized landfast ice in our study region (those that extend furthest from the coast) are found in the Laptev and East Siberian Seas (areas labeled A, B, and C in Figure 4).

**3.2 Comparing stability zones with areas of known ice stability**

To investigate whether the stability zones are reasonable,  Due to limited  data availability  the interferogram  had to be acquired as early as February before the time of maximum ice stability (Figure 3). Even so, it is clear that the area to the northwest of the island features stabilized ice (see "D" in Figure 4). The stabilized ice area appears triangular due to the much earlier acquisition date of the surrounding interferogram. This exact area features a shoal of < 10 m water depth leading to earlier formation of fast ice than the surrounding areas (Selyuzhenok et al., 2015) likely due to the formation of grounded ridges on the shoal resulting in increased stability. The ability of interferometry to identify stabilized ice in this region lends support to our approach.

We will also consider the landfast ice along the Alaska Beaufort Sea coastline,  in certain regions, the ice extent is similar over time scales from months to years (see areas highlighted in the figure). It was found that these regions ("nodes") of consistent landfast ice extent are often tied to the location of the 20-m isobath, a water depth associated with grounding of pressure ridges (Mahoney et al., 2014) (Figure 5a). A field study and indigenous knowledge also indicate persistent grounded ridges in the location of the node closest to Utqiaġvik, Alaska (Meyer et al., 2011).

We created three interferograms acquired during the period April 8 – May 9 to cover the same stretch of coastline. The respective master images exhibit a sharp discontinuity in backscatter (see arrows in Figure 5b) along the  location of the landfast ice edge . Determining the landfast ice edge can in some instances be achieved by evaluating a single  image as here, but is not consistent and only works in cases where there are stark discontinuities in backscatter due to different ice types or a severely deformed landfast ice edge.  These interferograms reveal a wide range of fringe densities, ranging from near constant phase for areas close to the coast to the point where fringes are dense enough to almost merge near the landfast ice edge. It is also apparent that the fringe density does not linearly increase with distance from the coast, but rather changes along two distinct discontinuities.

One discontinuity separates the area of near-zero phase change from an area with relatively low fringe density. This discontinuity indicates the boundary between bottomfast and floating ice . The second discontinuity appears to largely coincide with locations of the nodes identified by Mahoney et al. [2007a; 2014], which are thought to be associated with reoccurring grounded ice features (Figure 5a) (Meyer et al., 2011). This finding is expected since grounded ridges are known to stabilize the landfast ice leading to reduced strain shoreward of the grounding points (Mahoney et al., 2007b;Druckenmiller, 2011).

The discontinuity is not a straight line, but features multiple curves towards land ending in seaward points. At these points, the stability is higher than adjacent areas with the same distance from shore similar to the expected increased stability directly shoreward of grounded ridges. We performed a subjective, manual mapping (i.e. without the use of a specific threshold) of the strong phase gradients in Figure 5d and concluded that the ice regions separated by these discontinuities consist of non-stabilized, 5 stabilized, and bottomfast ice (Figure 5e). Although the landfast ice edge can in some instances be mapped using a single backscatter image, the mapping of bottomfast and stabilized ice cannot be discriminated as is apparent by comparing the map from Figure 5e with Figure 5b.

**4 Discussion**

**4.1 Methodological limitations for mapping stability zones**

Although there are a number of sources of uncertainty that affect our map of landfast ice and its relative stability, it is clear that not all landfast ice is equally stable and at least three different zones of stability have been identified. In some areas, the map of bottomfast ice has had to be approximated on the sub-km scale due to ambiguities associated with low fringe density or fringes parallel to the bottomfast ice edge (Dammann et al., 2018c). We also acknowledge that small islands or sandbars not represented by our coast mask may be erroneously identified as bottomfast ice. However, we greatly reduced such errors by not mapping areas 15   that appears to be low-laying land and sediment bars in the SAR backscatter images. This can lead to the appearance of sporadic areas of bottomfast ice, separated by areas with a phase resembling that of bottomfast ice, but which in reality is land. In areas where the landmask (Wessel and Smith, 1996) does not appear to fit the coastline, 
[revised manuscript text omitted]

Scharroo, R., and Visser, P.: Precise orbit determination and gravity field improvement, Journal of Geophysical Research, 103, 8113-8127, 1998.

Screen, J. A., and Simmonds, I.: The central role of diminishing sea ice in recent Arctic temperature amplification, Nature, 464, 1334-1337, 2010.

Selyuzhenok, V., Krumpen, T., Mahoney, A., Janout, M., and Gerdes, R.: Seasonal and interannual variability of fast ice extent in the southeastern Laptev Sea between 1999 and 2013, Journal of Geophysical Research: Oceans, 120, 7791-7806, 10.1002/2015JC011135, 2015.

Selyuzhenok, V., Mahoney, A., Krumpen, T., Castellani, G., and Gerdes, R.: Mechanisms of fast-ice development in the south-eastern Laptev
Sea: a case study for winter of 2007/08 and 2009/10, Polar Research, 36, 1411140, 2017.

Smith, T. G.: Polar bear predation of ringed and bearded seals in the land-fast sea ice habitat, Canadian Journal of Zoology, 58, 2201-2209, 1980.

Solomon, S. M., Taylor, A. E., and Stevens, C. W.: Nearshore ground temperatures, seasonal ice bonding, and permafrost formation within the bottom-fast ice zone, Mackenzie Delta, NWT, Proceedings of the Ninth International Conference on Permafrost, University of Alaska Fairbanks, Fairbanks, Alaska, 2008, 1675-1680, 2008.

Stephenson, S. R., Smith, L. C., and Agnew, J. A.: Divergent long-term trajectories of human access to the Arctic, Nat Clim Change, 1, 156-160, 10.1038/Nclimate1120, 2011.

Stevens, C. W., Moorman, B. J., and Solomon, S. M.: Detection of frozen and unfrozen interfaces with ground penetrating radar in the nearshore zone of the Mackenzie Delta, Canada, Proceedings of the Ninth International Conference on Permafrost, University of Alaska Fairbanks, Fairbanks, Alaska, 2008, 1711-1716, 2008.

Stevens, C. W., Moorman, B. J., and Solomon, S. M.: Interannual changes in seasonal ground freezing and near-surface heat flow beneath bottom-fast ice in the near-shore zone, Mackenzie Delta, NWT, Canada, Permafrost and Periglacial Processes, 21, 256-270, 2010.

Stevens, C. W.: Controls on Seasonal Ground Freezing and Permafrost in the Near-shore Zone of the Mackenzie Delta, NWT, Canada, University of Calgary, 2011.

Stroeve, J. C., Serreze, M. C., Holland, M. M., Kay, J. E., Malanik, J., and Barrett, A. P.: The Arctic's rapidly shrinking sea ice cover: a research
synthesis, Climatic Change, 110, 1005-1027, 10.1007/S10584-011-0101-1, 2012.

Thomas, D. N.: Sea ice, John Wiley & Sons, Chichester, United Kingdom, 2017.

Tibbles, M., Falke, J. A., Mahoney, A. R., Robards, M. D., and Seitz, A. C.: An InSAR habitat suitability model to identify overwinter conditions for coregonine whitefishes in Arctic lagoons, Transactions of the American Fisheries Society, In press.

Vincent, F., Raucoules, D., Degroeve, T., Edwards, G., and Abolfazl Mostafavi, M.: Detection of river/sea ice deformation using satellite
interferometry: limits and potential, Int J Remote Sens, 25, 3555-3571, 2004.

Werner, C., Wegmüller, U., Strozzi, T., and Wiesmann, A.: Gamma SAR and interferometric processing software, Proceedings of the ERS-ENVISAT symposium, Gothenburg, Sweden, 2000, 1620

Wessel, P., and Smith, W. H.: A global, self-consistent, hierarchical, high-resolution shoreline database, Journal of Geophysical Research: Solid Earth, 101, 8741-8743, 1996.

Wilson, K. J., Falkingham, J., Melling, H., and De Abreu, R.: Shipping in the Canadian Arctic: other possible climate change scenarios, Proceedings of the International Geoscience and Remote Sensing Symposium, Anchorage, AK, 2004, 1853-1856, 2004.

Yu, Y., Stern, H., Fowler, C., Fetterer, F., and Maslanik, J.: Interannual Variability of Arctic Landfast Ice between 1976 and 2007, J Climate, 27, 227-243, 10.1175/JCLI-D-13-00178.1, 2014.

[Figure]

**Figure 1: (a) Oct - Mar (Freeze-up) and (b) Apr - Sep (break-up) monthly mean landfast sea ice extent (1976 - 2007) derived from sea ice charts based on optical instruments and SAR. The data for this figure was obtained from the National Snow and Ice Data Center (Yu et al., 2014).**

[Figure]

**Figure 2: Conceptual scheme of landfast sea ice where different regimes possess different levels of stability.**

[Figure]

**Figure 3: 52 Sentinel-1 interferograms derived from image pairs acquired between February and May, 2017. Numbers on images represent date ranges where the colors  signify the months of February-May **

[Figure]

**Figure 4: InSAR-derived map of**  **from 52 Sentinel-1 image pairs acquired** **. Letters A-D mark areas discussed in the text. Land is masked out in grey.**

[Figure]

**Figure 5: (a)** Landfast ice edge occurrence mapped for the period 1996–2008 over the Alaska Beaufort Sea (Mahoney et al. 2014).

correspond to  frequent landfast ice edge  referred to as " (d) map of landfast ice, grounded landfast ice, and bottomfast ice superimposed on the interferograms, and (e)

[Figure]

[Figure]

**Figure 7: Interferogram over the Nares Strait ice arch in 2017 covering the time period 6 - 12 Apr. (a). Smaller panels show consecutive interferograms within the box for 12 - 18 Apr (b), 18 - 24 Apr (c), 24 - 30 Apr (d), 30 Apr - 6 May (e), and 6 - 12 May (f). The dashed line represents the line separating the fast and moving ice in Figure 6b. Land is masked out in light gray.**

**Table 1:** Landfast sea ice stability regimes and assigned stability zones identified using InSAR and typical deformation rates

| | Landfast ice regime | Stability | Stability zone | Identified by | Deformation rate (cm/km/month) |
|---|---|---|---|---|---|
| 1 | Bottomfast sea ice (i.e., ice frozen to or in broad contact with the sea floor) | Completely stable. Ice is frozen to or resting on the sea floor restricting lateral motion. Vertical tide jacking may occur  as the ice thickens. | Bottomfast | No identifiable phase difference from the adjacent land | 0 |
| 2 | Floating ice sheltered in lagoons or fjords | High stability. Ice is largely enclosed by land and is sheltered from more dynamic ice. Deformation is dominated by cm- to dm-scale thermal creep and fracture. | Stabilized | Poorly defined, widely spaced fringes or abruptly reduced fringe spacing compared to offshore ice | ~0.1 – ~1 |
| 3 | Floating ice sheltered by grounded ridges or islands | Moderate stability. Ice is supported by point features largely inhibiting break out events. In addition to thermal creep, internal stress from more dynamic ice can propagate in between pinning points resulting in dm- to m-scale non-elastic deformation. | | | |
| 4 | Floating ice extensions | Low stability. Dominated by m-scale deformation from ice, wind, and ocean forcing. Persistent inelastic deformation can lead to accumulated strain on the order of tens of meters on time-scales exceeding several weeks. The ice may remain attached (Mahoney et al., 2004) or can break-off from the stabilized ice. | Non-stabilized | | < ~100 |

**Table 2:** Approximate area coverage of landfast ice  (in thousand km$^2$).

| Area | Bottomfast  | Stabilized | Non-stabilized | Total area of landfast ice | Non-stabilized / stabilized |
|---|---|---|---|---|---|
| Beaufort Sea | 2.5 | 35 | 30 | 65 | 0.86 |
| Chukchi Sea | 1.8 | 0.95 | 27 | 29 | 28 |
| East Siberian Sea | 5.1 | 40 | 81 | 126 | 2.0 |
| Laptev Sea | 4.1 | 33 | 164 | 201 | 5.0 |
| Kara Sea | 2.5 | 16 | 38 | 56 | 2.4 |

The bottomfast zone is constrained between its outer extent interpreted from the phase and the coast as interpreted from the backscatter scenes. The stabilized zone is constrained between its outer extent as interpreted from the phase and the bottomfast ice or the landmask (Wessel and Smith, 1996). The non-stabilized ice is constrained between the outer extent of  coherence and the bottomfast ice, stabilized ice, or the landmask.

---

## Referee Report (RR2)

[referee-annotated manuscript omitted]

---

## Author Response (AR2)

Dear reviewer,

Thank you so much for providing yet another very thorough review. The comments you have provided are very insightful. We agree with all of them and acknowledge that by following all of your suggestion have led to a substantially improved manuscript. We are very grateful for this.

The main improvements have been:

1) More structured method section, which now includes examples of the mapping of different zones. We agree that this was needed.
2) Expanded result section by splitting the interferogram figure into 5 separate figures. This has greatly helped providing much more insight into the mapping and mapping strategy.
3) Improved mapping of different areas. Here we realize that some areas were not optimally mapped. In other areas the decision to map certain areas were not clear. Here, we included example areas for each region, which has provided much more clarity to the mapping process.
4) We realize that mapping perfectly all areas on this scale is implausible and would also require local knowledge and other datasets for certain areas. We have now made it clearer that what we produce here is not a complete and perfect map of ice stability, but rather demonstrate the approach and application that potentially has to be complimented by other data products.

Please see detailed responses below. Thank you so much again!

With best regards,

Dyre Oliver Dammann

Dear authors,

Thanks for incorporating my comments and suggestions from the first round. The manuscript has already been improved. However, there is still some work to do in my opinion. I commented the manuscript with suggestions and questions. Main points are:

Still not clear how you classify fringe patterns. Many areas which look very similar are classified differently in different regions. See my comments in the manuscript to the Laptev Sea, for example. Therefore, I suggest to describe the manual procedure and add representative examples in Methods, as well as in the Results.

Structure of the paper can be improved. The entire chapter on the ice arch can be split and parts could be moved to the Introduction, Methods, Results and Discussion accordingly. The chapter on the comparison of your results with previous studies should be a part of Discussion. Your very results could be presented in a more detailed way, again with examples.

Furthermore, I insist on a professional proofreading to make the writing clearer and smoother. Many sentences could be simplified and shortened. I hope to see the updated version of your manuscript soon!

Best regards,

Your reviewer

Detailed comments:

landfast sea ice

Done

I'm not sure this information is important here. What does it really tell to the reader? Besides, you might reconsider Laptev Sea and East Siberian Sea zones as commented below. Better provide the information where is the largest area of stabilised and non-stabilised ice.

I absolutely agree. That is much better. I have now replaced with (P1,L16): "These seas also contain the largest extent of stabilized (East Siberian) and non-stabilized (Laptev) landfast ice."

not sure this was in the paper?

Great point. Taken out

I think this is included in the previous sentence

Agree. Taken out

this sentence seems to be not in place, also I would expand on what do you mean by "internally" and processes influencing this deformation

Absolutely, this was out of place. Now moved further down and changed to (P2,L6): "Although the landfast ice is stationary, it does deform at the cm- to m-scale on timescales of days to months due to forcing from wind, currents and drifting ice (Dammann et al., 2016)."

probably you should add which regions are you referring to here, or pan-Arctic? what about area decline?

This has been further clarified by changing to (P2,L1): "Similar to the drifting pack ice, landfast ice has declined significantly during the last few decades, in particular in terms of delayed freeze up in the Beaufort (Mahoney et al., 2014) and Laptev (Selyuzhenok et al., 2015) Seas as well as significantly reduced extent in the Chukchi Sea (Mahoney et al., 2014)."

choose one

Good point. Done

can you explain more detailed?

This has been further explained (P2,L3): "Later freeze up critically impacts stakeholders through reduced stability of the landfast ice in response to fewer grounded ridges that can withstand wind, ocean, or ice forcing (Dammann, 2017)".

Are these your own categories you are using in this study? Or they are common? In the latter case you should give some references and not say that you suggest these categories.

We are not aware of literature that utilize these categories, which is something we came up with based on this work. The categories have now been included in the text.

I would also list the categories in the text

Done

what properties?

Taken out

what means substantial here?

Taken out

it is not clear to me what different products are there, because you mention later backscatter change detection. So I would understand that the backscatter is the only product.

This has been changed to "different techniques", which includes backscatter gradient differencing (SAR), boxcar image cross correlation (SAR), mean-value temporal compositing (MODIS).

maybe add how exactly, because it is not obvious from the previous sentence - no motion, no topography is involved

I like that idea. Now included much more (P3,L16): "InSAR is dependent on similar acquisition geometry, scatterers, and atmospheric conditions at the time of the two acquisitions to retain a non-zero interferometric coherence, which ranges between 0 and 1. InSAR has been used to successfully map the boundary of landfast ice (Meyer et al., 2011) through identifying the ice that has not shifted more than a few meters over weeks and hence retain interferometric coherence. InSAR has also provided information pertaining to landfast ice dynamics (Li et al., 1996;Morris et al., 1999;Vincent et al., 2004;Marbouti et al., 2017) and topography (Dammann et al., 2017;Dierking et al., 2017) by evaluating the phase change between acquisitions."

using Sentinel-1 interferometry

Included

I don't understand what is meant here. What and how do you evaluate? Please be very clear in the objectives what are you doing in the paper

Largely taken out and now states (P3,L39): "We further explore limitations of the technology and possible applications."

I think this section lacks some clear structure. I propose to have 3-4 sub-chapters: 1) InSAR principles; 2) Sentinel-1 data you used; 3) data processing; 4) mapping of zones with examples. Maybe additional chapter with "validation" datasets (Mahoney et al., 2014 and potentially Selyuzhenok et al., 2015)-in this case you would not need to describe these in details in Results.

Also, the work with backscatter is missing totally in Methods and data.

Great suggestion. These subchapters have been included and the methods section restructured accordingly. Backscatter is now mentioned in the processing section (P4,L35): "The complex Sentinel-1 data was processed to obtain the backscatter in order to interpret features that sometimes can be visibly identified such as the landfast ice edge, fracturing, and ice roughness and types. The data was further processed for interferometry."

not sure you should mention study area in the header - it is pan-Arctic?

Good point. Taken out

comparison with what?

This has been changed to "validation" and this paragraph has now been moved to the new validation subsection of methods, which is now expanded upon and further helps clarify this.

I would maybe move this section to the first place

Done

not sure if you need this if you do not perform unwrapping

Taken out

according to the formula you should get a half of wavelength which is appr. 3 cm for the line-of-sight displacement. If you refer to lateral (horizontal), provide another formula with the incidence angle.

Taken out in response to previous comment

I would include this into the next sub-chapter where I would describe the approach you are using in more details.

Done

what was the time interval for interferograms? You have it on Fig.3 but it also needs to be in the text.

This has been included (P5,L6): "In this work we predominately utilize acquisitions with a temporal lag of either 12 or 24 days depending on data availability. For this timespan, the coherence over landfast ice was found to be generally high due to its stationary nature."

what about Sentinel-1 debursting?

This has been included (P5,L12): "The IW images initially consist of independent bursts and swaths which we combined to utilize the full extent of the acquisition. We further coregistered multiple acquisitions to ensure that the images cover exactly the same area with sub-pixel accuracy."

in Gamma software

Included

it would be nice to have representative and detailed examples on mapping of each class.

This has now been included

not clear to me what do you mean by that

Changed to: "including changing wind and ocean currents"

not clear what is absolute stability

This has been further clarified (P6,L3): "The zones themselves are based on relative stability in terms of whether the ice is anchored or sheltered. Determining absolute stability (i.e. whether an area is stable enough for a specific use, such as ice roads) would be problematic to determine from fringe density alone since there are many factors that affect fringe density in addition to stability, including changing wind and ocean currents, satellite viewing geometry, and the prevalent mode of ice deformation (Dammann et al., 2016). A measure of whether the ice is stable would also depend on the specific stakeholders and their dependence on stability. As an example, on shorter time scales, industry ice roads would be able to accommodate less strain than community ice trails due to different mode of transportation and user specific needs. Further steps to identify such thresholds are outlined in Dammann et al. (2018a)."

please simplify this sentence, it is enough to say that the near-zero phase change represents bottomfast ice.

This has been simplified and split (P5,L25): "Furthermore, Dammann et al. (2018c) showed that bottomfast ice can be mapped based on a near-zero phase change (since highly stable) where the ice is frozen to the seafloor. We here build on this work by suggesting that InSAR can be used to map three different zones of relative stability: bottomfast ice, stabilized ice, and non-stabilized ice (Table 1)."

why to consider 4 classes at all? Again, is it established classification or yours? To my mind, having 4 classes and then reducing them to 3 is a bit confusing. It is not possible in this study to distinguish two types of stabilised ice - so just mention in the very beginning that the stabilised class in your study can be attributed to two mechanisms of formation

Agree. This is a good point. This has been taken out and the classes merged.

some words about this class for completeness.

This has been taken out and more was included about this in the introduction (P2,L10): "These categories include (1) bottomfast ice, (2) floating ice enclosed in lagoons or fjords or sheltered by point features such as grounded ridges or islands, and (3) floating ice extensions (Table 1)"

please merge this paragraph with 2.1. See also my general comment to the Methods section

Good suggestion. Done

please find a better title - you use the same one in the Methods (and it sounds as a method). I would expand this section - these are your main results and you dedicate them 0.5 page. Include more areas in the description and describe them more deliberately.

Agree. In response to a comment further down, what was Section 3.2 has now been moved to the discussion section. The results now consist of two sections "Evaluating landfast ice stability" and "Large-scale mapping of stability zones"

this sounds to me as Conclusions.

Good point. This has been moved to conclusions

I would add detailed examples (figures) for some key regions.

This has now been included

introduce the areal calculation of ice classes - it appears out of sudden. Also specify that the area calculations are not complete because of some data gaps.

Good idea. This has been included to introduce:

Section 3.1 (P6,L23): "We constructed a series of Sentinel-1 interferograms along the coastlines of five marginal seas of the Arctic Ocean during 2017: the Beaufort, Chukchi, East Siberian, Laptev, and Kara Seas. As seen in the in the interferograms (Figures 4-8), the landfast sea ice varies substantially between the seas in terms of the extent and interferometric fringe density."

And Section 3.2 (P7,L24): "Interferograms enabled the mapping of landfast ice stability zones based on subjective interpretation of interferometric fringes (Figure 9). The resulting stability map allows for a large-scale comparison and analysis of bottomfast, stabilized, and non-stabilized landfast ice within and between the different seas. For this comparison, we have listed the 2017 area extent of each stability zone and marginal sea in Table 2. However, it is important to note that this list is not complete since this analysis omitted some island groups and included some data gaps."

specify that you are talking about the area

Done

I thought all classes are based on that -including bottomfast ice. Maybe this class is the least ambiguous but still it is subjective.

Absolutely. Good catch. This has been included and rephrased (see two comments up)

Why "" on classes?

Removed.

explain why and provide examples

This is now been further clarified by including (P7,L7): "In contrast to the Beaufort and Chukchi Seas, stabilized ice extends several tens of km offshore without being sheltered by coastline morphology or islands (Figure 6c). These large areas also lack clear indications of the presence of grounded ridges as found by smaller areas of stabilized ice (Figure 6d) and in the Beaufort Sea."

And

(P7,L15): "Some regions of the eastern Laptev Sea lack a clear discontinuity, but at the same time feature locally reduced fringe density, indicative of stabilized ice (Figure 7c). These areas, we also consider to be stabilized (Figure 7c), but possibly as a result of different ice type or thickness rather than through grounding or sheltering."

I would say that in Kara Sea it is not vast areas, i.e. very comparable with Beaufort and Chukchi Sea

Agree. This has been rephrased

Chukchi Sea also belongs partly to Russian coast. I would remove this comparison

Done

please mark the archipelago on the map.

Done

But you classified most of this area as stabilised although big part of it doesn't look like this.

Some areas may have a significant fringe response, but when looking in contrast to the surrounding ice marked non-stabilized, the difference is apparent. This is now more visible with the introduced Figure 8c.

Also the region around Pyasina River seems to be partly incorrect.

To us, this seems to be correct. The interferogram over this area has been enlarged in Figure 8d.

Bely Island seems to have some bottomfast ice from the west and south.

You are right, the southern part should be marked as bottomfast. However, along the western part seems to be due to an inadequate landmask. The southern part has now been mapped.

Sharapov Shar Bay (west Yamal peninsula) seems to have stabilised ice.

Absolutely, this has been changed

just largest by area?

Yes, changed

C is classified as non-stabilised on Fig. 4. Mistake?

Yes, not sure why this was suddenly gone. Made sure this area appears as stabilized now.

to my mind this section belongs to Discussion

This has been moved to the discussion

not the stability zones but your mapping of them

Rephrased

but this area is identified based on the April interferogram? Say that the February interferogram was not helpful in mapping because likely it is too early for stabilising...From here also that the area D is likely larger than shown. the current wording about triangular area is confusing.

This has been changed (P9,L4): "The full area extent of the stabilized ice cannot be established due to limited data availability in the region and thus the surrounding interferogram had to be acquired as early as February before this region had stabilized."

the island is quite far away from the area of stabilised ice, i would not connect them in the sentence.

Taken out

could you map the shoals (they are several) on your figures using the bathymetry information?

This has now been done and added to a new figure (Figure 11).

I actually thought about some similar analysis here as in Beaufort Sea, with multiyear fast ice edge position available from this study

I don't understand what do you want to say here in addition to what already has been said

Taken out

why future tense? please make the using of tenses consistent

Changed

it is not identified on the map - either insert the node A in the text or put the name on the map.

Included on the map. This sentence has also been moved to the methods section in response to other suggestion.

why do you need this sentence here? it should be clear from the methods already.

This context has been changed by moving the section to the discussion. The sentence is also changed to (P8,L22): "We examine three acquisitions from 8-17 Apr along the Beaufort Sea coast.."

why? consistent to what? please elaborate

This has been changed. Please see next comment.

what types? here you are talking only about landfast ice edge - so the contrast should be between fast ice and water/pack ice? also this I don't understand. In general you should provide a reference to this sentence.

This has been changed and a reference provided (P8,L24): "It is worth noting that relying backscatter to discriminate landfast or drifting ice only works in cases where there are noticeable differences in backscatter between the landfast and drifting ice or a severely deformed landfast ice edge as a result of shear interaction with the pack ice (Druckenmiller et al., 2013)."

this paragraph seems to break the flow between the previous and the following paragraphs. Please consider restructuring. Also, I don't really understand why do you need to use the term "discontinuities", because you have already mapped/classified fast ice zones for all regions in the same way. You can just refer to these zones instead of introducing discontinuities and then concluding that they correspond to the stability zones.

This has been totally restructured in what is now Section 4.1. We also now refer to the zones as you suggest.

as I said in the first review, I don't see it with the node B - the border between stabilised and non-stabilised does not coincide with this node. And actually the node is more pronounced to the east of your current marking -and there is no coincidence there as well. In case of the node A - the stabilised zone

almost coincides with the overall fast ice extent, so it is not representative to my mind. Node C - maybe yes.

We have totally restructured this section (now 4.1) and now discuss the similarity with the landfast ice edge and also the mismatch with node B.

this should be merged with the paragraph on the lines 17-23, as you say already there about ridges.

Done

it is still not clear to me, sorry. If it is an important message, please provide examples on the figure and clarify the sentences again. I would also not expect to have a straight line anyway, you don't need to specify it.

We have taken out the mention of a straight line, further clarified the sentence and provided examples in the figure.

this we know already - the analysis is done for all regions in the same way.

Taken out

what about distinguishing stabilised and non-stabilised zones from backscatter? You should mention this and discuss the differences you see on backscatter and interferometry. And also this sentence seems to be not in place.

The sentence has been moved and we have now introduced arrows in Figure 10a which indicates locations of grounding points on the backscatter image as obtained from InSAR. This demonstrates the lack of features that signify the change in stability.

if it is a finding of this study you should not include this reference. Otherwise, say that another study already reported on this problem.

This is now stated (P9,L12): "Dammann et al. (2018c) determined that in some instances, bottomfast ice has to be approximated on the sub-km-scale due to ambiguities associated with low fringe density or fringes parallel to the bottomfast ice edge."

this should be highlighted in Methods with examples

Done

what is the source of the coast mask?

The global, self-consistent, hierarchical, high-resolution shoreline database (GSHHG). This has now been referenced earlier, which we think helps clarify the source of the landmask throughout

does this sentence belong to the pre-previous one? Or using backscatter leads to this effect? Please clarify and provide a figure with an example.

This has now been moved to the methods section with an example (P5,L30): "Bottomfast sea ice appears with near identical phase values to low-lying coastal areas, but is discriminated from land by identifying the coastline using the backscatter signature in a composite image with backscatter and phase (Figure 3b). Subtle coastal features such as sediment bars are often not captured by the landmask (Wessel and Smith, 1996), which can give the wrongful appearance that (1) areas of near-zero phase should have been mapped as bottomfast and (2) bottomfast ice appear in sporadic areas along the coast separated by floating ice."

please provide more information on the origin of the landmask. Is it the same coast mask you are talking about before?

We have here introduced the landmask now in the method section with reference, which clarifies the origin of the landmask.

because it changed since 1996 or because of errors?

Both: "…coastline due to errors or coastline changes…"

I might disagree because the area of the bottomfast ice is very small compared to the area of the whole fast ice and errors in mapping could actually lead to significant variations in the areal estimations.

This has been changed: "…hence mapping on a pan-Arctic scale will inevitably contain inaccuracies that are likely to impact our findings."

examples?

Included now: Figure 6c and 7c

different from western Arctic? or what is meant here? please expand here.

Yes, included.

How can you tell that the ice is non-stable there? I don't see actually an example of "landfast ice seaward of offshore islands". Please provide examples.

This is now clarified with examples (P9,L35): "Such ice regime is expected to feature reduced dynamically-induced strain (and therefore fewer interferometric fringes) in non-stabilized ice making it appear more stable. This is visible in the different fringe densities of the non-stabilized ice in Figure 4d and 6d."

spectrum of stability for sure, it is clear that stabilised/non-stabilised is way too broad (although much better than nothing, I agree), but is it really possible to identify more areas with this method? Maybe better say that it might be necessary to evaluate fringe density differently in different regions, also using additional information, i.e. bathymetry, etc

Expanded upon this (P9,L26): "Expanding upon the classes presented here would likely require a different set of evaluation criteria for fringes in depending on regions. Additional data such as bathymetry would also likely strengthen such analysis."

Add an introducing sentence before - "We focused on some examples of possibly erroneous classification...

Added this (P10,L5): "We focused on some examples with possibly suboptimal classification."

but there are clearly some regions of lower fringe densities, why do you ignore them?

We agree a couple of areas here should (and now have been) be identified as stabilized following our consistent scheme. We have now further clarified (P9,L34): "We focused on some examples with possibly suboptimal classification. One potential candidate for reclassification is landfast ice in sheltered bays such as the Khatanga Gulf in the western Laptev Sea, which exhibited predominately high fringe densities (Figure 7a) and was hence identified largely as non-stabilized despite being nearly landlocked (Figure 9)."

again, I think that the introducing of additional zones is not really possible (or not even needed) but just to use additional information (coastal set up) for reclassifying / cautious mapping

This has now been clarified (P9,L39): "Such additional classification would depend on other datasets such as a landmask or bathymetry to identify level of restricted ice movement in response to likely forcing conditions."

I disagree with classifying all the area as non-stabilised. How does the fringe density on interferogram 10-22 differ from the density on interferogram 12-24? Also, there is clearly an area of different fringe density in the middle of 12-24 as well as the eastern part of 8-2 which is classified as non-stabilised. Also, why the Buor-Khaya Bay is non-stabilised?

We initially did not mark these areas as stabilized based on our criteria of needing a visible fringe gradient or having fringes difficult to determine. However, we agree it may be worth outlining the areas you indicate since they quite significantly distinguish themselves from the surrounding ice. We have therefore included the areas you specify in the stabilized category. Their location is largely now also explained by the validation in what is now Figure 11.

In the East Siberian Sea the entire Chaunskaya Bay is classified as stabilised although a significant part of it features dense fringes. Whereas in the Laptev Sea areas of much wider fringes are classified as non-stabilised.

We moderated to (P10,L4): "…we classify much of the landfast ice in this region as non-stabilized". We have also changed the high fringe density in Chaunskaya Bay to non-stabilized

again, I don't see how can you compare the "stable" from Eicken and "stable" from your study. Maybe their definition of stable can well fit to your non-stabilised criterion? Therefore, I don't think there is an apparent contradiction to Eicken. We simply don't know what exactly they call "highly stable", right? Or provide more information on Eicken's stability

This has been changed to (P10,L5): "this suggests that the criteria for stabilized ice as used here is different than in Eicken et al. (2005) and can provide new information related to stability in the region."

see the previous comment. Maybe just say that your study may provide some new insights into ice stability in this region.

Done. See previous comment

and the other part? based on what do you suggest it?

This sentence has been slightly moved and changed to incorporate this (P10,L6): "Based on the overall fringe counts and patterns, the majority of the phase response is due to lateral displacement and potentially only partially due to vertical displacement (circular fringe patterns with low density – see Dammann et al. (2016) due to tidal motion.)

reconfirm the one month please. or generalise to the entire winter period to make it relevant to your study. Maybe refer to break out events, possible in the region?

This has been expanded upon (P10,L9): "This would be consistent with a recent SAR backscatter analysis of landfast ice in the Laptev Sea (Selyuzhenok et al., 2017), which showed that areas identified as landfast ice in operational ice charts may actually contain pockets of partly mobile ice. This was shown for the month after initial landfast ice formation, but could possibly result in more dynamic ice throughout spring due to reduced ice thickness."

how do you know that?

We found this in our 2016 work. The citation is now included

February?

Yes, due to the new interferogram introduced. Changed.

this paragraph contains many different aspects which are not related to each other. Try to make a smoother flow

Good point. This paragraph is now made much smoother and consistent (P10,L24):

"Sentinel-1 IW imagery are predominately acquired over land, hence it is likely not possible to construct interferograms away from the coast to cover extensive landfast ice approaching the 250 km IW swath such as that in the East Siberian Sea. The data availability of these images further restricts the temporal baseline between images to a minimum of 12 days, shorter than past work to identify landfast ice (Mahoney et al., 2004;Meyer et al., 2011;Dammann et al., 2016). Further studies should investigate the effect of different temporal baselines on the stability product. A shorter baseline will result in higher temporal resolution. However, with a shorter baseline (e.g. Sentinel-1 6-day baseline), the mapping of the seaward landfast ice edge may incorporate stationary pack ice. A longer baseline will result in lower interferometric coherence. With a 12-day baseline, some regions already feature consistent coherence loss such as the Kotzebue Sound region. Such regions can most often be identified through a spatially gradual progression from high coherence to a complete loss of coherence, where an exact map of landfast ice type boundaries is not possible. It is worth mentioning that this technique can only be used before the onset of melt when widespread coherence loss occurs, hence it is not possible to evaluate the retreat of bottomfast ice or reduction of ice stability in response to melt."

maybe you also should mention that future studies should investigate the effect of varying temporal baseline on the stability mapping?

Done (see above)

maybe better mark the entire Bay and not the town? It took me a while to understand that you are talking about the southern part of the Bay and not about the point with arrow

Good point. Done

I'm not sure it is gradual but the boundary is hard to see, true

Changed to "spatially inconsistent progression"

to my mind you should split this whole chapter between Introduction, Methods, Results and Discussion. In the beginning you are talking about importance of ice arches, previous studies on them etc. Then you provided clear results of your analysis.

This is a good idea. This has been done

this chapter sounds like Conclusions and should be merged with them

This has been done

is "and" missing here?

No, but changed the sentence to make it more readable (P12,L15): "Bottomfast ice is important because it helps aggregating subsea permafrost, which serves to constrain the location of permafrost-rich shorelines"

i don't get the connection

Taken out

"appears to be meaningful in most regions"? I don't think it was proven

Taken out

sea ice

Done

ice stability

Done

see my comment on that above. I'm not convinced that the nodes of Mahoney et al. 2014 correspond to your classes boundary. I would describe it more cautiously.

We have now described this better. See response to earlier comment

there are no islands as I see

There are a number of islands stabilizing the ice cover especially in the form of barrier islands enclosing lagoons.

as also previously said, I think that not the additional zones would help but adaptation of the method to the different regions and additional sources of information. Indeed, more zones could be distinguished based on the fringe rates but you argue that you want to keep the approach simple. In this case I don't think you can come up with more zones than you have already.

We agree. We have changed this to (P11,L31): "This makes it challenging to directly adapt the proposed scheme to the East Siberian and Laptev Sea without further incorporating an InSAR time series analysis and local knowledge of the region. Introducing other data pertaining to coastal morphology, bathymetry, and regional wind and ocean forcing climatology would also further strengthen the stability analysis."

is it anticipated from your study or from the referred one? either move to Discussion or remove the reference.

Removed

consider removing references from Conclusions

Taken out all

I would remove d) or combine it with c)

Done

this caption is a little messy

Cleaned up now

where do the rates come from?

These came from assessing the interferograms in this work, but this column has been removed since we found it somewhat misleading that the zones are based on the rate values.

what is this?

Specified to: "Area fraction: non-stabilized / stabilized"

Ice

[revised manuscript text omitted]

**2.4 Mapping of landfast ice zones**

In this work, we evaluate relative ice stability based on fringe spacings within individual interferograms. This allows us to identify variations within an area imaged under largely the same conditions. Trends from higher to lower fringe density will, in such cases, likely correspond to increasing ice stability. We hypothesizeTherefore, interferograms can provide information related to the relative ice spatial variations. Meyer et al. (2011) demonstrated that interferometry can be used to map the landfast ice based on a coherent phase response. Their work also suggested that fringe patterns are significantly impacted by grounded ridges by reduced density can reveal of the fringes. Furthermore, Dammann et al. (2018c) showed that bottomfast ice can be mapped based on a near-zero phase change where the ice is frozen to the seafloor. We build on this work by suggesting that InSAR can be used to map three different zones of relative stability zones: bottomfast ice, stabilized ice, and non-stabilized ice (Table 1). The often strong fringe gradient leading to an area of near-zero phase change has been shown to represent the boundary of where the ice is frozen to the sea floor and can subsequently be used to map bottomfast ice (Dammann et al., 2018c). In Table 1, the two sheltered regimes will both lead to reduced fringe density and can be difficult to discriminate based on InSAR data alone.

These regimes are therefore assigned to the zone "stabilized ice". The three zones (i.e. bottomfast ice, stabilized ice, non-stabilized ice) 
[revised manuscript text omitted]
. The respective master images exhibit a sharp discontinuity in backscatter (see arrows in Figure 5b) along the general location of the landfast ice edge from Figure 5a and can be assumed to be the landfast ice edge. Determining the landfast ice edge can in some instances be achieved by evaluating a single amplitude image as here, but is not consistent and only works in cases where there are stark discontinuities in backscatter due to different ice types or a severely deformed landfast ice edge. The interferograms indicate in this case a similar landfast ice edge by a complete loss of coherence seaward of the discontinuity apparent in Figure 5b (Figure 5c). These interferograms reveal a wide range of fringe densities, ranging from near constant phase for areas close to the coast to the point where fringes are dense enough to almost merge near the landfast ice edge. It is also apparent that the fringe density does not linearly increase with distance from the coast, but rather changes along two distinct discontinuities.~~

~~**One discontinuity separates the area of near-zero phase change from an area with relatively low fringe density. This discontinuity indicates the boundary between bottomfast and floating ice as two of these interferograms were validated both on Elson Lagoon near Utqiaġvik and the Colville Delta (
[revised manuscript text omitted]

---

## Author Response (AR3)

Dear Reviewer,

Thank you again for providing a thorough review, which helped once gain to greatly improve this paper. We have made most of your suggested changes and you can find replies to each of your detailed comments below. We had several people reading over to provide corrections to the English language prior to the last resubmission and were confident this would suffice. However, since some errors made it through last time, this time we obtained professional proofreading.

Thank you again for your help with the manuscript.

Best regards,

Dyre Dammann

Detailed comments:

pan-Arctic?

Done

please check punctuation

Included a comma

I'm not sure about this sentence. Habitat for fauna? Then I don't see how it could evolve. If you mean humans - then the previous sentence has already introduced that.

Taken out

check punctuation.

This sentence was split (P1,L13): "This first comprehensive assessment of Arctic bottomfast sea ice extent has revealed that most of the bottomfast sea ice is situated around river mouths and coastal shallows. The Laptev and East Siberian Seas dominate the aerial extent, covering roughly 4.1 and 5.1 thousand $km^2$, respectively."

mapping scheme

Changed

for

Done

check punctuation

Done.

check the tenses throughout the manuscript. There is a mix between present and past tenses.

Done. Sentence simplified (P1,L19): "In a case study of the Nares Strait, we demonstrate that interferograms may reveal early-warning signals for the break-up of stationary sea ice."

is this part necessary? it makes the sentence too long and difficult to grasp. If you want to leave it, make it another sentence.

This has been changed (P1,L31): "It is further recognized that sea ice conditions for future Arctic marine operations will be challenging, and will require substantial monitoring and improved observations (Arctic Council, 2009). This improvement will require observations at a local and regional scale, in order to provide assessment of environmental hazards and effective emergency response (Eicken et al., 2011)."

this relates more to the introduction part on ice arches. Nothing to do with remote sensing.

Good catch. Moved

SAR backscatter

Done

typically does not give? let's not be so radical :)

Agree :) Changed

to the landfast ice stability

Took out the subsections in the results in response to other suggestion

with?

Changed

InSAR phase stability, referred to as InSAR coherence, largely depends on topography coupled with perpendicular baseline, and temporal stability of the scatterers on the ground surface. Coherence ranges between 0 (pure noise) and 1 (no noise), and serves as a measure of the quality of the interferogram.

Thank you for providing this. This has been included. All of this and info about coherence is moved to Section 2.1.

InSAR coherence has been successfully used to map the landfast ice edge through identifying areas of slow motion featuring high coherence?

Good sentence. This has been replaced, but the mention of coherence has been taken out since we now choose to introduce coherence first in Section 2.1.

local case studies

Included

this has also vague relation to the chapter's focus. Consider removing or find a better place for this part. The last sentence sounds like conclusion.

Agree. This has been moved to conclusions

In this study,..

Changed

information on ice stability which is relevant to

Included

their applicability?

Yes, changed

stability zones?

done

SAR

Yes, changed

Combine with the part on coherence in the previous chapter. Decide where it is better to introduce it.

Good point. All is placed in Section 2.1.

I think this should go to the data processing. You do not need to refer to interferograms here, just general S-1 data coverage used in this study etc...

Good point. This has been changed (P4,L19): "We acquired over one hundred SAR images, covering nearly the entire continental coastlines of the Beaufort, Chukchi, East Siberian, Laptev, and Kara Seas."

... that is why we focus on these coastlines?

Yes. This is now changed (P4,L21): "In this work, we focused on the Alaska and Russian marginal seas of the Arctic Ocean. These coastlines have high economic significance for the shipping and natural resource industries, and also feature dynamically diverse ice regimes."

check the order of words in the sentence

This has been changed (P4,L27): "In addition to images obtained for the large-scale mapping of stability zones, a series of six consecutive image pairs were acquired covering the Nares Strait and the break-up of an ice arch during spring 2017."

large-scale mapping?

Included

This should go Discussion

Agree. Done

break-up

Changed

this should go to Results

Moved to discussion as there is no mention of the ice arch in the results.

to visually identify ice features (...)

Done

Maybe add more details on backscatter processing. All acquired images were processed to backscatter or only some?

Done (P4,L31): "All the complex Sentinel-1 data was processed to obtain backscatter, in order to visually interpret features (e.g., landfast ice edge, fracturing, ice roughness and types)."

ice thinning?

Done

how do you make sure that the melt onset did not happen yet?

We can't really. We have moderated this sentence to (P5,3): "In this work, we obtained images as close to late April as possible. This timeframe was found to be ideal for our purpose, as ice thickness is near its maximum, leading to maximum stability and minimizing impacts from the onset of melt."

We also discuss the impacts of melt now further above in Section 2.3 (P4,L38): "This was likely predominately due to surface melt, as air temperature reached above freezing between SAR acquisitions. Other possible contributing reasons for coherence loss in this region could include ice motion, subsurface ice thinning from river runoff, and low signal-to-noise ratio."

(i.e. different time period)?

Done

Did you also use Gamma for backscatter processing?

Yes, included.

And here you can add that at all 52 interferograms were produced etc

Included (P5,L14): "The result was fifty-two interferograms, covering almost the entire coastline between the Canadian Archipelago and the Barents Sea."

stability zones?

Included

Do you mean here within one interferogram? why largely the same conditions? They are the same.

Yes. We argue not completely the same as forcing conditions will vary within one scene. This has been further clarified by including (P5,L17): "This allows us to identify variations within an area imaged under largely the same conditions (e.g., close to the same wind and ocean forcing)."

not clear to me

Changed to (P5,L19): "Therefore, interferograms can provide information related to the spatial variations in stability."

This sentence is difficult to understand. Do you start here with difficult cases, opposite to the Fig. 3a? Then introduce the coastline mask, that it can match poorly to the actual position of the coast, then that to overcome this you use backscatter. Provide more details on the work with backscatter.

This is now further clarified (P5,L27): "The shoreward boundary of bottomfast ice is difficult to obtain from the interferogram alone, since the phase signatures over bottomfast ice and low-laying coastal areas are similar. The use of the landmask (Wessel and Smith, 1996) is not ideal, since subtle coastal features such as sediment bars are often not captured. We therefore delineate the coastline (i.e., shoreward boundary of bottomfast ice) using the backscatter signature in a composite image with backscatter and phase (Figure 3b). Plotting bottomfast ice with the landmask can thus give the wrongful appearance that (1) areas of near-zero phase should have been mapped as bottomfast and (2) bottomfast ice appears in sporadic areas along the coast separated by floating ice."

make a better separation between parts on mapping of different classes. Maybe new paragraph?

Done

Where are examples on them? Before you use examples without identifying their geographic locations, be consistent.

Changed

I don't understand this sentence. why suddenly freshwater lakes?

Taken that part out as it was unnecessary

Why to use passive voice (here, before and after)? You mapped the zones!

Changed

what means remaining here?

This is now rephrased (P4,L38): "Non-stabilized ice is identified as landfast ice (i.e., areas featuring relatively high interferometric coherence) otherwise not marked as bottomfast or stabilized ice. Non-stabilized ice commonly features clear fringe patterns (Figure 3e).

it can never be 0 in reality. It is always above zero, and areas of low coherence can feature values of say 0.2. Just say "relatively high".

Good point. Changed

I understand what you mean here but it might be very confusing for a new reader and not necessary in my opinion.

Good point. Especially with rephrasing the sentence (see second to last comments above) this is already explained. Took this out.

I think this can be moved from Methods, sounds like Discussion, and I'm sure in Discussion you talk about that.

Good idea. This has been moved to Section 4.1.

Landfast

done

Results

done

Landfast ice stability zones?

done

consistent with the location of grounded ridges? Keep it simple. But also question - how do you know about ground ridges here?

We explain this further in the discussion. However, all we are intending to state here is that these patterns resemble a pattern we would expect near grounded ridges. This is now better clarified (P6,L21): "The line of discontinuity features several seaward points, an expected pattern surrounding grounded ridges. This is because grounded ridges

result in a shoreward increase in stability that does not extend to areas immediately to the side of the ridges (the along-shore direction). Examples of likely grounding points are indicated with white arrows in Figure 4b, and similar patterns are also apparent near the Mackenzie Delta (Figure 4c)."

what arrow exactly?grounding points?

This is now better described (see last comment).

?

Taken out

mark it on the figure. is it Bering Strait?

Changed.

why consistent?

Taken out

follows?

Changed

?

Taken out

Laptev Sea

Changed

Pyasina

Changed

not clear what you define as "around archipelago" and "confined by the islands". Mark non-stab. ice in Figure 8c.

Clarified (P7,L13): "In these archipelagos, the ice confined by islands is largely stabilized (Figure 8c)."

Areal analysis? Maybe you don't need a separate section for this...Think about it.

Agree. Good idea. We took that section out.

I would combine Fig. 9 with previous figures that the reader can directly compare interferograms and your mapped zones without need to turn pages back and forth.

Not a bad suggestion. However, we are concerned the figures would not fit on a page. The stability maps would therefore also be enlarged and take up more space than what they strictly need to. We think it works best like this, but are open to exact suggestions for how to rearrange this if need be.

calculated the area of each stability zone

Changed

the area calculations are not complete

Changed

part of this is in the section before. Also, if you use the table, why to mention every number then in text?

Shortened this considerably and took out redundancy. Also, took away numbers as you suggest.

InSAR technique?

Done

map?

Done

Other

Changed

The backscatter exhibits a visible discontinuity...(I don't see it sharp everywhere)

Changed to (P8,L31): "These images exhibit, in certain locations, a sharp discontinuity…"

areas of known stabilisation points?

Done

backscatter mosaic from three images (8, 15, 17 April)

Done

shorten the sentence. And move it to the later part where you talk again about backscatter.

Done

identified by backscatter

Done

is

Done

?

Changed to (P9,L3): "Certain sections of the border between stabilized and non-stabilized ice extend relatively far from the coast (see black arrows in Figure 10d)."

I would make a new paragraph

Done

mapping discriminated?smth is wrong here

Changed (P9,L7): "Although the landfast ice edge can in some instances be mapped using a single backscatter image, stabilized ice cannot easily be discriminated from non-stabilized ice."

you could mark borders and dates of interferograms on this figure as well.

Done

established?

Changed to "determined"

surrounding what?adjacent?

Changed to "one interferogram"

I guess it is a tautology

Good point. Taken out

earlier than where?

Changed to: "earlier ice formation than surrounding areas"

as I said before, this needs to be explained in more details in Methods and/or Results

This is now clarified further in response to earlier comment

can you expand what exactly are the differences? forming of polynya?

Included (P9,L37): "This generally results in reduced ice forcing and landfast ice strain in contrast to the western Arctic."

this should go to Results

This has been moved to results and a new section, Section 3.2. The remaining of this subchapter was cut down and split where a small part made it into the conclusions and a part into the introduction.

please add scale bar

Done

this needs better phrasing. Can you show backscatter-derived landmask vs "official" landmask? Is the white outlined areas indeed ice or land, I'm confused.

This has been further clarified: "Phase/backscatter composite near a delta. This example exhibits a poor match between the landmask (transparent black shading) and low-laying coastal areas. Here, bottomfast ice (white outline) had to be mapped against the coastline, as identified in the backscatter data."

It would be difficult to delineate a landmask based on the SAR imagery alone and this is part of what causes the potential bias. We are now explaining this further in the discussion where we have included (P9,L24): "We have reduced such errors by not mapping areas that appears to be low-laying land in the SAR backscatter images. However, discriminating between ice and low-laying land can be difficult based on strictly SAR. Here, other remote sensing systems such as optical could be applied to further reduce biases from coastline errors. In areas where the landmask does not appear to fit the coastline due to errors or coastline changes, mapping intricate coastal morphology can be a time-consuming task—hence mapping on a pan-Arctic scale will inevitably contain inaccuracies. It is also worth mentioning that the other stability zones are mapped against the landmask, also likely resulting in errors. However, as the extent of these zones are larger, the relative contribution of such errors will be much smaller."

please mark b),c), and d) on a) for the orientation, here and on the following figures

Done

why not to give legend as on Figure before?

Done

mark dates of images on the figure

Done

[revised manuscript text omitted]